palaeontology

Erythrosuchidae, Permo-Triassic mass extinction, osteology, *Vjushkovia triplicostata*

**Author for correspondence:**
Susannah C. R. Maidment
e-mail: susannah.maidment@nhm.ac.uk

# The postcranial skeleton of the erythrosuchid archosauriform *Garjainia prima* from the Early Triassic of European Russia

Susannah C. R. Maidment[1,2], Andrey G. Sennikov[3,4], Martín D. Ezcurra[2,5], Emma M. Dunne[2], David J. Gower[1], Brandon P. Hedrick[6], Luke E. Meade[2], Thomas J. Raven[1,7], Dmitriy I. Paschchenko[3] and Richard J. Butler[2]

[1]The Natural History Museum, Cromwell Road, London SW7 5BD, UK
[2]School of Geography, Earth and Environmental Sciences, University of Birmingham, Edgbaston, Birmingham B15 2TT, UK
[3]Borissiak Paleontological Institute RAS, Profsoyuznaya Street 123, Moscow 117647, Russia
[4]Institute of Geology and Petroleum Technologies, Kazan Federal University, Kremlyovskaya Street 4, Kazan 420008, Russia
[5]Sección Paleontología de Vertebrados, CONICET—Museo Argentino de Ciencias Naturales 'Bernardino Rivadavia', Ángel Gallardo 470 (C1405DJR), Buenos Aires, Argentina
[6]Department of Cell Biology and Anatomy, School of Medicine, Louisiana State University Health Sciences Center, New Orleans, LA 70112, USA
[7]School of Environment and Technology, University of Brighton, Lewes Road, Brighton BN2 4GJ, UK

SCRM, 0000-0002-7741-2500; MDE, 0000-0002-6000-6450; EMD, 0000-0002-4989-5904; DJG, 0000-0002-1725-8863; BPH, 0000-0003-4446-3405; LEM, 0000-0001-7829-5193; RJB, 0000-0003-2136-7541

Erythrosuchidae were large-bodied, quadrupedal, predatory archosauriforms that dominated the hypercarnivorous niche in the aftermath of the Permo-Triassic mass extinction. *Garjainia*, one of the oldest members of the clade, is known from the late Olenekian of European Russia. The holotype of *Garjainia prima* comprises a well-preserved skull, but highly incomplete postcranium. Recent taxonomic reappraisal demonstrates that material from a bone bed found close to the type locality, previously referred to as '*Vjushkovia triplicostata*', is referable to *G. prima*. At least, seven individuals comprising cranial remains and virtually the entire postcranium are represented, and we describe this material in detail for the first time. An updated

phylogenetic analysis confirms previous results that a monophyletic *Garjainia* is the sister taxon to a clade containing *Erythrosuchus*, *Shansisuchus* and *Chalishevia*. Muscle scars on many limb elements are clear, allowing reconstruction of the proximal locomotor musculature. We calculate the body mass of *G. prima* to have been 147–248 kg, similar to that of an adult male lion. Large body size in erythrosuchids may have been attained as part of a trend of increasing body size after the Permo-Triassic mass extinction and allowed erythrosuchids to become the dominant carnivores of the Early and Middle Triassic.

# 1. Introduction

Erythrosuchidae is a clade of archosauriform archosauromorph reptiles that radiated to occupy the apex terrestrial predator niche in the aftermath of the devastating Permo-Triassic mass extinction (PTME) [1–3]. Large-bodied (2–5 m in length) and quadrupedal, erythrosuchids possessed large skulls relative to their body size [4–6]. A large cranium appears to have evolved in concert with other carnivorous adaptations (e.g. dorsoventrally tall and subrectangular cranial profile) and is perhaps associated with the clade evolving hypercarnivory [6]. Erythrosuchidae was one of the earliest archosauromorph groups to diversify after the PTME and comprises seven to nine valid species known from the latest Early Triassic to Middle Triassic of South Africa, China and Russia [4–11], with undescribed material also reported from India [12]. The earliest members of the group are *Garjainia prima* and *G. madiba*, from the late Olenekian of Russia and South Africa, respectively, and *Fugusuchus hejiapanensis* from the late Olenekian or early Anisian of China [6,8,9,13–15].

The holotype of *Garjainia prima* is a well-preserved specimen comprising a skull and an incomplete postcranium from Kzyl-Sai II, near Orenburg, Russia, and has received detailed study [9,13,15]. A second erythrosuchid, '*Vjushkovia triplicostata*', from Rassypnaya, also close to Orenburg, and from a similar stratigraphic horizon to the holotype of *G. prima*, is represented by the cranial and postcranial remains of at least seven individuals [6,16–25]. Butler *et al.* [6] recently formally synonymized *Garjainia prima* and '*Vjushkovia triplicostata*', referring all material previously described as *V. triplicostata* to *G. prima* (see also [9,15]). As a result, *G. prima* is now one of the most completely known archosauromorphs from the Early Triassic. Butler *et al.* [6] described the skull material referred to *G. prima*, but the postcranial material of the hypodigm of '*V. triplicostata*' has previously not been comprehensively or thoroughly described and, with exception of the tarsus, has been figured only as line drawings [16,23]. Here, we describe and figure in detail the extensive and well-preserved postcranial material of '*V. triplicostata*' for the first time and use it to reassess phylogenetic affinities, make the first assessments of erythrosuchid body mass, and to reconstruct locomotor musculature.

# 2. Systematic palaeontology

Diapsida Osborn, 1903 [26] *sensu* Laurin [27]
Sauria Gauthier, 1984 [28] *sensu* Gauthier, Kluge & Rowe [29]
Archosauromorpha Huene, 1946 [30] *sensu* Dilkes [31]
Archosauriformes Gauthier, Kluge & Rowe, 1988 [29]
Erythrosuchidae Watson, 1917 [32] *sensu* Ezcurra *et al*. [33]
*Garjainia* Ochev, 1958 [13]
**Garjainia prima** Ochev, 1958 [13]
**Holotype:** PIN 2394/5, a partial skeleton of a single individual [9,13].

**Referred specimens:** Almost the entire skeleton, with the exception of some caudal vertebrae and elements from the carpus and manus, from a bone bed of a single locality. The quarry was approximately 100 × 50 m in area. The material is representative of at least seven different individuals of different sizes [9,15,16] (see electronic supplementary material for a full list of postcranial material; see [6] for details of cranial material).

**Occurrence:** The holotype is from the Kzyl-Sai II 2 locality, in the Fedorovkian Gorizont of the Yarengian Supergorizont. The referred specimens, previously assigned to '*Vjushkovia triplicostata*' [16] are from the Rassypnaya locality, in the Gamian Gorizont of the Yarengian Supergorizont. Both sites are of late Olenekian (Early Triassic) age [13,15,20]. The Rassypnaya locality is 1.5 km northeast of Rassypnoe village, on the right bank of the Ural River in the Ilek District, while the Kzyl-Sai II 2

locality is approximately 150 km to the southeast, 0.5–1 km west of Andreevka village, Akbulak district. Both are in Orenburg Province, Russia.

**Revised diagnosis.** *Garjainia prima* can be differentiated from other archosauromorphs by the following unique combination of character states (autapomorphies indicated with *): premaxilla with a longitudinal groove on the lateral surface of the premaxillary body (also present in *G. madiba*); palatal process gently curved ventrally with its main axis sub-parallel to the alveolar margin of the bone (long axis of the process posteriorly intercepts that of the alveolar margin in *G. madiba*); five tooth positions (six in *G. madiba*); nasal with an anteroposteriorly long descending process that forms an extensive longitudinal suture with the maxilla*; antorbital fossa absent on the horizontal process of the maxilla (also probably present in *G. madiba*); antorbital fenestra trapezoidal, with its main axis orientated from anteroventrally to posterodorsally, in association with the horizontal process of the maxilla being much taller posteriorly*; prefrontal strongly flared laterally in dorsal view*; skull roof with a longitudinal fossa on its dorsal surface that harbours a longitudinal median prominence in its posterior half*; straight suture between postfrontal and postorbital (possibly present in some individuals of *Shansisuchus shansisuchus*); jugal and postorbital without a ball-like boss on their external surfaces (ball-like boss is present in *G. madiba*); basioccipital with a median tuberosity on its ventral surface*; interclavicle with a rhomboidal posterior ramus*; mammillary processes (*sensu* [34]) present on neural spines between fifth and twelfth presacral vertebrae*; mid-dorsal vertebrae with variably present thin accessory laminae that subdivide the bases of the prezygapophyseal centrodiapophyseal and centrodiapophyseal fossae*; second sacral rib bifurcated laterally* (rib developed as a single unit in other erythrosuchids); and anterior caudal ribs that bifurcate distally with a posteroventral projection extending from the distal end* (Modified from [9]).

# 3. Description

## 3.1. Material and methods

The vast majority of the referred material was excavated from a single bone bed in 1953–1954, although some material was collected subsequently at the same site in 1974 (see [6] for historical context). A minimum of seven individuals is preserved. There is a continuous series of presacral, sacral and anterior caudal vertebrae that were probably originally preserved in articulation (electronic supplementary material, figure S1), and at least some of the pectoral and pelvic elements can be articulated with one other (electronic supplementary material, figure S3); it is therefore likely that there was more association of material in the field, perhaps such that the material was mostly articulated. However, no detailed records of the original excavation, which is known to have used a bulldozer, or quarry maps exist, and thus associations are uncertain, although some inferences can be made (see below). See electronic supplementary material for a complete list of specimens, additional figures, and measurements of all elements.

## 3.2. Axial skeleton

### 3.2.1. Vertebrae

PIN 951/64 is a series of vertebrae, including the axis, 23 additional presacral vertebrae, two sacral vertebrae, and parts of four caudal vertebrae. The first 26 vertebrae (axis to sacrals) appear to be a continuous, originally articulated series (electronic supplementary material, figure S1); it is less clear if the four caudal vertebrae are those from the very base of the caudal series of the same individual or whether there are elements missing in this region. The count of 25 presacral vertebrae (including the missing atlas) is consistent with that present in the proterosuchid *Proterosuchus alexanderi* (NMQR 1484) and that inferred for *Erythrosuchus africanus* by Gower [4]. The centra are generally anteroposteriorly short relative to their heights, and proportions do not change substantially along the preserved length of the column, contrasting with the strongly anteroposteriorly compressed vertebrae around the cervico-dorsal transition of *E. africanus* and *Shansisuchus shansisuchus* [10,34]. All neurocentral sutures are completely fused, unlike the condition in *E. africanus* (AG Sennikov 2019, personal observation). A number of other isolated vertebrae are also preserved (see electronic supplementary material). The following description is based on the articulated series of vertebrae (PIN 951/64) except where specified. At the end of this section, isolated vertebral material that differs from the condition seen in PIN 951/64 is discussed.

Of the atlanto-axis complex, in PIN 951/64, only a mostly complete axis is preserved, but this complex is well represented in other specimens, including a complete and very well-preserved axis

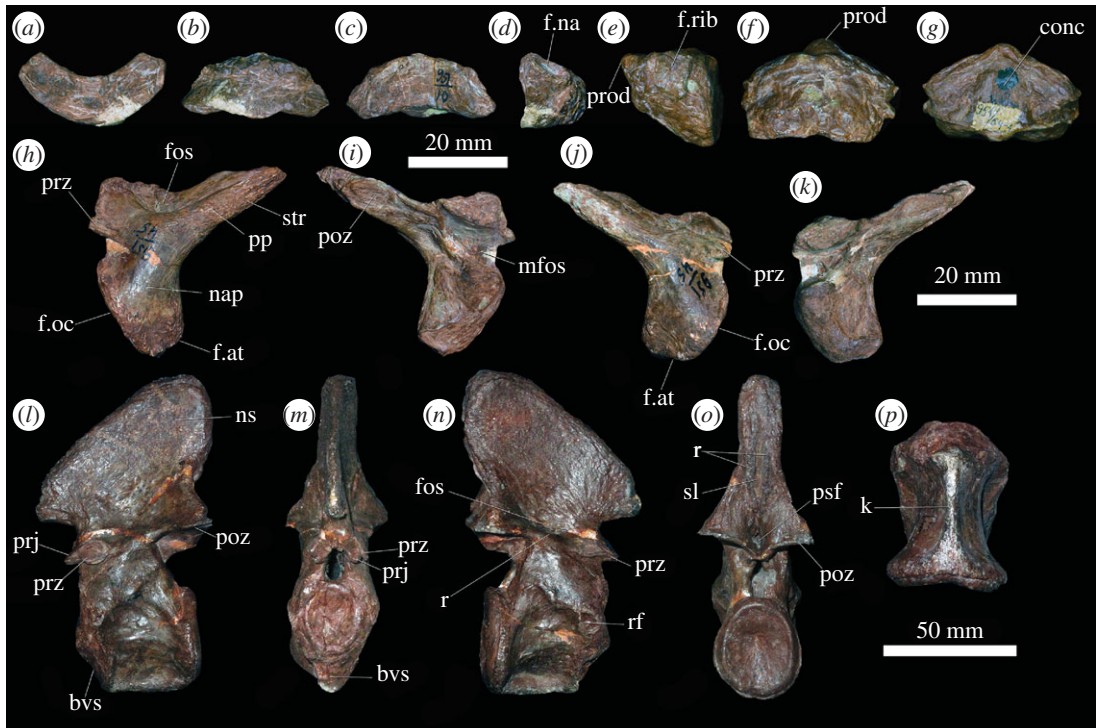

**Figure 1.** (*a–d*) PIN 951/10-3, atlantal intercentrum in (*a*) posterior, (*b*) ventral, (*c*) dorsal and (*d*) right lateral views. (*e–g*) PIN 951/64-1, odontoid process in (*e*) left lateral, (*f*) ventral and (*g*) dorsal views. (*h–k*) PIN 951/45-2, atlantal neural arches in (*h,j*) lateral and (*i,k*) medial views. (*h,i*) PIN 951/45-2-1, left neural arch; (*j,k*) PIN 951/45-2-2 right neural arch. (*l–p*) PIN 951/25-3, axis, in (*l*) left lateral, (*m*) anterior, (*n*) right lateral, (*o*) posterior and (*p*), ventral views. **bvs,** bevelled surface for intercentrum; **conc**, concavity on dorsal surface of odontoid; **f.at**, facet for atlantal intercentrum; **f.na**, facet for neural arch; **f.oc**, facet for occipital condyle; **f.rib**, rib facet; **fos**, fossa; **k**, keel; **mfos**, fossa on medial surface of atlantal neural arch; **nap**, neural arch pedicle; **ns**, neural spine; **poz**, postzygapophysis; **pp**; posterior process; **prj**, anterior projection on axis; **prod**, anterior projection on odontoid; **prz**, prezygapophysis; **psf**, post-spinal fossa; **r**, ridge; **rf**, rib facet; **sl**, slot; **str**, striations. Scale bars equal to 20 mm for (*a–k*), and 50 mm for (*l–p*).

(PIN 951/25-3), two atlantal intercentra (PIN 951/10-2, 3), two odontoid processes (PIN 951/64-1, PIN 951/45-4) and three pairs of atlantal neural arches (PIN 951/45-1, 2, 3; figure 1). There are no definite proatlas elements, and it is also unclear if a separate axial intercentrum was present or not. The atlantal intercentrum (figure 1*a–d*) is a wedge-shaped bone that in ventral view is approximately twice as wide transversely as long anteroposteriorly, as is the case in other early archosauromorphs (e.g. *Azendohsaurus madagaskarensis*: [35]; *Prolacerta broomi*: BP/1/2675; *Euparkeria capensis*: [36]). The ventral surface of the atlantal intercentrum is transversely convex and the dorsal surface is transversely concave, but slopes from posterodorsally to anteroventrally to form an articular surface for the occipital condyle of the skull. The posterior surface of the atlantal intercentrum is flattened, and the dorsal part of this surface articulated with the odontoid process (=atlantal centrum). Laterally, on each side of the element, there is a facet for articulation with the atlantal neural arch. This facet has an oval outline, the long axis of which angles anteroventrally to posterodorsally.

The atlantal neural arch comprises paired elements that approach one another dorsally on the midline, although it is unclear whether or not a midline contact occurred (figure 1*h–k*). The neural arch pedicle is anteroposteriorly broad relative to its transverse width, with a convex lateral surface and a concave medial surface. It is divided ventrally into two articular facets: a smaller one that articulated with the facet on the atlantal intercentrum, and a larger one that extends from anterodorsally to posteroventrally and which articulated with the occipital condyle. Both of these facets have gently transversely convex articular surfaces. The prezygapophyseal facet is a small, rectangular, flattened surface that faces dorsolaterally (best preserved in PIN 951/45-1). Medial to this, the neural arch bends medially towards the midline. The postzygapophyseal facet is a flattened oval surface that faces ventromedially and is positioned on the medial surface of an elongate, triangular posterior process (figure 1*h–k*). This condition resembles that in other early archosauriforms (e.g. *Proterosuchus fergusi*: SNSB-BSPG 1934 VIII 514; *Proterosuchus alexanderi*: NMQR 1484; *Euparkeria*

*capensis*: [36]), but it differs from the proportionally shorter and plate-like posterior process of *Prolacerta broomi* (BP/1/2675). The posterior process of the atlantal neural arch of *Garjainia prima* continues for a short distance posterior to the postzygapophysis, terminating in a blunt tip. The ventral part of the lateral surface of this posterior process is heavily striated, indicating probable muscle attachment. Similar striations occur on the posterior and ventral parts of the neural arch pedicle. The dorsal and medial part of the lateral surface of the neural arch is depressed, forming a fossa that is delimited laterally by a low ridge, as also occurs in *Euparkeria capensis* [36] and proterochampsids [37]. By contrast, the dorsolateral region of the atlantal neural arch is not depressed, but mostly flat in *Prolacerta broomi* (BP/1/2675) and proterosuchids (e.g. *Proterosuchus fergusi*: SNSB-BSPG 1934 VIII 514; *Proterosuchus alexanderi*: NMQR 1484). Medially, the atlantal neural arch of *Garjainia prima* possesses a small, deep, triangular fossa, immediately medial to the prezygapophysis.

The odontoid process is wedge shaped in lateral view (figure 1*e–g*). Its anteroventral surface is transversely convex and flat to slightly concave anteroposteriorly. It has a small, rounded projection on the midline dorsally, immediately ventral to which is an anteriorly opening foramen. The dorsal surface of the element has a median concavity, which is subrectangular in outline in dorsal view. This concavity is defined laterally by low raised convexities. The posterior articular surface of the odontoid, which articulated with the axial centrum, has an oval outline with a transversely extending long axis and a width greater than 160% of the dorsoventral height. This articular surface is gently concave. Laterally, there are quadrangular articular facets for an atlantal rib. When the odontoid is placed in articulation with the axial centrum, the facets for the first (atlantal) and second (axial) cervical ribs are positioned very close to one another.

The two preserved examples of the axis are similar in size to one another, and in both cases the neural arch and centrum are indistinguishably fused with one another (figure 1*l–p*). The centrum is not fused to the axial intercentrum or odontoid process, similar to the condition in *Prolacerta broomi* and other non-eucrocopodan archosauriforms [9]. The length of the centrum is 120% of its posterior dorsoventral height, resembling the condition in the holotype of *Garjainia prima* [9] and *Sarmatosuchus otschevi* (PIN 2865/68), but contrasting with the either proportionally shorter or longer centra of *Prolacerta broomi* and other non-eucrocopodan archosauriforms (see details of this comparison in [9]). The anterior articular surface is slightly taller dorsoventrally than wide transversely and is deeply concave (figure 1*l*). Ventral to the articular surface, the anterior end of the centrum is bevelled and slopes from anterodorsal to posteroventral. This bevelled surface is transversely convex and gently concave anteroposteriorly (figure 1*l–n*) and presumably articulated with the axial intercentrum. However, the latter is not preserved. The posterior articular surface of the axis is considerably taller dorsoventrally than the anterior articular surface. The posterior articular surface is also slightly taller than wide and deeply concave, confirming the apparent suboval outline of the posterior articular surface in the holotype of *G. prima*.

The axis centrum is spool-shaped in ventral view, with anteroposteriorly concave lateral surfaces (figure 1*p*). A low, anteroposteriorly extending break-in-slope at the mid-height of the centrum divides the lateral surface into upper and lower parts, both of which are anteroposteriorly and dorsoventrally concave, as is also the case in the holotype of the species [9] and *Prolacerta broomi* (BP/1/2675), but contrasting with the absence of such a break-in-slope of the concave lateral surface in *Proterosuchus fergusi* (SNSB-BSPG 1934 VIII 514), *Sarmatosuchus otschevi* (PIN 2865/68), *Erythrosuchus africanus* (BP/1/5207) and *Euparkeria capensis* (SAM-PK-5867). A large nutrient foramen is present on the anterior end of the lateral surface, immediately dorsal to this break-in-slope, as is the case in the holotype [9]. Several other minute foramina are present on the lateral surface, mostly close to the anterior end of the element and ventral to the break-in-slope. The ventral part of the centrum is strongly compressed to form a distinct median keel that extends along the entire length of the element (figure 1*p*), resembling the condition in *Prolacerta broomi* (BP/1/2675) and *Proterosuchus fergusi* (SNSB-BSPG 1934 VIII 514; SAM-PK-11208). By contrast, in the holotype of *Garjainia prima*, the ventral part of the centrum is also strongly compressed, but it does not form a sharp median keel [9]. A subcircular rib facet is positioned on the anterior margin of the lateral surface of the centrum, immediately lateral to the mid-height of the anterior articular surface (figure 1*n*), and there is no evidence of a second rib facet, similar to the condition in the holotype of *G. prima* [9], *Prolacerta broomi* (BP/1/2675), *Proterosuchus fergusi* (SNSB-BSPG 1934 VIII 514) and *E. africanus* (BP/1/5207).

The axis neural canal is taller dorsoventrally than it is wide transversely. The prezygapophyseal facet is a small, subcircular articular surface that faces dorsomedially and which lies directly above the rib articular facet (figure 1*l–n*). There are small, triangular, anterior projections immediately in front of the prezygapophyses (figure 1*l,m*). The postzygapophyses are large posterior projections that

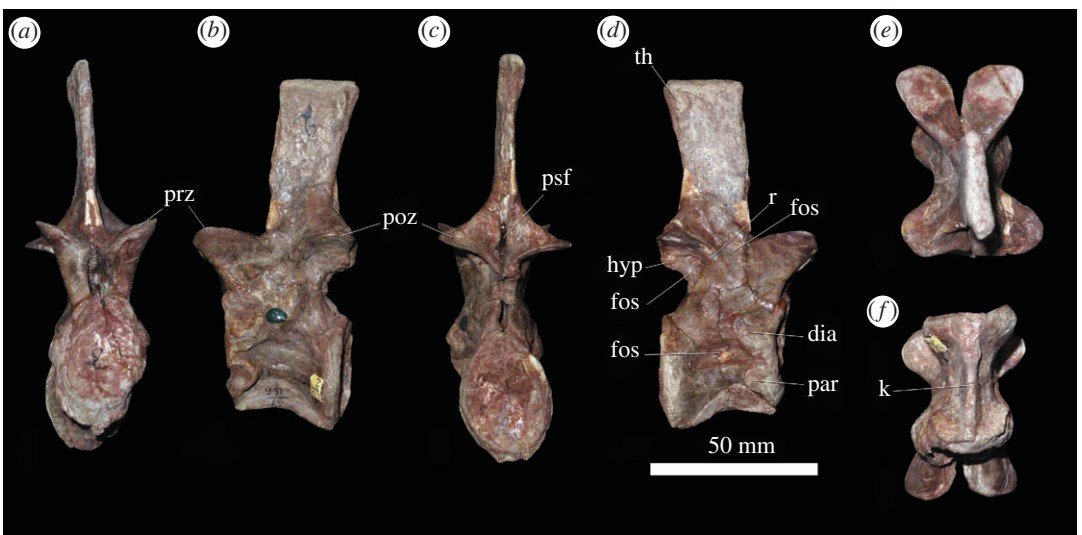

**Figure 2.** PIN 951/64-3, presacral 3 in (*a*) anterior, (*b*) left lateral, (*c*) posterior, (*d*) right lateral, (*e*) dorsal and (*f*) ventral views. **dia**, diapophysis; **fos**, fossa; **hyp**, hyposphene; **k**, keel; **par**, parapophysis; **poz**, postzygapophysis; **prz**, prezygapophysis; **psf**, post-spinal fossa; **r**, ridge; **th**, transverse thickening. Scale bar equal to 50 mm.

extend posteriorly slightly beyond the posterior margin of the centrum and face almost entirely ventrally (figure 1*l*–*o*). The neural spine is a large, triangular plate in lateral view that extends posterodorsally. It has an anterior projection that extends distinctly beyond the anterior margin of the centrum. The posterior margin of the spine terminates approximately level with the posterior margins of the postzygapophyses, resembling the condition in the holotype of *Garjainia prima* [9] and *Sarmatosuchus otschevi* (PIN 2865/68). By contrast, the neural spine extends considerably beyond the level of the postzygapophysis in *Proterosuchus fergusi* (SNSB-BSPG 1934 VIII 514) and *Shansisuchus shansisuchus* [38], fig. 20a,b). Most of the lateral surface of the spine is striated for probable muscle attachment. A shallow fossa is present on the base of the neural arch laterally (figure 1*n*), midway between the prezygapophysis and the postzygapophysis, and the ventral margin of this fossa is defined by a low ridge that extends between these two processes. This same morphology is also present in the holotype of *G. prima* [9] and *S. otschevi* (PIN 2865/68), but it is absent in *Erythrosuchus africanus* (BP/1/5207). In posterior view, there is a deep but transversely narrow post-spinal fossa between the postzygapophyses (figure 10). Dorsal to this, on the neural spine, there is a narrow midline slot, bordered laterally by sharp ridges. Most of the posterior surface of the spine is heavily striated for probable muscle attachment.

Presacral 3 (figure 2) has a centrum that is almost exactly as long as it is dorsoventrally high, resembling the only slightly longer than tall anterior postaxial cervical vertebrae of *Sarmatosuchus otschevi* (PIN 2865/68) and the holotype of *Garjainia prima* [34]. By contrast, the anterior postaxial cervical vertebrae of *Erythrosuchus africanus* [4] (SAM-PK-3028) and *Shansisuchus shansisuchus* ([38], fig. 20e) are considerably shorter than tall, and the converse occurs in *Prolacerta broomi* (BP/1/2675), *Teyujagua paradoxa* [39], *Proterosuchus fergusi* (SNSB-BSPG 1934 VIII 514, SAM-PK-11208) and *Euparkeria capensis* (SAM-PK-5867). Anterior and posterior articular surfaces of the referred specimens of *G. prima* both have oval outlines, are strongly concave, and their heights are approximately 150% of their transverse widths (figure 2*a*,*c*). They resemble those of the holotype, showing that narrowness of the facets have not been substantially affected by taphonomic compression. The centrum is spool-shaped in ventral view and is strongly transversely compressed at its ventral margin to form a sharp ventral midline keel (figure 2*f*), as occurs in other early archosauriforms, although the ventral keel of *E. africanus* (BP/1/4680) is transversely thicker than in *G. prima*. In lateral view, the ventral margin of the centrum is concave and the ventral margin of the anterior articular facet is placed dorsal to the level of that margin in the posterior facet, resembling the condition in other erythrosuchids and most archosauromorphs [9,40]. The anteroventral margin of the centrum is bevelled probably for articulation with a small intercentrum. The parapophysis and diapophysis are positioned close together on the anterior margin of the centrum; the parapophysis on the anteroventral corner, and the diapophysis just dorsal to this, approximately level with the mid-height of the anterior articular facet (figure 2*d*). A thickened ridge extends posteriorly from the

parapophysis, extending across the entire lateral surface of the centrum, and separating the ventral keel from a more dorsally positioned deep oval fossa (figure 2b,d), as occurs in the holotype of *G. prima*, *E. africanus* (SAM-PK-3028) and *G. madiba* (BP/1/5360).

The neural arch of presacral 3 lacks sharply defined laminae and is dorsoventrally taller than wide, resembling the condition of proterosuchids and *Erythrosuchus africanus* [4,9]. The prezygapophysis is a large process that extends anterodorsally and overhangs the anterior articular surface of the centrum (figure 2a,b). Its broad, flattened articular surface faces dorsomedially at a low angle (ca 20°) to the horizontal. The postzygapophysis has a similarly broad, flattened articular surface that faces ventrolaterally, again at a low angle to the horizontal. A short, laterally facing surface below the medial margin of the postzygapophysis may be a hyposphene that would have articulated with a notch between the prezygapophyses of the succeeding vertebra (figure 2a–e). Ezcurra *et al.* [9] described the absence of a hyposphene or hypantrum in the anterior postaxial cervical vertebrae of the holotype of *Garjainia prima*, but the apparent absence of these features may be as a result of poor preservation. A low, thick ridge extends onto the lateral surface of the neural arch from the anterior margin of the postzygapophysis, and there is a shallow fossa on the lateral surface of the arch, anterior to this ridge, as occurs in the holotype of *G. prima* [9], *Proterosuchus fergusi* (SNSB-BSPG 1934 VIII 514) and *Erythrosuchus africanus* (SAM-PK-3028). In more posterior vertebrae, this ridge becomes the postzygodiapophyseal lamina. There is another shallow fossa on the lateral surface of the arch, immediately ventral to the postzygapophysis and posterior to the ridge described above. Posteriorly, a deep post-spinal fossa between the postzygapophyses extends onto the base of the posterior surface of the neural spine (figure 2c), which is a condition widespread among early archosauromorphs (e.g. holotype of *G. prima*: [9]). The neural spine is elongate and rectangular in lateral view and is directed dorsally, with a squared off distal end, resembling the condition present in various early archosauromorphs (e.g. *Sarmatosuchus otschevi*: PIN 2865/68; *Teyujagua paradoxa*: [40]; some specimens of *Proterosuchus fergusi*: SNSB-BSPG 1934 VIII 514, SAM-PK-11208; *Euparkeria capensis*: SAM-PK-5867). The neural spine of *G. prima* is similar in height to the height of the neural spine of the axis and is slightly transversely thickened at its apex. There is a low rugose thickening on the lateral surface of the spine, adjacent to its posterodorsal corner, but it lacks the broader and flat distal surface present in *Euparkeria capensis* (SAM-PK-5867).

The morphology of the succeeding three vertebrae is very similar to that of presacral 3 (electronic supplementary material, figure S1). There is a strongly developed keel on the centrum of presacral 4, but it becomes gradually less sharply defined in presacrals 5 and 6. Centrum proportions gradually become taller than anteroposteriorly long. The parapophyses and diapophyses become gradually more broadly separated from one another; the parapophyses remain on the anteroventral corner of the centrum, but the diapophyses are located increasingly dorsally. Presacral 4 has a very thin, low ridge that extends from the base of the prezygapophysis towards the base of the postzygapophysis, and this is also present in more posterior vertebrae, resembling the condition in the holotype [9]. Presacral 4 lacks any development of a transverse thickening or rugosity on the lateral surface of the distal neural spine, but beginning with presacral 5 there is a discrete, well-developed and strongly rugose thickening that has a quadrangular outline in lateral view and is positioned adjacent to the spine apex and immediately anterior to the midpoint of the lateral surface of the spine (electronic supplementary material, figure S1). These lateral thickenings represent mammillary processes and give the spines of presacrals 5 and 6 a cross-shaped profile in dorsal view, resembling the condition in *Shringasaurus indicus*, '*Chasmatosaurus*' *yuani* and *Proterosuchus alexanderi*, whereas mammillary processes are also present in the cervical series but starting from the sixth presacral or posteriorly in *Prolacerta broomi* and *Proterosuchus fergusi* [34,41]. By contrast, mammillary processes are absent in the neural spines of other erythrosuchids (e.g. *Erythrosuchus africanus*, *Guchengosuchus shiguaiensis*, *Shansisuchus shansisuchus*; [34]) and their apparent absence in the fifth cervical vertebra of the holotype of *Garjainia prima* is probably as a result of lack of preservation (PIN 2394/5-13). Coincident with this, the entire apex of the neural spines of the presacrals 5 and 6 of this referred specimen of *G. prima* (PIN 951/64) become more strongly transversely expanded. Ventral to the rugose thickening on the lateral surface of the spine is a series of low, anteroposteriorly extending striations that form a line of probable muscle scars extending ventrally towards the postzygapophysis (electronic supplementary material, figure S1).

Presacral 7 is the anteriormost vertebra in which the diapophysis is set at the end of a distinct transverse process (electronic supplementary material, figure S1). This process is short and extends ventrolaterally from the base of the neural arch. Immediately anteroventral to the articular facet of the diapophysis is a small additional articular surface, suggesting that this is the first presacral to

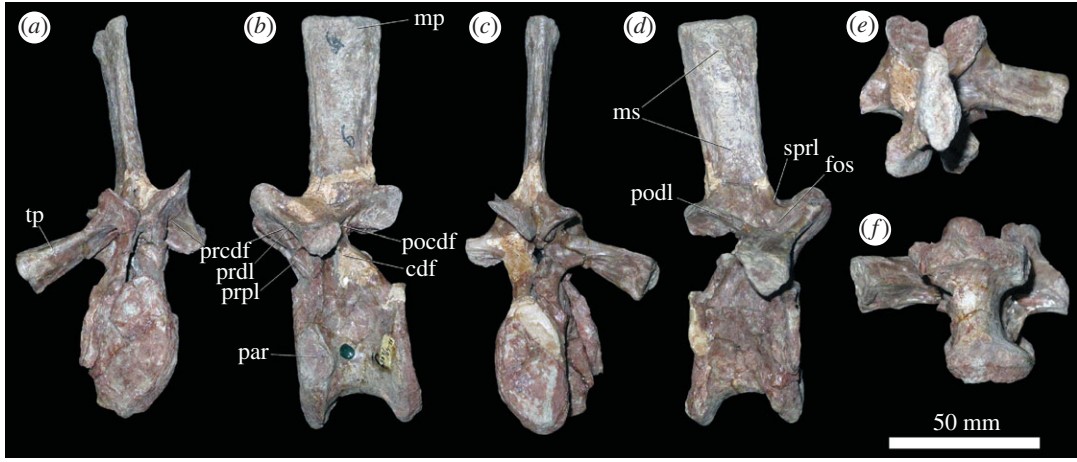

**Figure 3.** PIN 951/64-9, presacral 9 in (*a*) anterior, (*b*) left lateral, (*c*) posterior, (*d*) right lateral, (*e*) dorsal and (*f*) ventral views. **cdf,** centrodiapophyseal fossa; **fos,** fossa; **mp,** mammillary process; **ms,** muscle scars; **par,** parapophysis; **pocdf,** postzygapophyseal centrodiapophyseal fossa; **podl,** postzygodiapophyseal lamina; **prcdf,** prezygapophyseal centrodiapophyseal fossa; **prdl,** prezygodiapophyseal lamina; **prpl,** prezygoparapophyseal lamina; **sprl,** spinoprezygapophyseal lamina; **tp,** transverse process. Scale bar equal to 50 mm.

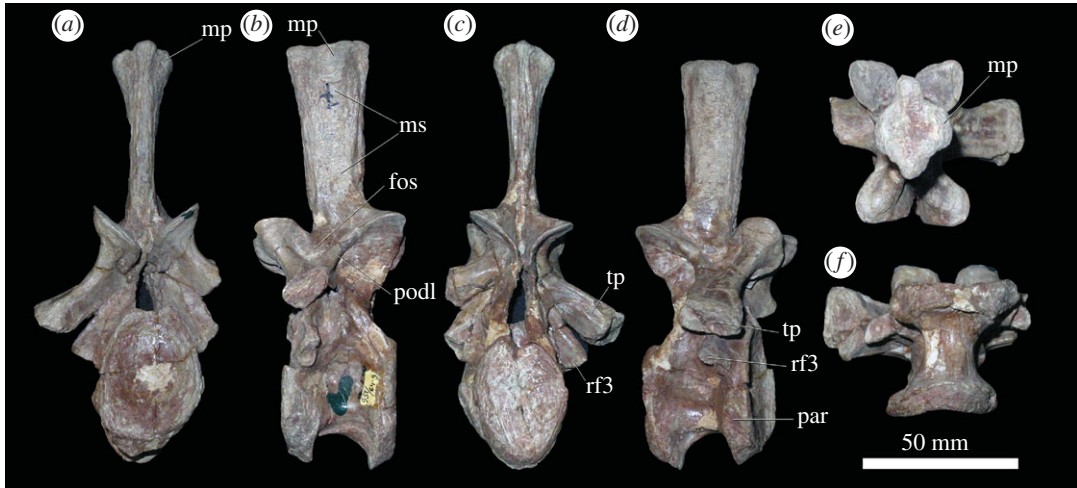

**Figure 4.** PIN 951/64-11, presacral 11 in (*a*) anterior, (*b*) left lateral, (*c*) posterior, (*d*) right lateral, (*e*) dorsal and (*f*) ventral views. **fos,** fossa; **mp,** mammillary process; **ms,** muscle scars; **par,** parapophysis; **podl,** postzygodiapophysial lamina; **rf3,** rib facet for third rib head; **tp,** transverse process. Scale bar equal to 50 mm.

articulate with a triple-headed rib. The presence of an accessory rib articulation around the cervico-dorsal transition is present in *Prolacerta broomi* and various non-eucrocopodan archosauriforms (e.g. *Proterosuchus fergusi*, *Proterosuchus alexanderi*, *Chasmatosuchus rossicus*, *Vonhuenia friedrichi*, *Sarmatosuchus otschevi*, *Erythrosuchus africanus*, *Guchengosuchus shiguaiensis*, *Garjainia madiba*, *Cuyosuchus huenei*; [34]), aphanosaurian avemetatarsalians [42] and some loricatan pseudosuchians (e.g. *Batrachotomus kupferzellensis*: [43]). The transverse process of presacral 7 of *Garjainia prima* overhangs a very deep fossa on the lateral surface of the neural arch, the posterior margin of which is formed by a low posterior centrodiapophyseal lamina, and the anterior margin of which is formed by an anterior centrodiapophyseal lamina on which lies the third, accessory articular facet. The parapophysis is still positioned on the anteroventral corner of the centrum. Thick and low prezygodiapophyseal and postzygodiapophyseal laminae are present. The thin lamina that extended between the pre- and postzygapophysis in presacrals 3–6 is absent in presacral 7 and more posterior vertebrae (electronic supplementary material, figure S1). Other aspects of the morphology of this vertebra are similar to those of the proceeding elements.

Evidence for articular surfaces for triple-headed ribs is limited to presacrals 7–12 (figures 3 and 4; electronic supplementary material, figure S1), which are interpreted as cervical vertebrae 7–9 and dorsal

vertebrae 1–3. These vertebrae are broadly similar in morphology. Their centra have articular facets that are deeply concave and taller than wide, and the centra are generally taller than anteroposteriorly long (ca 115%; figure 3a,c), but not as anteroposteriorly compressed as the 'pectoral' centra of Erythrosuchus africanus and Shansisuchus shansisuchus [4,34,37]. The centra of presacrals 7–12 of Garjainia prima are spool-shaped in ventral view, with strongly concave lateral surfaces, and with ventral surfaces that are either rounded transversely or have a weak median ridge (figure 3f and electronic supplementary material, figure S1). The lateral surfaces of the centra are very deeply excavated fossae, as occurs in most early archosauriforms [34]. The anteroventral corners of the centra are bevelled for articulation with intercentra (figure 3b). The parapophysis is positioned on the anteroventral corner of the centrum in presacrals 7–10, but begins to shift more dorsally in vertebrae posterior to this. The transverse process becomes increasingly elongate and more horizontally directed, and in presacrals 8–10 the accessory facet for the third rib head is clearly set on the end of a distinct ventrolaterally extending apophysis (figure 3a–d). In presacral 11 (figure 4), the area in which the third accessory facet would have been is broken, but this is clearly present on the left side of presacral 12.

As the transverse process becomes longer and more horizontally directed, a complex of well-developed neural arch laminae and fossae develop, as also occurs in other erythrosuchids [4,6,9,10,44]. In presacrals 11–12, these include prezygodiapophyseal, prezygoparapophyseal and paradiapophyseal laminae, framing a deep prezygapophyseal centrodiapophyseal fossa; a posterior centrodiapophyseal lamina, which together with the paradiapophyseal lamina frames the centrodiapophyseal fossa; and a postzygodiapophyseal lamina, which together with the posterior centrodiapophyseal lamina frames the postzygapophyseal centrodiapophyseal fossa (figure 4b,d; electronic supplementary material, figure S1). The bases of these fossae are generally poorly preserved, but appear to be blind, without obvious foramina, contrasting with their presence in the dorsal vertebrae of Erythrosuchus africanus [44,45]. Dorsal to the transverse process, a well-developed fossa is present on the base of the lateral surface of the neural spine, which is bounded by the postzygodiapophyseal lamina and a spinoprezygapophyseal lamina (figure 4d), resembling the condition in other non-eucrocopodan archosauriforms [34]. Neural spines of these vertebrae are similarly tall to those of preceding vertebrae. Presacrals 7–10 (figure 3) possess a strong, rugose mammillary process on the lateral surfaces of the distal ends of the neural spines, as in preceding vertebrae, but these are only very weakly developed in presacrals 11 and 12 (figure 4). The mammillary processes also extend up to presacral 12 in Prolacerta broomi (BP/1/2675) and some specimens of Proterosuchus fergus [34], but extend more posteriorly in the dorsal series in Shringasaurus indicus [41], Proterosuchus alexanderi (NMQR 1484) and 'Chasmatosaurus' yuani (IVPP V4067). The absence of mammillary processes on the vertebrae of the holotype of Garjainia prima described as 'cervico-dorsals' by Ezcurra et al. [9] indicates that they represent vertebrae from posterior to presacral 12 in the vertebral column. Dorsoventrally extending probable muscle scars are visible on the lateral surfaces of the neural spines of all these vertebrae. These vertebrae lack the posterodorsal projection of the distal end of the neural spine present in the cervico-dorsal vertebrae of the holotype of Garjainia prima [9].

Presacrals 13–19 (figure 5; electronic supplementary material, figure S1) are similar in morphology to one another, representing middle dorsal vertebrae (dorsal vertebrae 4–10 if there are nine cervical vertebrae). The centra are subequal in anteroposterior length and dorsoventral height. The articular facets are taller dorsoventrally than transversely wide and deeply concave, and the anteroventral corner is bevelled for probable articulation with an intercentrum (figure 5a–d). Such articulations between middle dorsal centra and intercentra are preserved in a referred specimen of Shansisuchus shansisuchus [46]. The ventral margin of the centrum is strongly arched dorsally in lateral view, and in ventral view the centra are spool-shaped and strongly constricted at mid-length (figure 5f). There is a sharp median keel along the ventral margin in most of the vertebrae, although the ventral margin is more rounded transversely in some of them, resembling the condition in the middle dorsal vertebrae of Erythrosuchus africanus (NHMUK PV R3592). The fossae on the lateral surfaces of the centra are less well developed than in the preceding presacrals. The parapophysis is positioned on the anterodorsal corner of the centrum in presacral 13, considerably more dorsally than in presacral 12, and is on the base of the neural arch further posteriorly along with the vertebral series. Most of the transverse processes are broken, but the diapophyseal articular facet is preserved at the end of a long, horizontal transverse process when present. A third articular rib facet appears to be absent (figure 5b,d). The same complex of neural arch fossae and laminae are present as in the preceding dorsal vertebrae, but in some cases (e.g. presacral 13), thin accessory laminae subdivide the bases of the prezygapophyseal centrodiapophyseal and centrodiapophyseal fossae, which is a condition absent in other early archosauriforms of which we are aware. Hyposphenes do not appear to be well developed. Small

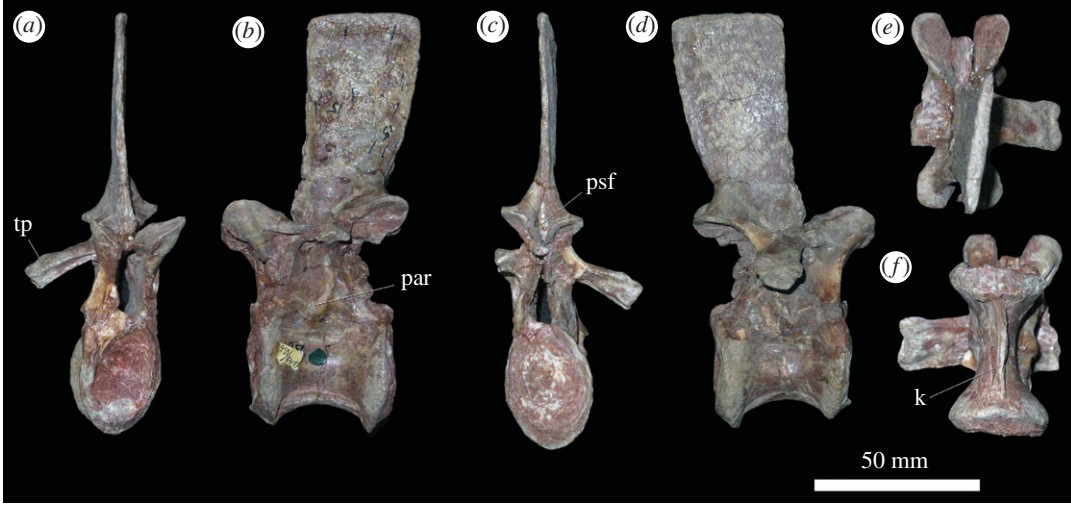

**Figure 5.** PIN 951/64-17, presacral 17, in (*a*) anterior, (*b*) left lateral, (*c*) posterior, (*d*) right lateral, (*e*) dorsal and (*f*) ventral views. **k**, keel; **par**, parapophysis; **psf**, post-spinal fossa; **tp**, transverse process. Scale bar equal to 50 mm.

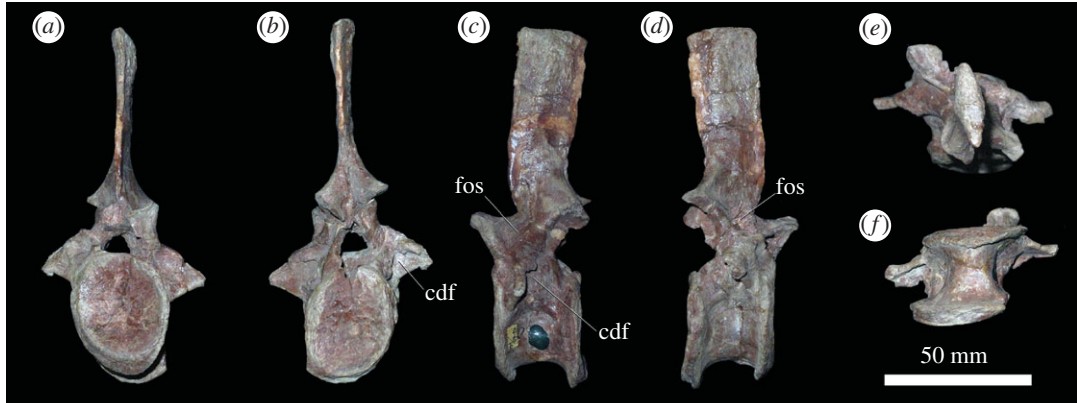

**Figure 6.** PIN 951/64-24, presacral 24 in (*a*) anterior, (*b*) posterior, (*c*) left lateral, (*d*) right lateral, (*e*) dorsal and (*f*) ventral views. **cdf**, centrodiapophyseal fossa; **fos**, fossa. Scale bar equal to 50 mm.

prespinal fossae are variably developed. A small post-spinal fossa is always present between the postzygapophyses, but does not extend onto the posterior surface of the neural spine. The neural spines lack transverse expansions distally (although their lateral surfaces retain the probable muscle scars present in preceding parts of the column), and they become increasingly anteroposteriorly expanded more posteriorly along with the series.

Presacrals 20–25 (figure 6; electronic supplementary material, figure S1) represent the posterior dorsal vertebrae (here interpreted as dorsal vertebrae 11–16). They differ from the preceding vertebrae in that they lack ventral keels on the centra (figure 6*f*), and in that the laminae and fossae on the neural arch are reduced: the prezygapophyseal centrodiapophyseal and postzygapophyseal centrodiapophyseal fossae are not well developed, and the centrodiapophyseal fossa is reduced in size. A fossa is still present on the base of the neural spine in lateral view (figure 6*a–d*). In most other aspects of their morphology, these vertebrae resemble those of the mid-dorsal vertebrae, although the centra of presacrals 23–25 are anteroposteriorly shorter than dorsoventrally tall. PIN 951/35-24 is another example of an ultimate dorsal vertebra, and probably articulates with PIN 951/37-3, an isolated sacral vertebra. It is better preserved than the posteriormost dorsal of PIN 951/64-24 (figure 6). The rib is fused to the vertebral apophysis (as also appears to be the case in PIN 951/64), and is very short, projecting laterally and ventrally, and curving slightly anteriorly towards its tip. The presence of the last dorsal rib fused to its respective vertebra is a rare condition among archosauromorphs and occurs in a few phylogenetically disparate species, such as *Proterosuchus alexanderi* (NMQR 1484) and *Azendohsaurus madagaskarensis* [35]. The lateral margins of the posterior articular facet are bevelled for articulation with the expanded anterior articular facet of sacral 1.

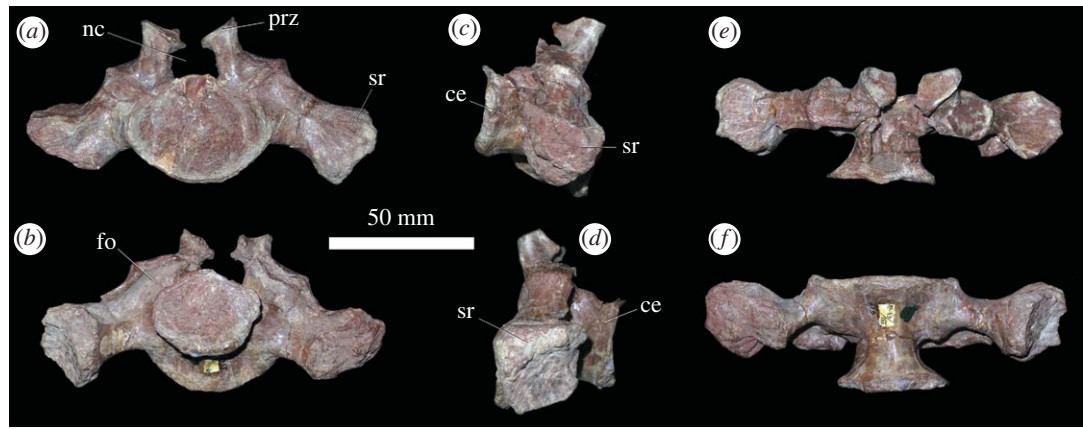

**Figure 7.** PIN 951/64-26, sacral vertebra 1 in (*a*) anterior, (*b*) posterior, (*c*) right lateral, (*d*) left lateral, (*e*) dorsal and (*f*) ventral views. **ce**, centrum, **fo**, fossa; **nc**, neural canal; **prz**, prezygapophysis; **sr**, sacral rib. Scale bar equal to 50 mm.

The sacrum is composed of two sacral vertebrae (figure 7 and electronic supplementary material, figure S2). The centra are spool-shaped in ventral view, with gently rounded ventral margins lacking keels or median ridges (figure 7*f*), resembling the condition in other erythrosuchids (e.g. *Erythrosuchus africanus*: NHMUK PV R3592; *Garjainia madiba*: [8]). The anterior and posterior articular surfaces of the sacral vertebrae are wider than tall, and strongly concave, but these articular surfaces are subcircular in *E. africanus* [4] and the first sacral vertebrae of *Shansisuchus shansisuchus* [38] and *G. madiba* [8]. However, the anterior facet is subcircular and the posterior is wider than tall in the second sacral of *G. madiba* [8]. In the first sacral of *G. prima*, the anterior articular surface is substantially wider and taller than the posterior articular surface, whereas in sacral 2, the dimensions of the anterior and posterior articular surfaces are similar to one another. This asymmetry between the anterior and posterior facets of the first sacral centrum is also present in *S. shansisuchus* [38], but it is not present in *G. madiba*, where both surfaces are subequal (BP/1/5525). The neural canals of the sacrals are expanded transversely and dorsoventrally relative to those of the posterior dorsal vertebrae (figure 7*a,b*). Neural arches and spines are largely missing.

Sacral rib 1 is massive and downturned, projecting ventrolaterally, as in other erythrosuchids (e.g. *G. madiba*: [8]; *E. africanus*: [4]; *S. shansisuchus*: [38]). There is a fossa on the posterior surface of the rib. The distal end of the rib is expanded anteroposteriorly, and posteriorly contacted sacral rib 2. The distal end of the rib is composed of a single body, as in the vast majority of archosauromorphs with the exception of *Proterosuchus alexanderi*, which has a bifurcated distal end of sacral rib 1 (NMQR 1484). Sacral rib 2 is more laterally directed than sacral rib 1 and is dorsoventrally compressed. It is strongly expanded at its distal end, primarily in the anterior direction, towards its contact with sacral rib 1. PIN 951/37-1, 2 are completely preserved sacral vertebrae 1 and 2 adhered in articulation with right and left ilia (electronic supplementary material, figure S2). The morphology is consistent with PIN 951/64 (figure 7), but in PIN 951/37 it can be determined that the distal end of the second sacral rib is bifurcated (electronic supplementary material figure S2; the area is damaged in PIN 951/64), with the presence of a posterolateral process similar to that of *Prolacerta broomi* [47], *Proterosuchus alexanderi* (NMQR 1484) and *Cuyosuchus huenei* [34]. By contrast, the distal end of sacral rib 2 is not bifurcated in other erythrosuchids, '*Chasmatosaurus*' *yuani*, and non-avemetatarsalian eucrocopodans [34,42,48]. The neural spines project slightly posterodorsally and are heavily striated for probable muscle attachment and are slightly expanded at their apices. The bases of the neural spines have slit-like fossae on their lateral surfaces.

Four anterior caudal vertebrae are present in PIN 951/64, which may represent the first four caudal vertebrae, although this is not certain. They are generally poorly preserved, but show that the anterior caudal vertebrae were similar to the dorsal vertebrae in possessing spool-shaped centra, with anterior and posterior articular facets that are taller than wide, strongly concave and bevelled ventrally at both anterior and posterior ends for articulation with the chevrons. The overall morphology of these caudals resembles that of the anterior caudal vertebrae of *Garjainia madiba* [8] and *Erythrosuchus africanus* [4]. The posterior articular surface is slightly offset ventrally relative to the anterior surface, as is the case in the most anterior caudal vertebrae of other archosauromorphs. Caudal ribs are fused to the vertebrae although sutures are still visible, and project from the neurocentral boundary laterally

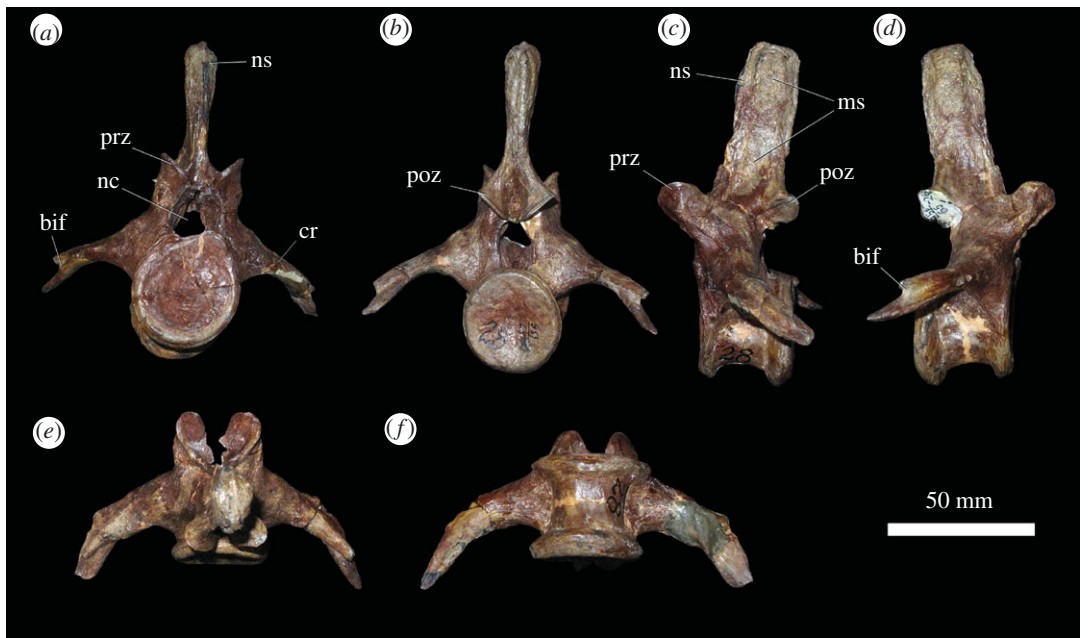

**Figure 8.** PIN 951/40-5, anterior caudal vertebra in (*a*) anterior, (*b*) posterior, (*c*) left lateral, (*d*) right lateral, (*e*) dorsal and (*f*) ventral views. **bif**, bifurcation at distal end of caudal rib; **cr**, caudal rib; **ms**, muscle scars; **nc**, neural canal; **ns**, neural spine; **poz**, postzygapophysis; **prz**, prezygapophysis. Scale bar equal to 50 mm.

and slightly ventrally. In PIN 951/40-5, there is evidence that the rib bifurcated distally, with a posteroventral projection extending from the distal end, which is a character state that we are unaware of in any other early archosauriform. Prezygapophyses are large, and their facets face dorsomedially at a high angle to the horizontal and project anteriorly beyond the anterior articular surface of the centrum. Postzygapophyseal facets face ventrolaterally at a similar angle to the horizontal and do not project posteriorly beyond the posterior articular surface. The neural spine is a narrow rectangular blade in lateral view. It is not transversely expanded distally, but does have the same row of probable muscle scars on its lateral surface as in the presacral vertebrae.

Several isolated caudal vertebrae preserve details not present in PIN 951/64. PIN 951/40-5 (figure 8) is a very well-preserved anterior caudal vertebra. Its caudal ribs curve posteriorly along their length and show a clear distal bifurcation. PIN 951/40-7 and PIN 951/40-8 represent slightly more distal caudal vertebrae. PIN 951/40-4 is a middle caudal vertebra with an elongate, low centrum, no neural spine as preserved (although it is possible that a short, posteriorly placed spine might have broken away), and thin, short transverse processes and caudal rib (figure 9).

Some of the isolated vertebrae are from slightly larger individuals than PIN 951/64 and suggest either intraspecific and/or ontogenetic morphological variation. Vertebrae numbered PIN 951/35 from the cervico-dorsal transition possess three rib articular facets, but differ from vertebrae of PIN 951/64 in that the facets are more strongly projected laterally, with a subcircular outline, a more strongly spool-shaped centrum in ventral view, a broad, flattened ventral margin rather than a ventral keel and proportionately shorter centra. A similar pattern is seen in some of the mid-dorsal vertebrae (e.g. PIN 951/35-5).

### 3.2.2. Ribs

Numerous complete and partial ribs are present, representing all parts of the axial column (figure 10). Cervical ribs (figure 10*a*,*b*) typically extended for the length of about two vertebrae. They are double-headed with a distinct, pointed anterior process that extends anteriorly beyond both the capitulum and tuberculum, as occurs in other archosauromorphs [49]. The rib shaft is relatively straight and is grooved medially and convex laterally. Triple-headed ribs come from the cervico-dorsal transition or 'pectoral' region (figure 10*c*,*d*), as discussed above. They have elongate, curved shafts that are compressed transversely, with a groove along their posterior surfaces. The tuberculum has the largest articular surface, followed by the capitulum. The third articular facet is the smallest and is placed medially and close to the tuberculum, as occurs in *Erythrosuchus africanus* [4]. A low flange is typically present on the anterior surface of the bone, immediately distal to the point at which the capitulum

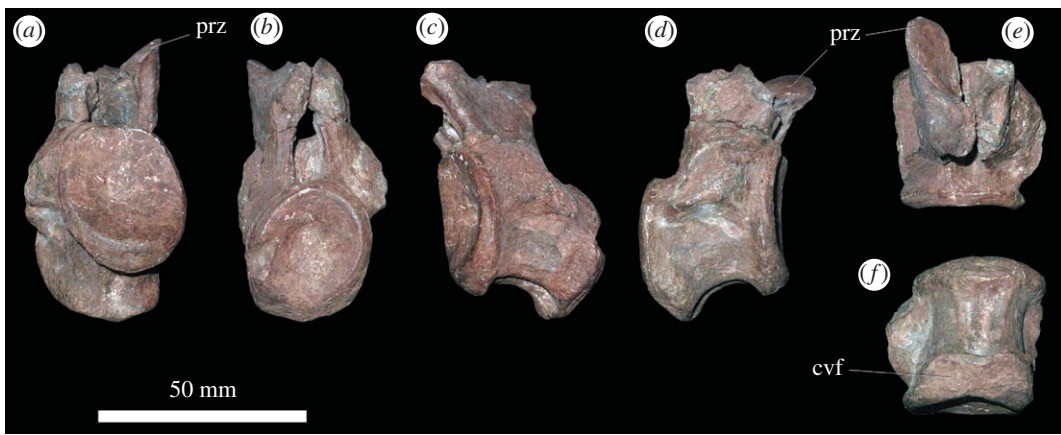

**Figure 9.** PIN 951/40-4 middle caudal vertebra in (*a*) anterior, (*b*) posterior, (*c*) left lateral, (*d*) right lateral, (*e*) dorsal and (*f*) ventral views. **cvf**, chevron facet; **prz**, prezygapophysis. Scale bar equal to 50 mm.

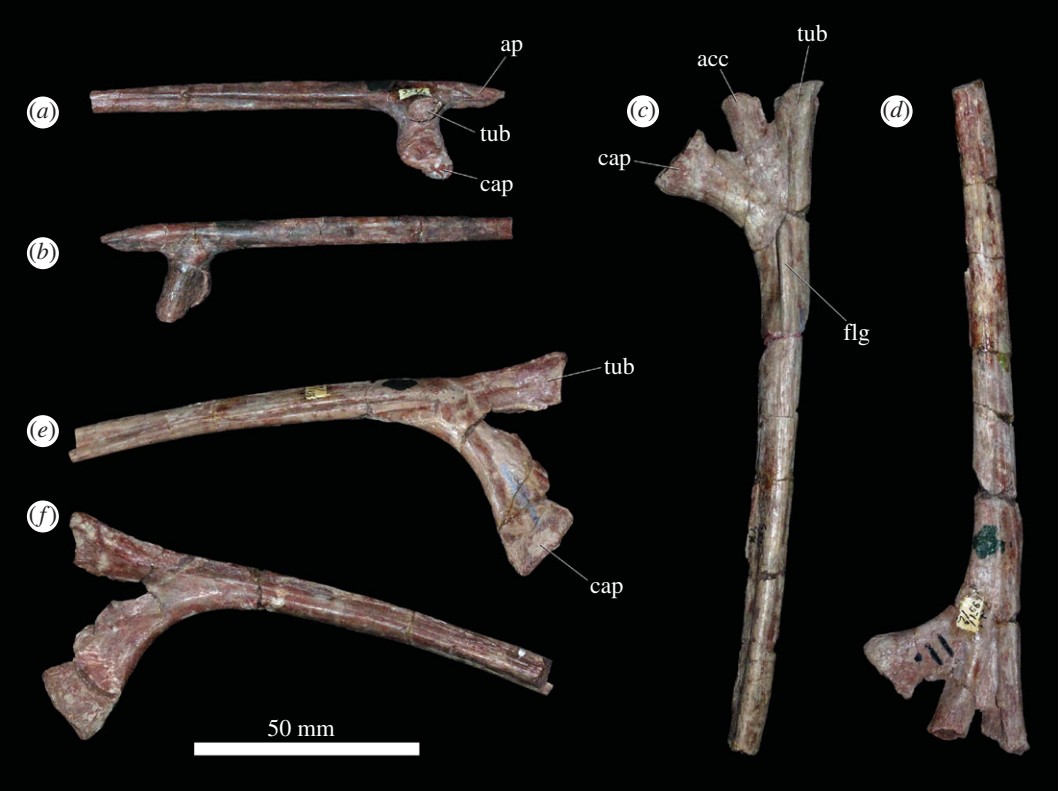

**Figure 10.** (*a,b*) PIN 951/24-6, cervical rib in (*a*) medial and (*b*) lateral views. (*c,d*) PIN 951/29-7, triple-headed 'pectoral' rib in (*c*) anterior and (*b*) posterior views. (*e,f*), PIN/28-5, double-headed middle dorsal rib in (*e*) anterior and (*f*) posterior views. **acc**, accessory articular facet; **ap**, anterior process; **cap**, capitulum; **flg**, flange; **tub**, tuberculum. Scale bar equal to 50 mm.

and tuberculum converge, and probably represents a remnant of the anterior process seen in the cervical ribs. This flange is reduced or absent in some of the triple-headed ribs, and it is likely that these ribs are more posterior in the cervico-dorsal transition. More posterior dorsal ribs (figure 10*e,f*) are double-headed, with the capitulum being larger than the tuberculum. They have curved along their lengths, lack flanges on the anterior surfaces and are grooved on their posterior surfaces.

### 3.2.3. Other axial elements

Thin rod-like bones probably represent fragments of the gastralia (figure 11*a–c*). Several small, thin, saddle-shaped bones represent intercentra (figure 11*f–h*). Based on articular surfaces, intercentra appear to have

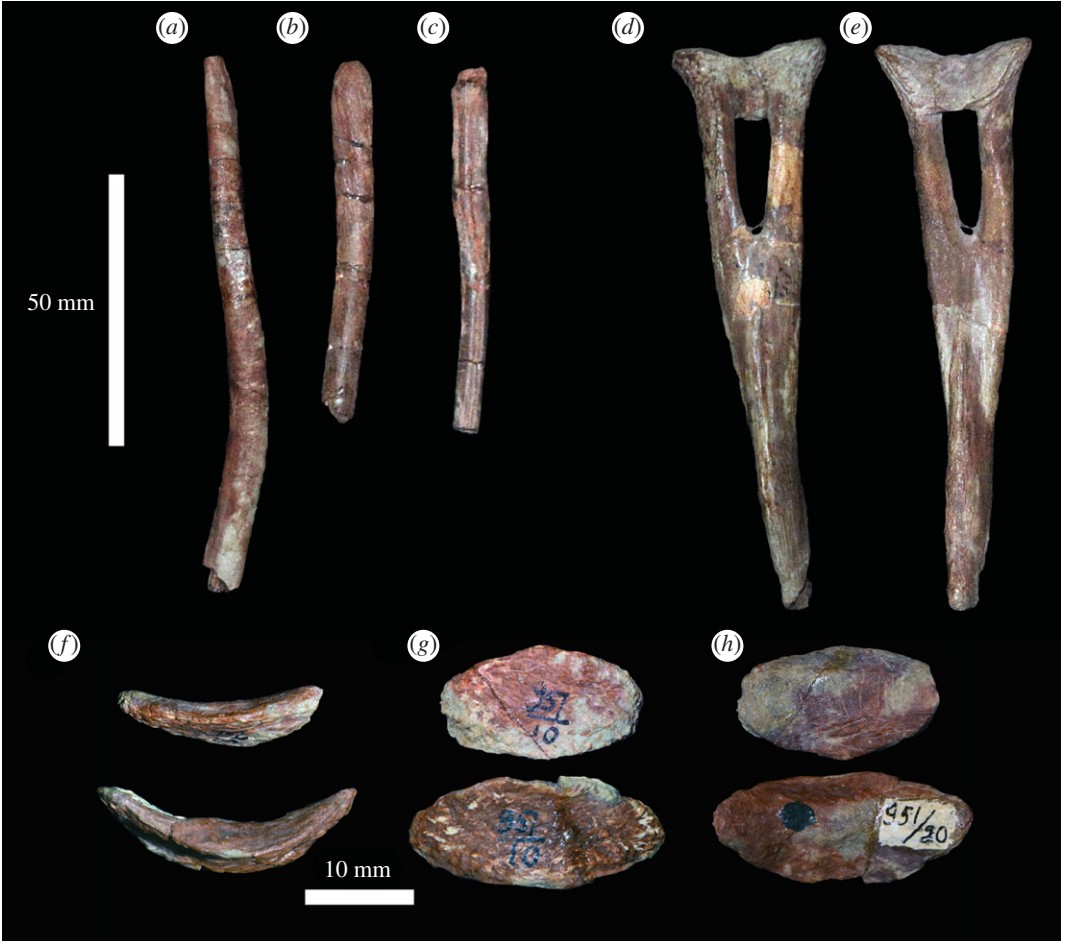

**Figure 11.** (*a*–*c*) PIN 951/133-1, 2, 3, examples of rod-like elements likely to be gastralia. (*d,e*) PIN 951/44, a chevron in (*d*) anterior and (*e*) posterior views. (*f*–*h*), PIN 951/10-15 (upper row) and PIN 951/10-4 (lower row), intercentra in (*f*) anterior/posterior, (*g*) ventral and (*h*) dorsal views. Scale bar equal to 50 mm for (*a*–*e*) and 10 mm for (*f*–*h*).

been present throughout the cervical and dorsal column, as occurs in *Erythrosuchus africanus* [4] and *Shansisuchus shansisuchus* [46]. The intercentra of *Garjainia prima* differ from those of *Shansisuchus shansisuchus* by their more slender, semi-lunar shape [46]. Three V-shaped chevrons are present (figure 11*d,e*), but all are incomplete distally. In lateral view, they are curved posteriorly along their length with no anteroposterior expansion at their distal ends. There is no evidence of any osteoderms in the material, as occurs in other erythrosuchids with the possible exception of *Erythrosuchus africanus* [4,50].

## 3.3. Appendicular skeleton

### 3.3.1. Pectoral girdle

At least 11 complete or partial pectoral girdles are present in the collection, and several can be paired with each other based on size and preservation (see electronic supplementary material) (figure 12). They range in size and in the degree of fusion between the coracoid and scapula; the fusion of the two elements appears variable and does not always correlate with size. The scapulocoracoid is described with the long axis of the scapula orientated dorsoventrally and with the broadest surface being lateral and medial.

*Coracoid.* The sutural contact between the coracoid and scapula is straight and orientated horizontally. Straight anterior, ventral and posterior margins of the coracoid merge with each other along gentle curves rather than sharp corners. The coracoid contribution to the glenoid occupies the posterodorsal corner of the element and is a gently concave indentation to the general D-shaped outline of the element. The anterior portion of the bone is transversely compressed and plate-like; posteroventrally the element is slightly transversely thicker, and posterodorsally the element is significantly transversely expanded for

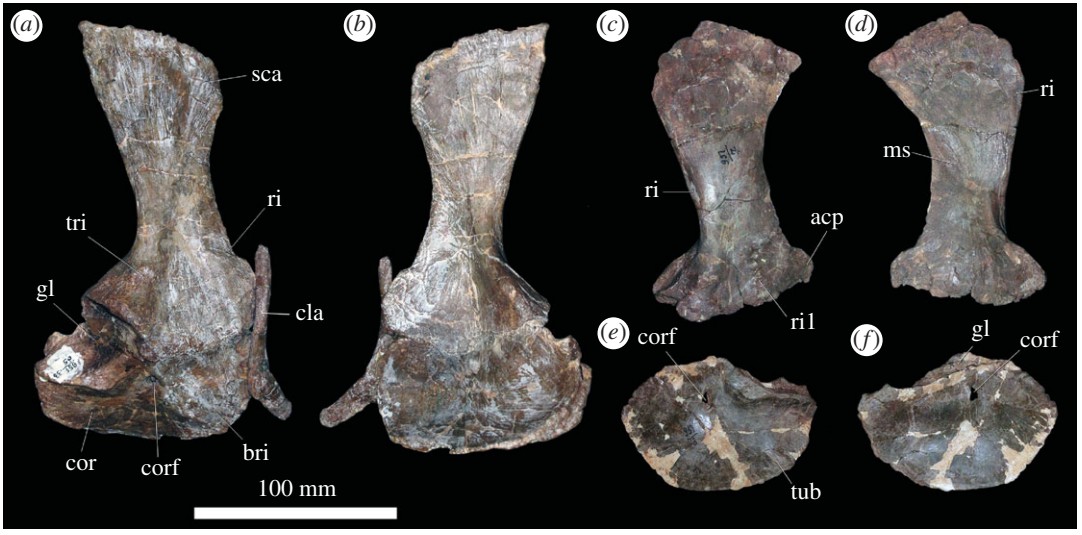

**Figure 12.** Scapulocoracoid. (*a*,*b*) PIN 951/4-1-1, right scapulocoracoid with fused clavicle in (*a*) lateral and (*b*) medial views. (*c*,*d*) PIN 951/7-1-1, right scapula in (*c*) lateral and (*d*) medial views. (*e*,*f*) PIN 951/2-1-2, left coracoid in (*e*) lateral and (*f*) medial views. **acp,** acromial process; **bri,** ridge for m. biceps; **cla,** clavicle; **cor,** coracoid; **corf,** coracoid foramen; **gl,** glenoid; **ms,** muscle scar; **ri,** ridge; **ri1,** ridge (see text); **sca,** scapula; **tri,** triceps tubercle; **tub,** tubercle. Scale bar equal to 100 mm.

the glenoid. The clavicle is preserved in contact with the anterior margin of the coracoid where it is preserved (figure 12*a*,*b*); in specimens in which it is not articulated, there is no clear facet for its contact.

In lateral view, the coracoid foramen pierces the centre of the bone just ventral to the articular surface for the scapula (dorsal margin), as in other archosauriforms, including *Erythrosuchus africanus* [4] (figure 12*e*,*f*) and *Shansisuchus shansisuchus* ([38], fig. 26a,f). The foramen is oval with the long axis extending dorsoventrally. Anterior to the coracoid foramen there is a shallow fossa. This fossa is bounded ventrally by a broad, gentle, smooth ridge that extends anteroventrally to the anteroventral corner of the coracoid. In the right coracoid of PIN 951/4-1-1 (figure 12*a*,*b*), this area is defined by two distinct, sharp ridges with a shallow fossa between them, although only a single ridge is present in other specimens (including the left element probably belonging to the same individual). The musculus (m.) biceps originates from this area in crocodylians and birds ([51] and references therein) and thus this ridge is perhaps associated with the attachment of this muscle. Posteriorly, the lateral surface of the coracoid is slightly rugose ventral to the glenoid, and a small protuberance is present along on the posteroventral margin. In the associated scapulocoracoid of PIN 951/2-1-2 (figure 12*e*,*f*), there is a distinct, laterally projecting tubercle in this area. This corresponds to the area of origin of m. coracobrachialis brevis ventralis in crocodiles and its homologue the m. coracobrachialis cranialis in birds [51] and this muscle probably arose from here in *Garjainia prima*. The same condition occurs in the holotype [9] and *G. madiba* [8]. The glenoid is defined by a distinct, laterally projecting lip that is dorsoventrally compressed where it extends from the posterior margin anteriorly, but this lip flares sharply as it turns dorsally to form a flattened, quadrilateral surface that is elevated from the lateral surface of the coracoid and extends to the dorsal margin of the bone at a point just posterior to the coracoid foramen. The glenoid itself is transversely broad, shallowly concave, and rather irregular and rugose. The glenoid fossa of PIN 951/4-1-1 faces somewhat laterally, although this is not the case in other specimens and is an artefact of deformation in this specimen.

In medial view, the coracoid is gently concave anteroposteriorly and dorsoventrally. The coracoid foramen exits the bone ventral to the dorsal margin (suture with the scapula), as in other archosauriforms (e.g. *Erythrosuchus africanus*: [4]; *Shansisuchus shansisuchus*: [38]). A shallow canal extends dorsally from the coracoid foramen towards the dorsal margin in a larger coracoid catalogued as PIN 951/2-2-2 and in the associated scapulocoracoid of PIN 951/2-1-2; this feature is more subtle in other specimens. The medial surface is smooth without clear evidence of muscle attachment features.

*Scapula.* The anterior margin of the scapula is strongly concave, while the posterior margin is straight to gently convex, as in other erythrosuchids (holotype of *Garjainia prima*: [9]; *Guchengosuchus shiguaiensis*: [52]; some specimens of *Erythrosuchus africanus*: NHMUK PV R3762a; *Shansisuchus shansisuchus*: ([38], fig. 26a) and *Euparkeria capensis* (SAM-PK-5867). By contrast, the posterior margin is more convex in lateral view in *Prolacerta broomi* (BP/1/2575), *Proterosuchus alexanderi* (NM QR 1484), '*Chasmatosaurus' yuani* (IVPP V2719) and *Sarmatosuchus otschevi* (PIN 2865/68). The dorsal margin is straight and angled anteroventrally when

the long axis of the blade is held vertically, whereas the ventral margin is approximately horizontal with a slight convexity. The dorsal end of the scapula is transversely compressed and flared anteroposteriorly; ventrally it narrows as the anterior and posterior margins converge, and then flares anteroposteriorly again as it approaches the contact with the coracoid. Anteroventrally, the ventral end of the bone is transversely compressed and plate-like; posteroventrally, it is greatly expanded transversely for the glenoid (figure 12a–d). The presence of a plate-like acromion also occurs in other non-eucrocopodan archosauriforms and *Euparkeria capensis* [34,50]. In overall shape the scapula is very similar to that of *E. africanus* [4], *Guchengosuchus shiguaiensis* [52] and some specimens of *S. shansisuchus* ([38], fig. 26), and almost identical to that of the holotype of *G. prima* (PIN 2394/5).

In lateral view, the lateral surface of the dorsal end of the scapula is slightly rugose, suggestive of a cartilaginous extension. Otherwise, the lateral surface of the blade is smooth and featureless. At the anteroposteriorly narrowest part of the scapular blade, there is a short ridge on the posterior margin; in birds and crocodiles, the m. scapulohumeralis caudalis originates in this area and it is likely that the ridge marks the scar for the origin of this muscle in this specimen of *Garjainia* (figure 12c: ri). A similar ridge in the same position also occurs in *Guchengosuchus shiguaiensis*, *Erythrosuchus africanus* and *Shansisuchus shansisuchus*, but it is absent in the holotypes of *Garjainia prima* (although the scapula is not well preserved in this region) and *G. madiba* [10]. Ventrally, the anterior margin of the scapula turns sharply and extends anteroventrally to form the anteriorly projecting acromial process. In PIN 951/4-1-1 (figure 12a,b), the clavicle articulates with the anterior surface of the acromial process. In other specimens, the anterior margin of the acromial process is flattened for this articulation. A short, low, but sharp ridge extends posterodorsally from the dorsal margin of the acromial process (figure 12a: ri); this is absent in the holotype [9]. In reptiles that retained a clavicle, the m. deltoideus clavicularis arises from the dorsal end of the clavicle, whereas in reptiles that lost the clavicle, the m. deltoideus clavicularis migrated onto the acromial process [53]. In *G. prima*, the ridge is located just ventral to the dorsal end of the clavicle, and thus the ridge might indicate that some part of the m. deltoideus clavicularis originated on the acromial process. In the specimens preserved, there is some variation in the degree of anterior projection of the acromial process; it appears to project less far anteriorly and at a slightly more gentle angle in the holotype and PIN 951/4-1-1 and 4-1-2; these are the same specimens in which the scapula and coracoid are fused.

On the lateral surface of each of a small pair of scapulae (PIN 951/7-1-1 and PIN 951/4-3), there is a narrow ridge with anteroposteriorly extending striations across it (figure 12c). The ridge extends from a location on the anterior margin just dorsal to the acromial process and extends posteriorly a short distance before turning ventrally to join the ventral margin (articular surface for the coracoid), approximately midway between the anterior and posterior surfaces (figure 12c: ri1). These ridges are symmetrical on both scapulae but are not observed in any of the other specimens present, or the holotype. These ridges do not appear to correspond with locomotor musculature in extant archosaurs [51] and their function is unknown.

On the posterolateral surface of the scapula, dorsal to the glenoid, there is a large, rugose, round tubercle (figure 12a: tri). In crocodiles, the m. triceps longus lateralis and its avian homologue the m. scapulotriceps originate from this region, and thus it is likely that this is the area of origin of the m. scapulotriceps in *Garjainia prima*. The scapula contribution to the glenoid is transversely expanded and in ventral view is approximately oval with the long axis orientated anteroposteriorly. It is shallowly concave and its edges are defined by a distinct lip. The medial surface of the glenoid region bears a dorsoventrally extending shallow groove, and the surface is rugose. It is possible that this rugosity represents additional origination of fibres of the m. scapulotriceps, but it could also be related to connective tissues associated with the joint capsule.

In medial view, the scapula is gently bowed along its length such that the medial side is dorsoventrally gently concave (figure 12b,d). Dorsally, a shallow ridge extends down the posterior margin just inset from the edge of the bone; this is clearer in larger specimens and is similar but less clear to the ridge reported in the scapula of the holotype ([9], fig. 20, r). Gower [4] reported a similar ridge in *Erythrosuchus africanus*, although it is slightly more ventral in that taxon. On the anteromedial surface of the blade, just dorsal to the anteroposteriorly narrowest part of the blade, there is a rugose, striated muscle scar that projects slightly medially. Sharpey's fibres on this muscle scar extend anterodorsally to posteroventrally. This muscle scar was not reported for *Erythrosuchus africanus* by Gower [4], but is present on the holotype of *Garjainia prima* [9]. The avian and crocodilian m. subscapularis originate on the medial surface of the scapula so this scar is probably the area of origin of this muscle in *G. prima*. The development of this muscle scar is variable, and the largest scapulae do not always have the best-developed muscle scars. Indeed, the scapulae with fused

coracoids (PIN 951/4-1-1 and 4-1-2) have less well-developed scars in this area than some of the smaller scapulae. The ventral end of the scapula is strongly concave anteroposteriorly due to the extremely strong transverse expansion of the glenoid and articular surface of the coracoid, which is expanded further medially than it is laterally.

*Clavicle.* A single right clavicle is preserved in articulation with the acromial process and anterolateral surface of the coracoid of PIN 951/4-1-1 (figure 12*a*,*b*). The clavicle is an anteroposteriorly narrow but dorsoventrally elongate rod of bone. Its anterior margin is concave while its posterior margin is convex, so that it is bowed in lateral view. It appears slightly incomplete dorsally, and ventrally it is broken along its posterior margin. The dorsal end of the bone does not taper as strongly as in *Prolacerta broomi* (BP/1/ 2675) and '*Chasmatosaurus' yuani* (IVPP V4067). It is anteroposteriorly compressed dorsally and expands ventrally to become transversely compressed and anteroposteriorly broader. At its ventral end, it curves anteriorly. Its ventral end is rounded and does not taper. Medially, at its ventral end, it is concave anteroposteriorly. The specimen is rather different from the clavicle of the holotype of *Garjainia prima* (PIN 2394/5-35), in which the curvature is similar, but the ventral end is twisted so that it is strongly anteroposteriorly compressed and broadly flared transversely. The differences in these elements are probably due to some plastic deformation of the clavicle of PIN 951/4-1-1.

*Interclavicle.* Huene ([16], pl. 14, fig. 7, 8]) figured two incomplete interclavicles PIN 951/20-1, 2. Their morphology appears entirely consistent with that of the holotype of *Garjainia prima*, as described in [9].

### 3.3.2. Humerus

Limb bones are described with orientations as in Gower [4] (figure 13). In ventral view, the humerus is dumbbell shaped with expanded proximal and distal ends and a relatively narrow shaft (figure 13*b*). The proximal end is slightly more expanded than the distal end, and the long axes are slightly offset from each other, as in *Guchengosuchus shiguaiensis* [52], *Garjainia madiba* [8], *Erythrosuchus africanus* [4] and *Shansisuchus shansisuchus* [38]. By contrast, the long axes of the proximal and distal ends are offset from each other at an angle of 45° or higher in proterosuchids and *Prolacerta broomi* [34]. The proximal surface of the element is slightly concave in all specimens, even in the largest, indicating incomplete ossification. The ventral surface of the proximal end is flat, smooth, and the proximal end is dorsoventrally compressed. In well-preserved specimens there is a small, round muscle scar just medial to the midline on the ventral surface of the proximal end. In crocodiles and birds, the m. coracobrachialis brevis and its homologues insert in this area, and this muscle scar may be related to the insertion of that muscle in *G. prima*. In ventral view, the medial edge of the humerus forms a proximodistally extending broad ridge that extends towards the shaft, tapering before the constricted shaft. The top of this ridge is rugose (the medial tuberosity), particularly on its medial surface in larger specimens. In birds and crocodiles, the m. subscapularis inserts on the medial tuberosity, and in birds, the m. subcoracoideus also attaches here. Laterally, the ventral surface of the proximal end is gently curved ventrally to form the deltopectoral crest. The outline of the crest in ventral view is a smooth semicircle. The deltopectoral crest is restricted to the proximal part of the humerus, as occurs in other erythrosuchids with the exception of *E. africanus*, where in large individuals it extends halfway down the humerus [4]. A tubercle, located proximal to the apex of the deltopectoral crest and on the margin of it, projects ventrally and is prominent, even in the smallest specimen (figure 13*b*). In birds and crocodiles, the m. pectoralis inserts on the apex of the deltopectoral crest, and it is possible that this tubercle represents the insertion of this muscle in *G. prima* because this tubercle is more prominent and more striated than the apex of the deltopectoral crest itself. However, in crocodiles, the m. supracoracoideus longus, intermedius and brevis also insert close to the apex of the deltopectoral crest. It is possible, therefore, that this tubercle is the insertion of the m. supracoracoideus complex as well.

The shaft is oval in cross-section and narrow. Centrally on the shaft, immediately distal to the narrowest part, there is a small, rugose muscle scar, visible on well-preserved specimens. Based on comparisons with crocodiles and birds, this may represent the origin of the m. brachialis. The distal end is slightly concave, similarly to the proximal end. The distal end of the ventral surface is divided into two similarly sized condyles separated by a shallow fossa. The articular surface of the lateral condyle extends onto the ventral surface of the bone, but that of the medial condyle does not, as occurs in '*Chasmatosaurus' yuani* (IVPP V2719), *Garjainia madiba* (BP/1/7336) and *Shansisuchus shansisuchus* [38]. However, the articular surface of both condyles is restricted to the ventral surface of the bone in *Erythrosuchus africanus* (SAM-PK-905). The ventromedial surface of the medial condyle is slightly rugose, while the lateral surface of the lateral condyle is drawn into a dorsoventrally

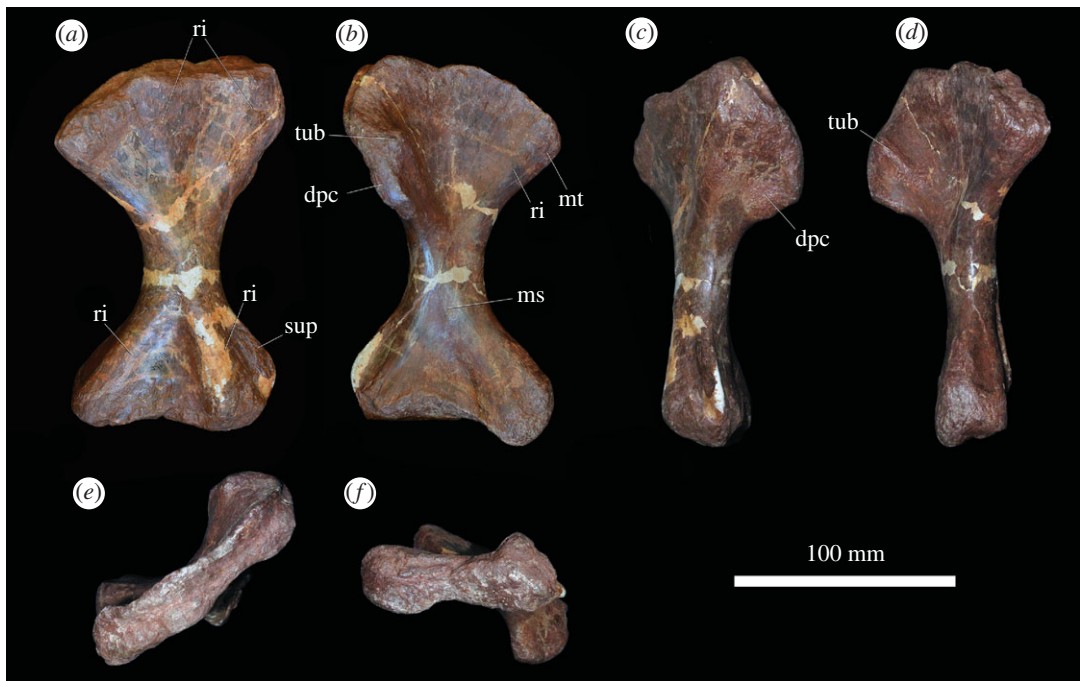

**Figure 13.** Humerus. PIN 951/36-1, right humerus in (*a*) dorsal, (*b*) ventral, (*c*) lateral, (*d*) medial, (*e*) proximal and (*f*) distal views. **dpc**, deltopectoral crest; **ms**, muscle scar; **mt**, medial tuberosity; **ri**, ridge; **sup**, supinator ridge; **tub**, tubercle. Scale bar equal to 100 mm.

compressed supinator ridge for the attachment of the supinator musculature [53], as in *Prolacerta broomi* (BP/1/2675), proterosuchids (e.g. '*Chasmatosaurus*' *yuani*: IVPP V2719) and other erythrosuchids (e.g. *Guchengosuchus shiguaiensis*: [52]; *E. africanus*: [4]; *G. madiba*: [8]). By contrast, a supinator ridge is absent in *Euparkeria capensis* [36] and proterochampsids [54]. The supinator ridge of *G. prima* does not extend as far ventrally as the lateral condyle.

In dorsal view, the proximal end of the humerus bears two broad, shallow ridges that extend proximally to form peaks on the margin of the bone; these are separated by a shallow fossa (figure 13*a*), as is the case in other early archosauriforms (e.g. '*Chasmatosaurus*' *yuani*: IVPP V2719; *Garjainia madiba*: BP/1/7336; *Erythrosuchus africanus*: SAM-PK-905). On the medial ridge there is an area of subtle rugosity immediately ventral to the proximal margin and this is likely to be the muscle scar for the m. scapulohumeralis caudalis based on comparisons with crocodylians and birds [51]. On the lateral surface of the lateral ridge, a distally extending rugose ridge is interpreted as an origin of the m. triceps brevis based on the often distinct scar here in extant archosaurs [51]. A humeral head for articulation with the scapula is absent, as is usual in early archosauromorphs [34]. The dorsomedial surface (the dorsal surface of the medial tuberosity) is rugose and striated. The dorsal surface of the deltopectoral crest is rugose in the largest specimens, and comparison with crocodylians and birds suggests that this rugosity is for the insertion of the m. deltoideus clavicularis and scapularis. The distal end of the humerus is divided into two prominent ridges that extend from the shaft distomedially and distolaterally. The distomedial ridge is rugose proximal to the condylar surface for the attachment of lower limb flexor musculature [53]. The ridges are separated by a deep fossa. A deep cleft separates the supinator ridge from the distolateral ridge and lateral condyle in dorsal view; this extends onto the distal end of the bone, which presumably housed the radial tendon [53], as occurs in proterosuchids and other erythrosuchids (e.g. '*Chasmatosaurus*' *yuani*: IVPP V2719; *G. madiba*: BP/1/7336; *E. africanus*: SAM-PK-905). The distal end of the supinator ridge is blunt.

### 3.3.3. Ulna

In the proximal view, the ulna is 'D'-shaped, with the flat medial margin and curved lateral margin (figure 14*f*). This overall proximal outline resembles that of most other early archosauriforms, but they have a slightly concave medial margin (e.g. '*Chasmatosaurus*' *yuani*: IVPP V2719; *Cuyosuchus huenei*: MCNAM PV 2669; *Garjainia madiba*: BP/1/6232r). By contrast, *Shansisuchus shansisuchus* [38] and

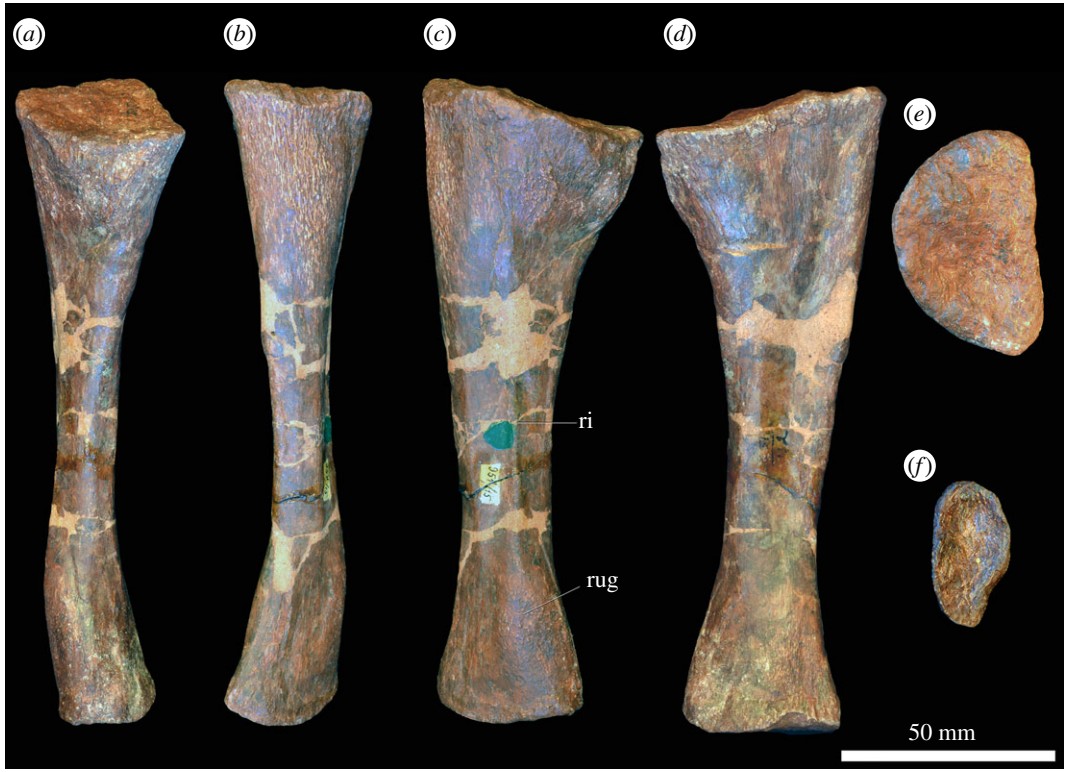

**Figure 14.** Ulna. PIN 951/38-1, right ulna in (*a*) dorsal, (*b*) ventral, (*c*) lateral, (*d*) medial, (*e*) proximal and (*f*) distal views. **ri**, ridge, **rug**, rugosity. Scale bar equal to 50 mm.

*Erythrosuchus africanus* (SAM-PK-905) have a proportionally transversely broader proximal end of ulna, with a more concave medial margin, and the ulna of *Proterosuchus alexanderi* is 'L'-shaped in proximal view (NMQR 1484). The proximal surface is flat to gently concave and there is a very low olecranon process, similar to the condition in proterosuchids (e.g. *'Chasmatosaurus' yuani*: IVPP V2719; *Proterosuchus fergusi*: SAM-PK-K140), erythrosuchids (e.g. *G. madiba*: [8]; *E. africanus*: [4]; *S. shansisuchus*: [38]), *Cuyosuchus huenei* (MCNAM PV 2669) and *Euparkeria capensis* (SAM-PK-5867). In lateral and medial views, the ulna is dorsoventrally expanded at the proximal end, tapers to the shaft and then expands dorsoventrally again slightly at the distal end (figure 14*c,d*). The dorsal margin is concave, especially proximally, while the ventral margin is straight to gently convex proximally. The lateral surface of the element is rugose dorsolaterally at the proximal end for articulation with the head of the radius. The lateral surface lacks a radial ridge, as is the case in other non-archosaurian archosauriforms [34,50]. The ulna is also rugose ventrally at the proximal end, presumably for insertion of the m. triceps. A clear, narrow ridge extends from distal to the rugosities down the centre of the shaft in lateral view and joins a prominent, raised, proximodistally oval rugosity at the distal end for articulation with the distal end of the radius. A similar ridge is present in *E. africanus* [4] and probably represents an intermuscular line. The medial surface of the ulna is flat proximally with a few proximodistally extending striations that may be related to attachment of the m. pronator quadratus. Gower [4] described a prominent ridge on the medial surface in *E. africanus*, but such a feature is not present in *G. prima*. Near the distal end, the ulna becomes dorsoventrally convex such that its cross-section becomes rounded. In the distal view, the bone is oval in outline with the long axis trending anteromedially, and it is shallowly concave (figure 14*e*), resembling the condition in *G. madiba* (BP/1/6232r). By contrast, the distal end of the ulna is transversely broader in *E. africanus* (SAM-PK-905) and *S. shansisuchus* [38].

### 3.3.4. Radius

In the proximal view, the radius is oval with a transverse long axis (figure 15*e*), resembling the condition in *Erythrosuchus africanus* (SAM-PK-905) and *Shansisuchus shansisuchus* [38], but the proximal end of the bone is proportionally dorsoventrally deeper in the latter two species. By contrast, the radius of *'Chasmatosaurus' yuani* is 'D'-shaped in the proximal view, with a flat medial margin. The proximal

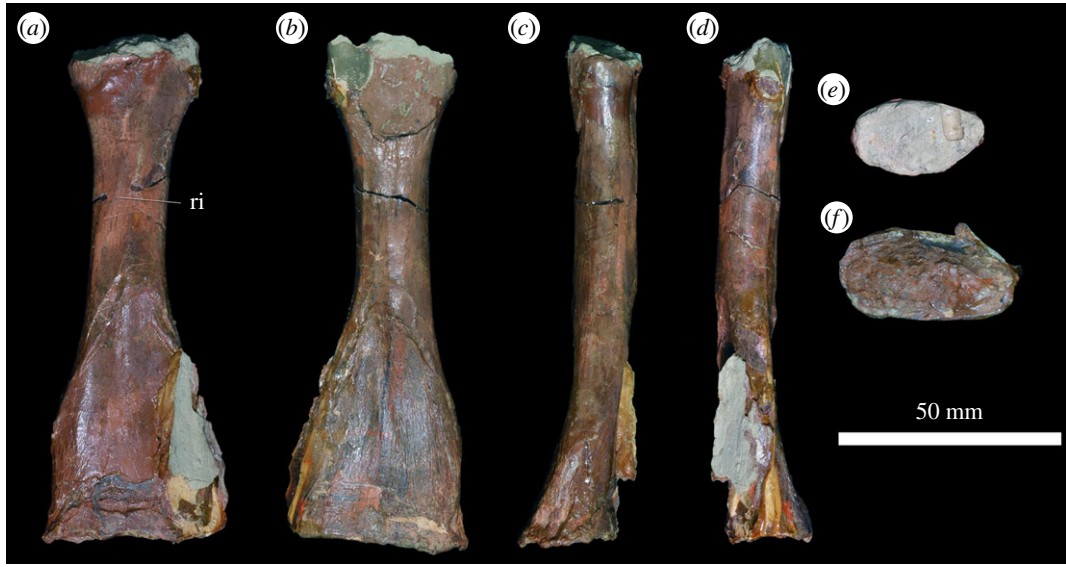

**Figure 15.** Radius. PIN 951/131 right radius in (*a*) dorsal, (*b*) ventral, (*c*) lateral, (*d*) medial, (*e*) proximal and (*f*) distal views. **ri**, ridge. Scale bar equal to 50 mm.

surface is convex. In dorsal view, the proximal end of the element is transversely narrower than the distal end and both dorsal and ventral margins are concave. *Garjainia prima* has a strongly transversely expanded distal end of radius, which is a character state that has been found as a synapomorphy of Erythrosuchidae [34]. The dorsal surface of the shaft is convex, and distal to the proximally expanded area there are two proximodistally extending ridges, probably representing intermuscular lines, that taper distally before the expansion of the distal end. In a smaller unnumbered specimen (figure 15), only a single ridge is present, and it extends further distally (figure 15*a*). Two longitudinal ridges are also present on the radius of *E. africanus* [4]. Medially, the radius is flat, and the flattened surface is delimited by dorsal and ventral ridges extending along the shaft. A roughened tubercle is present on the ventral ridge immediately proximal to mid-length. In the distal view, the radius has an oval to 'D'-shaped cross-section, with the flattened ventral edge and transversely orientated long axis (figure 15*f*), resembling the condition in *S. shansisuchus* [38]. By contrast, the radius of *E. africanus* has a subcircular profile in the distal view (SAM-PK-905). The distal end of the radius of *G. prima* is concave.

### 3.3.5. Ilium

In lateral view, the ilium has a short, semicircular and transversely broad preacetabular process, resembling the condition in *Prolacerta broomi* (BP/1/2676), *Proterosuchus alexanderi* (NMQR 1484) and '*Chasmatosaurus*' *yuani* (IVPP V4067) (figure 16). The preacetabular process of *Garjainia madiba* (BP/1/5525) is also semicircular, but proportionally shorter and dorsoventrally lower than in *G. prima*. Contrasting with all the latter taxa, the preacetabular process of *Erythrosuchus africanus* (NHMUK PV R3592), *Shansisuchus shansisuchus* [38], *Cuyosuchus huenei* [55] and *Euparkeria* (SAM-PK-6049) tapers anterodorsally and is slightly longer. The postacetabular process of *G. prima* is elongate and blade-like, with a slightly dorsally concave dorsal margin and a bluntly rounded posterior end (figure 16*a*). This process differs from the autapomorphically longer process of *G. madiba* [8] and also from the processes of *E. africanus* [4] and *S. shansisuchus* [38], which have a straight and a more strongly concave dorsal margin, respectively. A broad ridge extends horizontally along the postacetabular process, tapering before its distal end, making the lateral surface of the postacetabular process convex dorsoventrally. The ventral margin of the postacetabular process turns medially ventral to the ridge and is drawn out into a thin flange; this surface is striated and the m. caudofemoralis brevis would have originated there [56]. The dorsal margin of the ilium is folded slightly medially, so that in medial view the inner surface of the preacetabular process is strongly concave. It is bounded ventrally by the medial preacetabular shelf (*sensu* [56]), while the surface of the postacetabular process is more gently concave medially. The entire dorsal and dorsolateral margin of the ilium is heavily striated for attachment of the m. iliotibialis [56] as in other early archosauromorphs, such as '*Chasmatosaurus*' *yuani* (IVPP V4067), *E. africanus* [40] and *Dorosuchus neoetus* (PIN 1579/61). The posterior half of the lateral surface of the postacetabular process

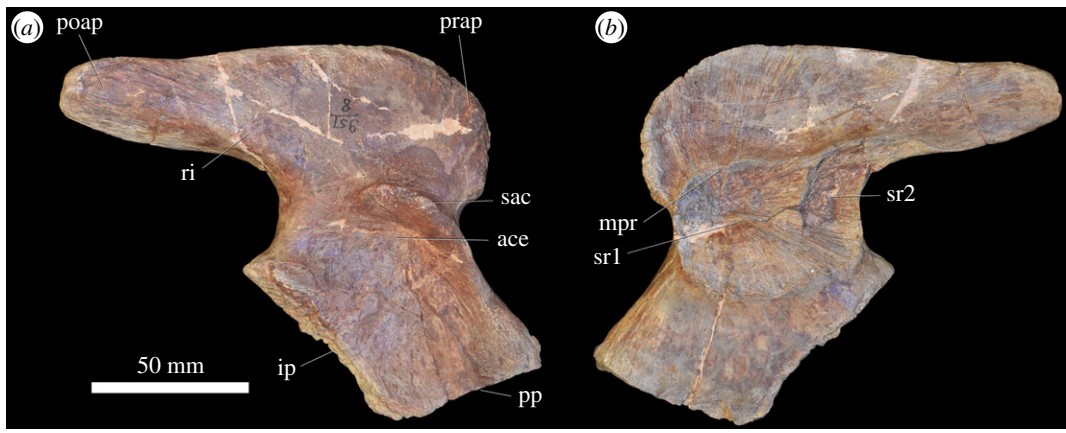

**Figure 16.** Ilium. PIN 951/8-2-1, right ilium in (*a*) lateral and (*b*) medial views. **ace**, acetabulum; **ip**, ischiadic peduncle; **mpr**, medial preacetabular ridge; **poap**, postacetabular process; **pp**, pubic peduncle; **prap**, preacetabular process; **ri**, ridge; **sac**, supracetabular crest; **sr1**, facet for sacral rib 1; **sr2**, facet for sacral rib 2. Scale bar equal to 50 mm.

is also heavily striated, probably associated with the origin of the m. fibularis and, more dorsally, the m. flexor tibialis externus based on the condition in extant archosaurs [51]. Dorsal to the acetabulum, the iliac blade is smooth, as in other non-archosaurian archosauriforms [34,50]. Along the dorsal margin of the acetabulum there is a distinct, rugose, laterally projecting supraacetabular crest that has a distinct posterior oval margin, resembling the condition in *Proterosuchus alexanderi* (NMQR 1484), *G. madiba* (BP/1/5525) and *E. africanus* (NHMUK PV R3592). In *G. prima* and the latter species, the supraacetabular crest ends well before the base of the ischiadic peduncle, but in *S. shansisuchus* this crest reaches this area and frames the complete dorsal border of the acetabulum [38]. The supraacetabular crest of *G. prima* tapers anteriorly towards the anterior edge of the pubic peduncle, similar to other archosauromorphs. The articular surfaces for the pubis and ischium are anteroposteriorly elongate, transversely compressed and rugose. The articular surface of the pubic peduncle is angled posteroventrally, while that of the ischiadic peduncle is angled anteroventrally, and they meet at an apex approximately level with the middle of the supraacetabular crest. Thus, the acetabulum is entirely closed medially, as in other non-archosaurian archosauromorphs [50], and it is shallowly concave (see electronic supplementary material, figure S3 for articulating pelvic elements). The ventral end of the ischiadic peduncle is strongly expanded posteriorly, forming an acute 'heel', as occurs in *G. madiba* [8] and *E. africanus* [4], but not in *S. shansisuchus* [38].

Medially, the concave medial surface dorsal to the medial preacetabular ridge is heavily striated (figure 16*b*), presumably for the attachment of epaxial musculature. The medial preacetabular ridge delimits the dorsal border of the articulation of the first sacral rib. This rib facet is highly rugose and shallowly concave and is bordered anteriorly and ventrally by distinct lips. The medial preacetabular ridge extends posteriorly to form the dorsal border of the articular facet for the second sacral rib, which is smaller than the facet for the first and is anteroposteriorly shorter. The facets for the first and second sacral ribs are separated by a vertical ridge. The shape and position of the sacral rib facets resemble those of *Garjainia madiba* [8], but the facet for the first sacral rib is extended further dorsally in *Erythrosuchus africanus* (NHMUK PV R3592) and *Shansisuchus shansisuchus* [38]. The medial surface of the acetabular region below the sacral rib facets is smooth and flat.

### 3.3.6. Pubis

The pubis is composed of a transversely thickened iliac peduncle, a rod-like shaft, and a pubic apron composed of a thin, angled, medially projecting plate (figure 17). The iliac peduncle is approximately oval in outline with the long axis trending dorsoventrally. It is divided subtly into two discrete surfaces: a dorsally facing surface that articulated with the pubic penduncle of the ilium, and a dorsolaterally facing surface that contributes to the acetabulum (figure 17*d*; electronic supplementary material, figure S3), as is the case in other early archosauromorphs [57]. Lying distal and lateral to the iliac peduncle on the dorsal surface of the pubis is a prominent pubic tubercle (*sensu* [56]), which continues anteriorly as a rugose ridge; this is the area of origin of the m. ambiens, pelvic ligaments, the m. obliquus abdominus and the m. pubotibialis [56]. The pubic tubercle is much more developed

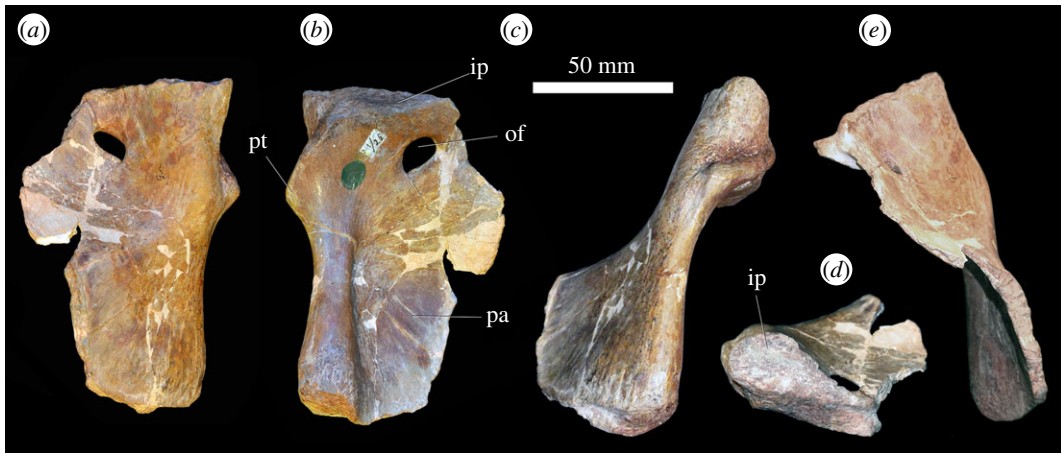

**Figure 17.** Left pubis, PIN 951/5-3, in (*a*) anterodorsal, (*b*) posteroventral, (*c*) lateral, (*d*) posterodorsal and (*e*) medial views. **ip**, iliac peduncle; **of**, obturator foramen; **pa**, pubic apron; **pt**, pubic tubercle. Scale bar equal to 50 mm.

than in *Erythrosuchus africanus* [4], whereas *Garjainia madiba* possesses a similarly developed tubercle, but it is mainly anteriorly oriented (BP/1/5360). Anterior to the ventral part of the iliac peduncle, the element is pierced by the obturator foramen, which is oval with the long axis trending anteroposteriorly (figure 17*a*,*b*). Immediately ventral to the obturator foramen, the pubis turns medially to form a broad, thin plate, concave on its dorsal surface and convex on its ventral surface. Distal to the pubic tubercle, the pubis turns ventrally, and the lateral margin is thickened. Medial to the thickened lateral margin, the pubis expands medially in a broad, anteroposteriorly thin, plate-like pubic apron, as occurs in other erythrosuchids and eucrocopods [34]. By contrast, the pubic apron is absent in *Prolacerta broomi* (BP/1/2676) and proterosuchids (e.g. *Proterosuchus alexanderi*: NMQR 1484; '*Chasmatosaurus*' *yuani*: IVPP V4067). The pubic apron of *G. prima* extends posterodorsally to join the dorsally facing plate almost at right angles to it. The distal end of the plate is gently ventrally convex. The distal end of the pubic shaft is flat to slightly concave and slightly anteroposteriorly expanded, resembling the condition in *E. africanus* (NHMUK PV R3592) and *Cuyosuchus huenei* (MCNAM PV 2669). The pubis met its counterpart on the midline, and based on the articulating material (PIN 951/48-53), the dorsal part of the pubic apron articulated with the ischium medioventrally below the ilium to form a well-developed pubo-ischiadic plate.

### 3.3.7. Ischium

The ischium is approximately rectangular in lateral view with the long axis extending from anterodorsal to posteroventral, and the iliac peduncle projecting anterodorsally from the anterodorsal corner of the bone in a natural position (figure 18; electronic supplementary material, figure S3). The iliac peduncle is transversely expanded, and the articular surface is oval in outline with the long axis oriented dorsoventrally. The articular surface is divided into a medial surface for articulation with the ilium, facing anterodorsally, and a lateral surface, which forms the ischial contribution to the acetabulum, facing dorsolaterally. Ventral to the iliac peduncle, there is a narrow facet that extends ventrally onto the ischiadic apron for articulation with the pubis (figure 18*a*), as is the case in other non-archosaurian archosauriforms [50]. The posterior surface of the iliac peduncle curves gently posteroventrally to form the dorsal margin of the ischium, and this dorsal margin is concave proximally in lateral view, before curving gently to become upwardly convex towards the distal end of the element.

Distal to the iliac peduncle, but proximal to mid-length, a ridge (the ischial ridge of [56]) arises on the lateral surface of the ischium and extends posteroventrally before tapering close to the distal end of the element, as occurs in *Erythrosuchus africanus* (NHMUK PV R3592). Medial to the ischial ridge on the dorsal surface of the ischium there is a shallow concavity that was probably the area of origin for the m. flexor tibialis internus 3 [56]. Distal to the ridge, the lateral surface of the ischium is rugose, and this rugose area broadens distally to the end of the element. This rugosity is probably for the origin of the m. adductor femoris 2 and, at the distal end, m. flexor tibialis internus 1 [56]. Ventral to the ischial ridge, the ischium extends ventromedially in a broad, transversely compressed ischial apron that is shallowly concave on its lateral side and shallowly convex on its medial side. The

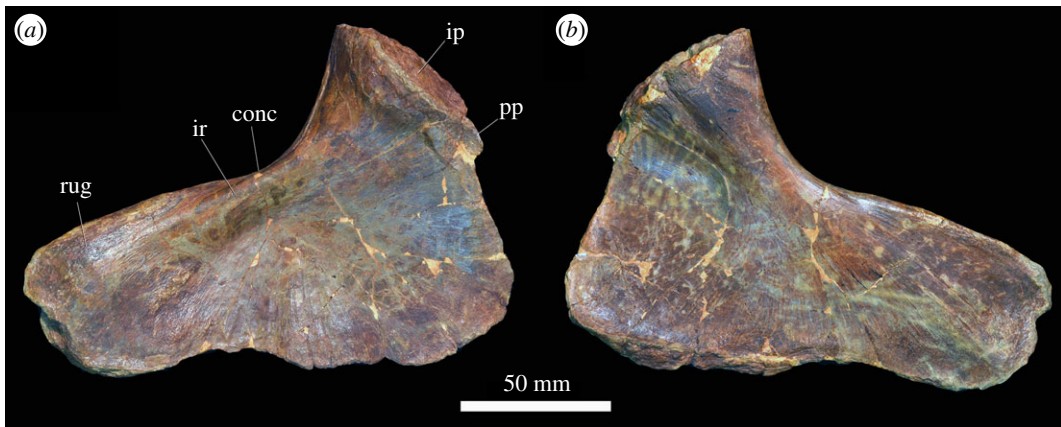

**Figure 18.** PIN 951/1-2-2 right ischium, in (*a*) lateral and (*b*) medial view. **conc**, concavity; **ip**, iliac peduncle; **ir**, ischial ridge; **pp**, pubic peduncle; **rug**, rugosity. Scale bar equal to 50 mm.

presence of a plate-like ischium in *Garjainia prima* resembles the condition in other non-archosaurian archosauriforms [34,50]. The ischium met its counterpart along its dorsoventrally narrow ventromedial edge to form a straight midline symphysis. On the distolateral surface of the apron there is a small tubercle on the largest specimen, developed as a rugosity on the smaller specimen. This muscle scar might be for the origin of the m. puboischiofemoralis externus 3 [56]. The ventral margin of the ischium is slightly concave in lateral view along the symphyseal margin, as occurs in *G. madiba* (BP/ 1/552) and *E. africanus* (NHMUK PV R3592), but in *Shansisuchus shansisuchus* the ventral margin of the ischium is deeply notched slightly posteriorly to the mid-length of the bone [38]. The distal end of the ischium possesses a pointed or a strongly convex edge on the dorsodistal margin, resembling the condition in *G. madiba* (BP/1/552) and *E. africanus* (NHMUK PV R3592). By contrast, the distal margin of the ischium of *S. shansisuchus* has a pointed end positioned approximately at mid-height [38].

### 3.3.8. Femur

In the dorsal view, the femur is slightly sigmoid, with the proximal end offset somewhat medially from the distal end (figure 19*c*). In lateral view, it is roughly straight, with the distal condyles projecting somewhat ventrally (figure 19*b*). The proximal end is 'C'-shaped in outline, with the opening facing ventrally; the articular surface is slightly concave, and in dorsal view, it is angled ventromedially (figure 19*e*), resembling the condition in *Garjainia madiba* (BP/1/5767). By contrast, the proximal outline of the femur is fusiform, with an anteromedial-to-posterolateral main axis and a broader medial edge, in *Erythrosuchus africanus* (NHMUK PV R3592) and *Shansisuchus shansisuchus* [38]. In ventral view, the proximal end is occupied by the large and deeply concave intertrochanteric fossa, as occurs in *Prolacerta broomi* (BP/1/2676), *Proterosuchus fergusi* (SAM-PK-K140), *G. madiba* [8] and *E. africanus* (NHMUK PV R3592), but *S. shansisuchus* and eucrocopodans lack such a fossa [34]. Medial to the fossa, the internal trochanter of *G. prima* projects medially as a transversely compressed flange with a gently convex medial margin. The internal trochanter reaches the proximal articular surface of the bone, as is the case in *Prolacerta broomi* (BP/1/2676), *Proterosuchus fergusi* (SAM-PK-K140) and *G. madiba* [8], but in *E. africanus* the internal trochanter is more distally located [4]. Immediately distal to the internal trochanter there is a small, rounded, and rugose area that continues onto the medial surface of the femur which is probably a muscle scar (figure 19*a*). The internal trochanter and the intertrochanteric fossa are thought to be the areas of insertion of the m. puboischiofemoralis externus, m. caudofemoralis longus et brevis and m. iliofemoralis in non-eucrocopodan archosauromorphs [58], and the small rounded scar distal to the internal trochanter may be the m. caudofemoralis longus et brevis insertion site in *G. prima* (see Discussion).

The lateral wall of the intertrochanteric fossa is transversely thickened and rounded in cross-section in ventral view. Proximally, a number of striations extend distally from the articular surface, probably associated with attachment of the joint capsule. A distinct, rugose, laterally projecting muscle scar is present on the ventrolateral surface of the femoral shaft at about mid-length, and it continues around onto the lateral and dorsal surfaces, where it extends proximally, before disappearing a short distance distal to the proximal articular surface (figure 19*a,b*). This muscle scar, also observed in *Erythrosuchus*

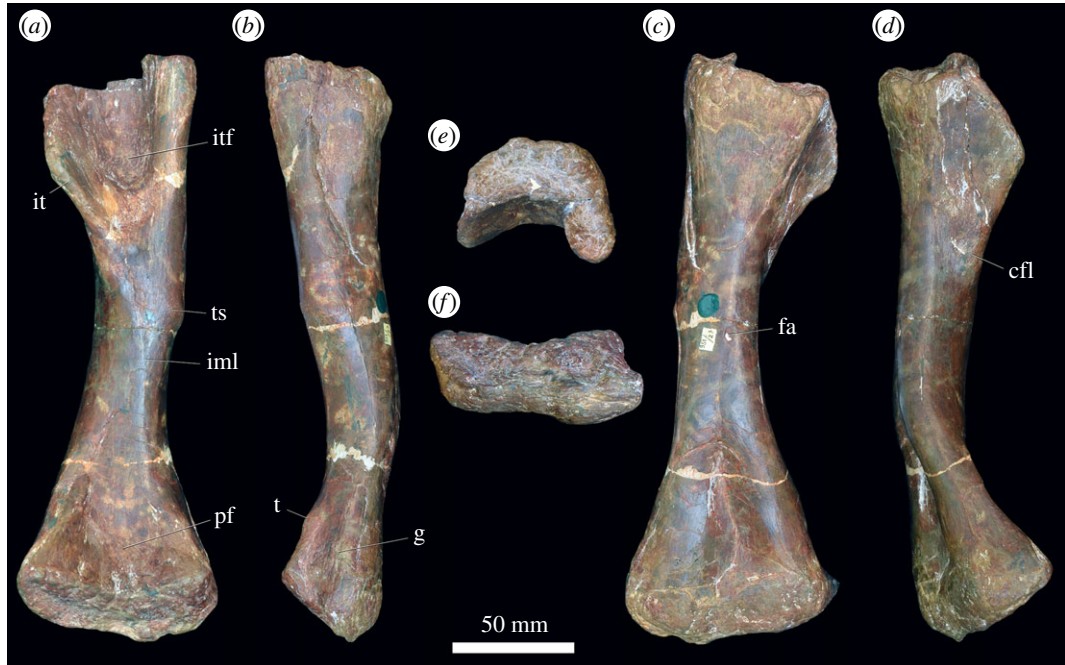

**Figure 19.** PIN 951/61-1, right femur, in (*a*) ventral, (*b*) lateral, (*c*) dorsal, (*d*) medial, (*e*) proximal and (*f*) distal views. **cfl**, muscle scar, probably for the m. caudofemoralis longus et brevis; **fa**, foramen for the femoral artery; **g**, groove; **iml**, intermuscular line; **it**, internal trochanter; **itf**, intertrochanteric fossa; **pf**, popliteal fossa; **t**, tubercle; **ts**, trochanteric shelf. Scale bar equal to 50 mm.

*africanus* [4] and *Garjainia madiba* [8], is interpreted as being homologous with the trochanteric shelf, for insertion of the m. iliofemoralis [58]. The lack of a lesser trochanter lying dorsal to the trochanteric shelf in *Garjainia prima* is indicative of a single-headed m. iliofemoralis following Hutchinson [58]. A distinct muscle scar for the insertion of the m. iliofemoralis is absent in *Prolacerta broomi* (BP/1/2676), *Proterosuchus fergusi* (SAM-PK-K140), *Dorosuchus neoetus* (PIN 1579/61) and *Euparkeria capensis* (SAM-PK-5883). Gower [4] was able to identify an additional muscle scar in *E. africanus* proximal to the trochanteric shelf, possibly for insertion of the m. ischiotrochantericus, but no other muscle scars can be observed in the preserved material of *G. prima*.

In ventral view, an intermuscular line (the 'adductor ridge') extends distally from the muscle scar for the m. caudofemoralis longus towards the lateral condyle (figure 19*a*), as also occurs in *Erythrosuchus africanus* [4]. A second intermuscular line extends from the trochanteric shelf to the same location as the first, and they merge proximal to the distal end. The area between these intermuscular lines is flat and slightly roughened for the attachment of the m. adductor femoris based on the condition in archosaurs [58]. Distally, where the lines merge, there is a small, ventrally projecting tubercle on the lateral condyle (figure 19*b*). In dorsal view, a large foramen, probably for the femoral artery, can be observed at approximately mid-length on the medial side of the shaft (figure 19*c*: fa). A shallow popliteal fossa is present between the distal condyles in ventral view. A shallow groove (figure 19*b*: g) separates the tibiofibular crest and fibular condyle from each other on the lateral surface of the distal end. The distal condyles are shallowly separated in dorsal view by a broad extensor fossa, as occurs in other early archosauriformes [34]. In the distal view, the femur has a relatively poorly developed tibiofibular crest separated from the fibular condyle by an obtuse angle. The lateral margin of the fibular condyle is convex. The tibial condyle has a posteromedially oriented apex, as occurs in *Prolacerta broomi* (BP/1/2676), *Proterosuchus fergusi* (SAM-PK-K140), '*Chasmatosaurus*' *yuani* (IVPP V2719), *Garjainia madiba* (BP/1/5767), and at least some specimens of *Shansisuchus shansisuchus* ([38], fig. 31a), and *Dorosuchus neoetus* (PIN 1579/61). By contrast, the margins of the tibial condyle are continuously convex in *E. africanus* (NHMUK PV R3592) and *Euparkeria capensis* (SAM-PK-5883).

### 3.3.9. Tibia

In dorsal view, the tibia is straight, with transversely expanded distal ends and proximal ends (figure 20*a*). In the proximal view, the outline of the proximal end is suboval with a transverse long axis (figure 20*e*). In the distal view, the outline of the distal end is subcircular, slightly transversely

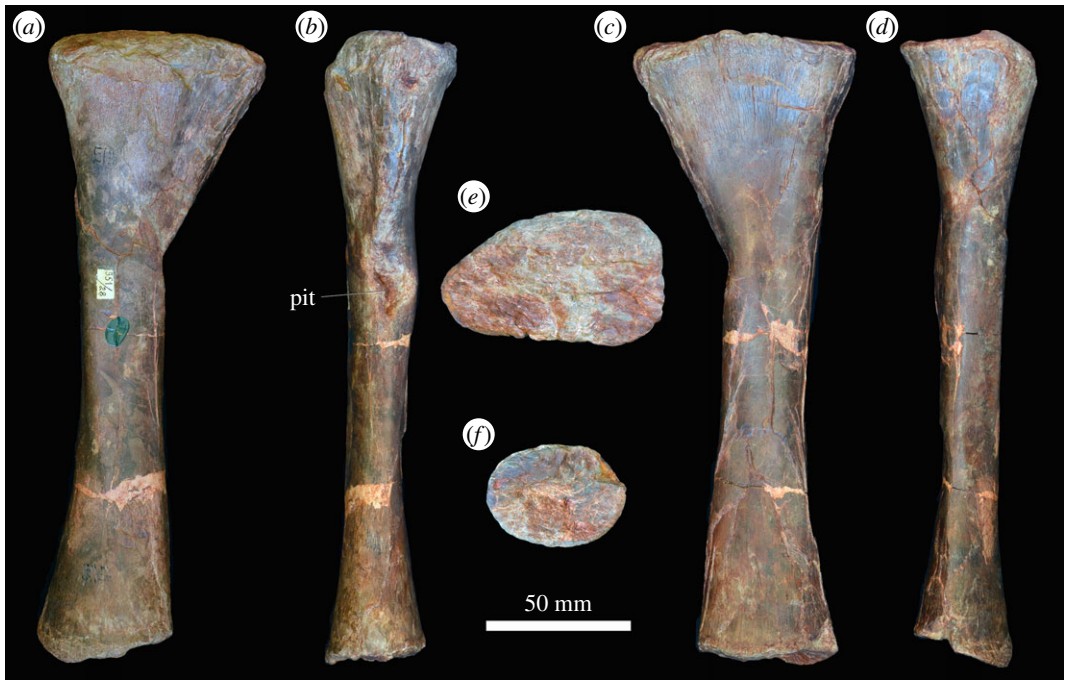

**Figure 20.** PIN 951/43-1 right tibia in (*a*) dorsal, (*b*) medial, (*c*) ventral, (*d*) lateral, (*e*) proximal and (*f*) distal views. **pit**, pit for the m. puboischiotibialis. Scale bar equal to 50 mm.

broader than dorsoventrally deep (figure 20*f*), thus contrasting with the subcircular distal surface of *Prolacerta broomi* (BP/1/2676) and '*Chasmatosaurus*' *yauni* (IVPP V2719). Proximally, in dorsal and ventral view, the medial margin of the proximal end is further expanded medially than it is laterally. The proximal end lacks a cnemial crest, as occurs in non-proterochampsid and non-dinosauromorph archosauromorphs [34,50]. The articular surfaces that received the distal end of the femur are not separated from each other on the ventral margin of the bone, as occurs in *Garjainia madiba* (BP/1/5525), *Erythrosuchus africanus* (NHMUK PV R3592), *Shansisuchus shansisuchus* [38], *Dorosuchus neoetus* (PIN 1579/61) and *Euparkeria capensis* (SAM-PK-6047B), but contrasting with the notched posterior margin present in *Prolacerta broomi* (BP/1/2676), *Proterosuchus goweri* (NMQR 880) and '*Chasmatosaurus*' *yuani* (IVPP V2719). Immediately ventral to the proximal expansion, there is a very deep, large pit on the medial surface (figure 20*b*). This is identical to the condition in '*Chasmatosaurus*' *yuani* (IVPP V2719) and *Erythrosuchus africanus* (NHMUK PV R3592), and Gower [4] suggested that the pit is the insertion site of the m. puboischiotibialis in the latter species. A ridge extends proximodistally down the shaft on the dorsolateral surface, and the lateral surface of the proximal end is flattened and slightly rugose, probably for muscle attachment. The dorsal surface of the proximal end is slightly rugose, probably for articulation of the fibula, although there is not a distinct facet proximally or distally. The distal articular facet is shallowly concave.

### 3.3.10. Fibula

The fibula is elongate and slender. In dorsal view, it is expanded proximally and distally, and these expansions are approximately equal at each end (figure 21). Both articular surfaces are dorsoventrally compressed ovals in outline with the long axis transverse, resembling the condition in *Proterosuchus goweri* (NMQR 880) and *Shansisuchus shansisuchus* [38]. The dorsal surface of the shaft is transversely convex, and approximately one-third of the way down the shaft there is a rugose muscle scar on the lateral side of the dorsal surface. The scar is oval in outline with the long axis trending proximodistally. This muscle scar is probably for insertion of the m. iliofibularis and is also present in *Erythrosuchus africanus*, although it appears to extend further ventrally in that taxon [4]. In ventral view, the proximal and distal ends are flattened for articulation with the tibia. Laterally, on the ventral surface of the proximal end, there is a rugose, prominent ridge extending distally to the probable m. iliofibularis scar on the dorsal surface. Below this muscle scar, this ridge is drawn out and forms the lateral margin of the bone, and medial to this ridge the ventral surface is shallowly concave.

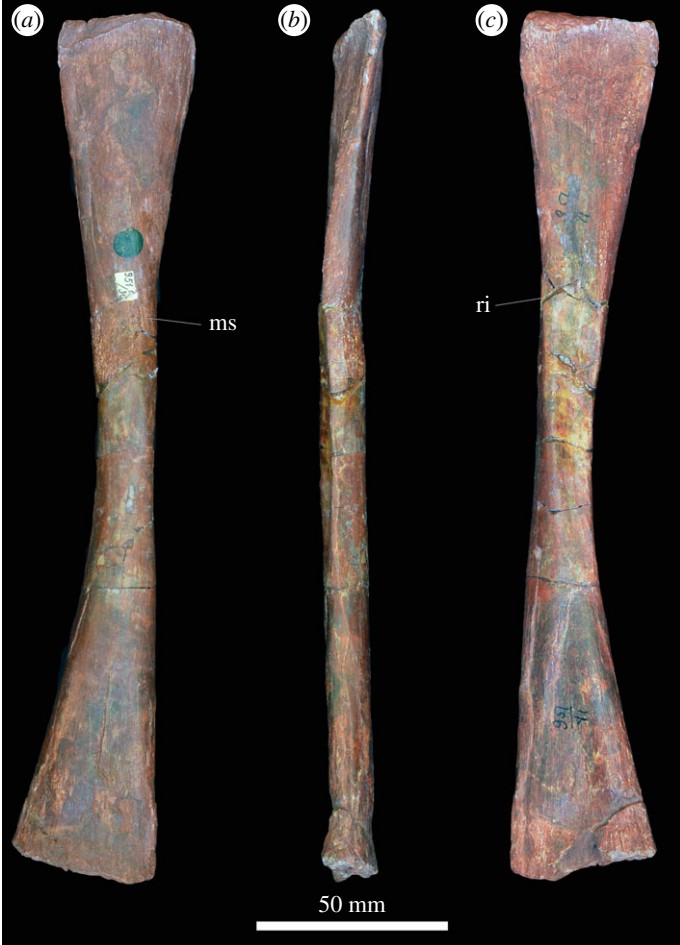

**Figure 21.** PIN 951/126, right fibula in (*a*) dorsal, (*b*) medial and (*c*) ventral views. **ms**, muscle scar; **ri**, ridge. Scale bar equal to 50 mm.

The ridge is also present in *Erythrosuchus africanus* [4]. In ventral view, there is a second, more rounded ridge that extends distolaterally and tapers immediately before the shaft begins to expand transversely for the distal end. This ridge bounds the shallowly concave surface on the medial side. The medial surface of the shaft is flattened.

### 3.3.11. Tarsus

The calcaneum (figure 22) and distal tarsal 4 of *Garjainia prima* were described in detail by Gower [23] ('*Vjushkovia triploicostata*' therein). They are not redescribed, but the calcaneum is illustrated here with detailed photographs for the first time.

### 3.3.12. Pes

*Metatarsals* (figure 23). Nine metatarsals are present, representing multiple individuals. The majority of the metatarsals are from right pedes, although PIN 951/87 and PIN 951/112 are interpreted to be from a left pes. PIN 951/87 is heavily crushed and eroded at both proximal and distal ends, but it is potentially metatarsal II, III or IV. Consequently, the following description is based on PIN 951/112 and the right metatarsals.

The other eight metatarsals are relatively well preserved with some partial reconstruction, although identification of material from each digit is tentative given the high number of individuals fossilized at the site. PIN 951/112 and PIN 951/110 are probably metatarsal I, because they lack the robust and medially flared hook-shape that characterizes the proximal end of metatarsal V of other archosauromorphs [23,34,50,59,60]. PIN 951/111, PIN 951/109 and PIN 951/88 are probably metatarsal II, and PIN 951/85 and PIN 951/84 could be from either metatarsal III or IV.

5000

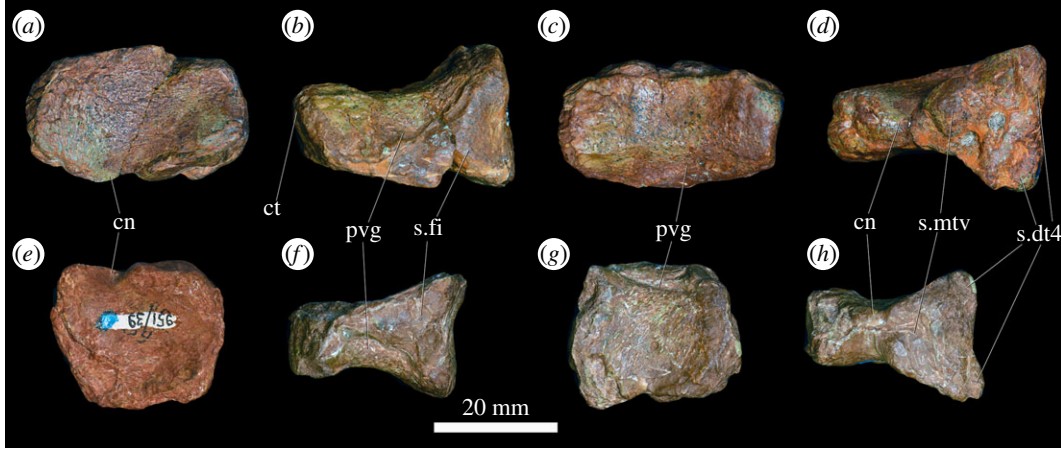

**Figure 22.** (*a–d*) PIN 951/81 right calcaneum; (*e–h*) PIN 951/39, left calcaneum in (*a,e*) dorsal, (*b,f*) proximal, (*c,g*) ventral and (*d,h*) distal views. The same elements in the same views were illustrated via drawings in Gower [23]. **cn**, calcaneal notch; **ct**, calcaneal tuber; **pvg**, proximoventral groove; **s.dt4**, articular surface for digit 4; **s.fi**, articular surface for fibula; **s.mtv**, articular surface for metatarsal V. Scale bar equal to 20 mm.

Metatarsal I (figure 23*a–d*) is stocky with a transversely broad and anteroposteriorly flattened midshaft relative to the other metatarsals. The proximal and distal ends are transversely expanded, and the distal end is more convex than the proximal end. This bone closely resembles the elements identified as metatarsal I in *Shansisuchus shansisuchus* [38].

Metatarsal II (figure 23*e–i*) is proximodistally elongate with transversely expanded proximal and distal ends and a transversely constricted midshaft. The proximal end is more transversely expanded than the distal end. The proximal end has a flat dorsal surface and a convex ventral surface. The proximal surface is flattened and rugose. The distal end has a convex dorsal surface, a gently concave ventral surface and a gently convex distal surface for articulation with the proximal phalanx. The dorsal surface of the proximal and distal ends has a slight proximodistally trending ridge that merges with the shaft a quarter of the way along the total proximodistal length of the bone. The ventral surface of the proximal and distal ends of the shaft is transversely concave forming a subtriangular depression. The dorsal surface of the distal end of the shaft is transversely convex medially, but transversely concave laterally. In lateral view, the dorsal surface of the shaft is straight and the ventral surface is concave proximodistally with mainly ventrally expanded proximal and distal ends, resembling the condition in *Erythrosuchus africanus* (BP/1/2096). The proximal and distal ends of the shaft are slightly rotated relative to each other such that the toe was probably slightly externally orientated during flexion, as occurs in *Erythrosuchus africanus* (BP/1/2096). On the lateral surface of the distal end, there is a subcircular ligament pit. The distal articular surface of the bone has a subrectangular profile, with a concave ventral margin. The distal end of the metatarsal II of *Erythrosuchus africanus* (BP/1/2096) is also subrectangular, but proportionally narrower than in *Garjainia prima*.

Metatarsal III or IV (figure 23*k–p*) is more elongate than metatarsal II and has a less transversely expanded proximal end, resembling the condition between the metatarsals II and IV of *Erythrosuchus africanus* (BP/1/2096). By contrast, metatarsal III is shorter than metatarsal II in this South African specimen. There is a proximal extension of the lateral surface of the proximal end, which creates a transversely concave proximal surface, as is the case in the metatarsal III, but not the metatarsal IV, of *Erythrosuchus africanus* (BP/1/2096). On the lateral surface of the shaft of PIN 951/85, at approximately mid-length, there is a slight tubercle, which could be either a taphonomic or pathologic artefact. On the lateral side of the ventral surface of the proximal end of the shaft, there is a proximodistally trending ridge. The distal end is more transversely rotated than in metatarsal II, resembling the metatarsals III and IV of *Erythrosuchus africanus* (BP/1/2096). The ligament pits on the medial surface of the distal end are relatively slightly deeper than in metatarsal II, as is the transversely concave articular surface of the proximal end. At the distal end of the dorsal surface, there is a greater anteroposterior expansion of the lateral side of the shaft, creating a subcircular rather than subtriangular transverse depression. The subtrapezoidal distal articular surface of the bone is proportionally broader than those of the metatarsals III and IV of *Erythrosuchus africanus* (BP/1/2096).

*Phalanges* (figure 24). Twenty-three phalanges are present. These phalanges resemble those of the foot of *Erythrosuchus africanus* (BP/1/2096), but we cannot rule out that they do not belong to the manus

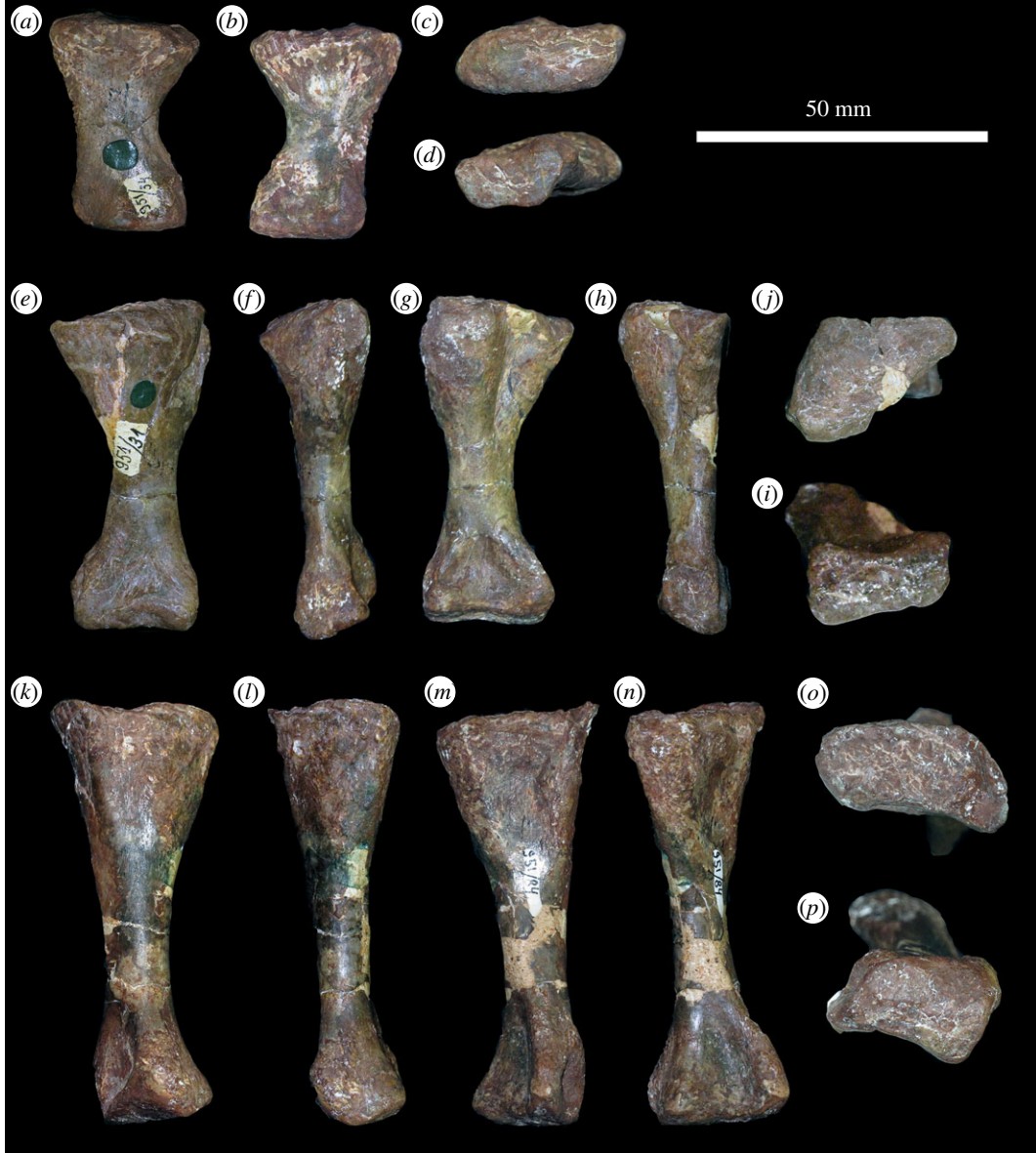

**Figure 23.** Right metatarsals I–III. (*a–d*), PIN 951/110, metatarsal I; (*e–j*), PIN 951/111, metatarsal II; (*k–p*), PIN 951/84, metatarsal III or IV. (*a,e,k*), dorsal view; (*b,g,m*), ventral view; (*c,j,o*), proximal view; (*d,i,p*), distal view; (*f,l*) medial view; (*h,n*), lateral view. Scale bar equal to 50 mm.

because the bones of the anterior autopodium are poorly known in erythrosuchid specimens [38]. Similarly, the attribution of individual phalanges to either a left or right autopodium, or to the position within it, is not easily determined given the extensive size and taphonomic variation of the elements. It is, however, possible to identify proximal, distal and ungual phalanges.

The proximal phalanges, best exemplified by PIN 951/113 (figure 24*a,b*), are stout with proximal and distal ends that are expanded both in a transverse and a flexor-plantar orientation. The proximal articular surface is subcircular to suboval in proximal view and concave transversely and in a flexor-plantar orientation. The distal articular surface is reniform in the distal view and convex. The lateral sides of the distal end have shallow collateral ligament depressions. The shaft is subcircular in cross-section and in lateral view the phalanges appear subtriangular, tapering distally. On the plantar surface, there are concavities on both the proximal and distal ends of the shaft. The overall morphology of these proximal phalanges resembles those of the pes of *Erythrosuchus africanus* [23] and *Shansisuchus shansisuchus* [38].

The distal phalanges, best exemplified by PIN 951/98 (figure 24*c,d*), are less elongate and have less transversely expanded proximal and distal ends, although the expansion of the proximal and distal ends

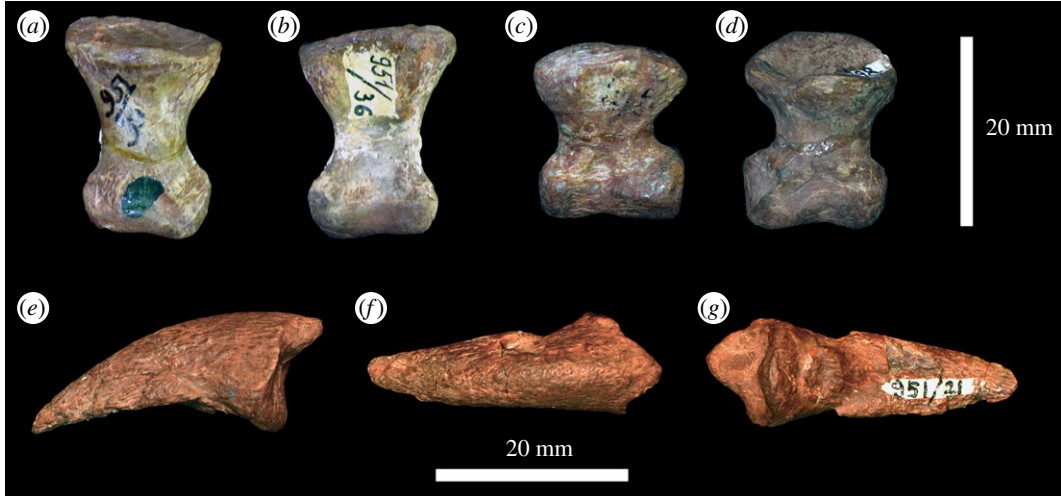

**Figure 24.** Phalanges. (*a,b*) PIN 951/113, proximal phalanx, in (*a*) flexor and (*b*) plantar view. (*c,d*) PIN 951/98, distal phalanx, in (*c*) flexor and (*d*) plantar view. Proximal ends of (*a–d*) towards the top of the figure. (*e–g*) PIN 951/21 ungual phalanx in (*e*) lateral, (*f*) flexor and (*g*) plantar view. Scale bars equal to 20 mm.

is still prominent. The distal articular surface is saddle-shaped with an anteroposteriorly trending groove dividing it into distinct ginglymi, as occurs in the foot of *Erythrosuchus africanus* (BP/1/2096).

Four well-preserved ungual phalanges are present. They are claw-shaped and of varying lengths. The articular surface is concave for articulation with the distal end of the preceding phalanx. The claw is slightly curved ventrally with a convex flexor surface and a flat to concave plantar surface. The surface of the ungual is gently rugose throughout.

## 3.4. Association of material

Some of the bone bed material of the referred specimens of *Garjainia prima* clearly articulates and, therefore, belongs to the same individual; other associations can be made based on size.

(1) An articulating series of vertebrae, 23 presacrals, two sacrals and four caudals (PIN 951/64).
(2) A posterior dorsal (PIN 951/35-23) and two sacral vertebrae (PIN 951/37-1, 2) based on articulation.
(3) The two largest scapulae, PIN 951/4-2 and PIN 951/2-2-1 appear to be a pair based on size and appearance, and the largest coracoid, PIN 951/2-2-2, articulates with the right element. Based on relative size, the largest humerus, PIN 951/36-1, and the isolated ulna, PIN 951/38-1, may also belong to this individual. An articulating set of ilia, sacrum, pubes and ischia (PIN 951/8-1, PIN 951/1-1, PIN 951/5-1, 2) were previously mounted for display and appear to belong to a single individual; the large femur (PIN 951/61-1) may also be from this individual based on size. The femur (PIN 951/61-1) and the largest humerus (PIN 951/36-1) are proportionately a similar size as those of *Erythrosuchus africanus* [4], and thus all of this material may pertain to a single large individual. Herein we make the assumption that this large postcrania belongs to a single individual based on proportional similarity with *Erythrosuchus africanus*, and base our body mass estimates on these elements (see below).
(4) A pair of fused scapulocoracoids, one with an associated clavicle (PIN 951/4-1-1 and 4-1-2), is from the same individual as the large right humerus (PIN 951/36-2) based on size and preservational appearance.
(5) A pair of medium-sized scapulae (PIN 951/7-1-2 and PIN 951/7-1-1) with articulating but unfused coracoids (PIN 951/2-1-1 and PIN 951/2-1-2) is probably from the same individual as a pair of humeri (PIN 951/36-4 and PIN 951/36-3) based on size and preservation. Based on size alone, these may be from the same individual as the previously mounted ulna and radius (PIN 951/38-2 and PIN 951/132).
(6) A very small partial coracoid (PIN 951/4-4), scapula (PIN 951/7-1-2) and humerus (PIN 951/36-2) are from the same individual based on size and preservation.
(7) A pair of ilia (PIN 951/8-2) articulates well with a left pubis (PIN 951/5-3) and a pair of ischia (PIN 951/1-2) and is clearly associated (electronic supplementary material figure S3).
(8) Articulating pelvic elements from both sides numbered sequentially PIN 951/48–53 are clearly from the same individual and, based on preservational appearance, the smaller femur (PIN 951/61-2) might also belong to this individual.

## 3.5. Phylogenetic analysis

### 3.5.1. Methods

We checked and updated the character scorings of *Garjainia prima* following the new information provided by [6] and this paper, as well as incorporated revised scorings for *Chalishevia cothurnata* and *Shansisuchus shansisuchus* following [11], in the most extensive phylogenetic dataset currently available for Permian and Triassic archosauromorphs ([34], as modified by subsequent authors). We worked on the latest version of this data matrix, which was recently published by Scheyer *et al.* [61] to test the phylogenetic position of *Colobops noviportensis*. In particular, the scorings of the lower jaw and postcranium of *Chalishevia cothurnata* were changed to missing data because the elements belonging to these regions have been recently reinterpreted as belonging to an indeterminate erythrosuchid rather than to the hypodigm of the species [11]. We modified the character 393 and changed some scorings accordingly. We deactivated 35 terminals before the tree searches, which were included by [3] only for the purpose of morphological disparity analyses, and also removed character 119 (following [62] and [10]), resulting in a dataset of 121 active terminals and 710 characters. The matrix is available in the electronic supplementary material.

The matrix of discrete morphological characters was analysed under equally weighted maximum-parsimony using TNT v. 1.5 [63,64]. The search strategies started using a combination of the tree-search algorithms sectorial searches, drifting, ratchet and tree fusing, until 100 hits of the same minimum tree length were achieved. The best trees obtained were subjected to a final round of tree bisection-reconnection (TBR) branch swapping. Zero-length branches in any of the recovered most parsimonious trees were collapsed. The following characters were considered additive: 1, 2, 7, 10, 17, 19–21, 28, 29, 36, 40, 42, 50, 54, 66, 71, 74–76, 122, 127, 146, 153, 156, 157, 171, 176, 177, 187, 202, 221, 227, 263, 266, 278, 279, 283, 324, 327, 331, 337, 345, 351, 352, 354, 361, 365, 370, 377, 379, 386, 387, 398, 410, 424, 430, 435, 446, 448, 454, 458, 460, 463, 470, 472, 478, 482, 483, 485, 489, 490, 504, 510, 516, 529, 537, 546, 552, 556, 557, 567, 569, 571, 574, 581, 582, 588, 648, 652 and 662, because they represent nested sets of homologies. Branch support was quantified using decay indices (Bremer support values) and a bootstrap resampling analysis, using 1000 pseudo-replicates and reporting both absolute and GC ('Group present/Contradicted'; i.e. the difference between the frequencies of recovery in pseudo-replicates of the clade in question and the most frequently recovered contradictory clade) frequencies [65].

### 3.5.2. Results

The analysis found 27 most parsimonious trees (MPTs) of 3868 steps and a consistency index of 0.23940 and a retention index of 0.62258. The topology of the strict consensus tree is identical to that of the analysis of ([61], fig. S3) and, in particular, Erythrosuchidae has the same interrelationships as in most recent analyses [9,11]. The Chinese species *Guchengosuchus shiguaiensis* and *Fugusuchus hejiapanensis* are recovered as successive sister taxa to a clade composed of (*Garjainia* (*Erythrosuchus africanus* (*Shansisuchus shansisuchus*+*Chalishevia cothurnata*))) (figure 25). However, our modifications of scorings produced some changes in support within Erythrosuchidae. Support values for the most basal branch of Erythrosuchidae and of the clade containing *Garjainia* and *Erythrosuchus* are almost identical to those in ([61], fig. S3). By contrast, the absolute and GC bootstrap frequencies increased from 47%/41% to 57%/ 51% for the clade containing *Guchengosuchus* and *Erythrosuchus* and from 64%/51% to 73%/63% for the clade containing *Erythrosuchus* and *Shansisuchus*. Bremer support for the clade containing *Guchengosuchus* and *Erythrosuchus* increased from 1 to 2. Bremer support values are the same and the absolute bootstrap frequencies are almost identical to those of the analysis of [61] for the genus *Garjainia* and for the *Shansisuchus shansisuchus* + *Chalishevia cothurnata* clade, but GC frequencies slightly increased in these branches (78% to 84% and 31% to 37%, respectively). If the position of *Garjainia prima* is *a priori* forced in an alternative topology (setting *Garjainia madiba* as a floating taxon) and the analysis is rerun, 13 extra steps are needed to recover this species as the sister taxon to *Erythrosuchus africanus* and 14 additional steps to find it as the sister taxon to *Shansisuchus shansisuchus* (figure 25).

## 3.6. Body mass

The body masses of *Garjainia prima* and *Erythrosuchus africanus* (based on NHMUK PV R3592) were estimated using the bivariate regression equations of [66] implemented in R v. 3.6.0 using the package

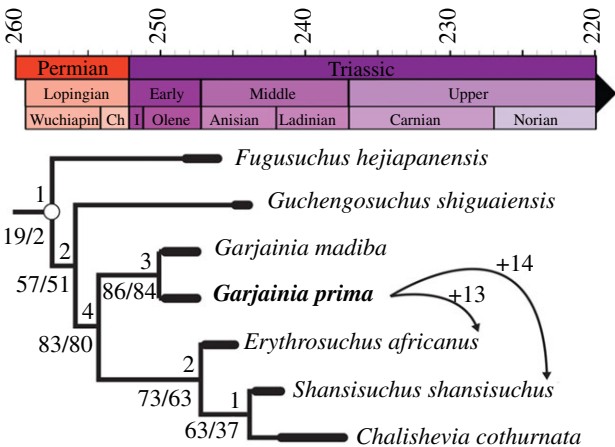

**Figure 25.** Erythrosuchid interrelationships in the strict consensus tree. Numbers above nodes are decay indices (Bremer support values), those below are absolute and GC bootstrap frequencies. +13 and +14 indicate the number of additional steps required to recover *Garjainia prima* in the specified positions. The non-erythrosuchid part of the tree has been omitted. The thick black lines represent chronostratigraphic uncertainty in the age of the species rather than actual temporal ranges. **Ch**, Changhsingian; **I**, Induan; **Olene**, Olenekian; **Wuchiapin**, Wuchiapingian.

*MASSTIMATE* [67,68]. For *G. prima,* humeral and femoral circumferences were taken from the potentially associated PIN 951/36-1 (107 mm) and PIN 951/61-1 (106 mm), respectively (see electronic supplementary material for all measurements), while for *E. africanus*, NHMUK PV R3592 is a partial skeleton in which the humerus circumference is 229 mm and the femur circumference is 198 mm. Body mass was calculated using both the raw data and phylogenetically corrected equations of [66]. For *G. prima,* the raw data gave a body mass of 198 kg, with lower and upper estimates of 147 and 248 kg respectively, while the phylogenetically corrected body mass was 207 kg with lower and upper estimates of 155 and 258 kg. The *E. africanus* individual appears to have been a much larger animal, with a body mass of 1.3 tonnes (lower estimate = 996 kg; upper estimate = 1683 kg) based on the raw data and 1.4 tonnes (lower estimate = 1052 kg; upper estimate = 1754 kg) based on the phylogenetically corrected equation.

Because we cannot be certain about the association of the femur and humerus of *Garjainia prima*, we also calculated body mass on humerus circumference only and femur circumference only for both *G. prima* and *Erythrosuchus africanus* using the equations given in ([66], table 4). This resulted in body mass estimates of 193.5 and 204.4 kg for *G. prima* for the humerus and femur, respectively, and 1452.2 and 1171.0 kg for *E. africanus* for the humerus and femur, respectively. These estimates are within the upper and lower confidence intervals when the two elements are considered together for both taxa.

## 3.7. Reconstruction of locomotor musculature

Proximal fore- and hind-limb musculature reconstruction is possible because of the excellent state of preservation of many of the appendicular elements with numerous clear muscle scars (figure 26). The extant phylogenetic bracket (EPB) of Erythrosuchidae comprises extant archosaurs (crocodylians and birds) and extant lepidosaurs (snakes, lizards, *Sphenodon*). The wide variety of body forms and modes of life displayed by these bracket taxa make inferring muscle homologies complicated. *Garjainia prima* is more closely related to extant archosaurs than to lepidosaurs, and its limb anatomy is more similar to crocodylians than birds. Thus, we base our muscle reconstructions primarily on the locomotor muscles of extant crocodylians. Data on crocodylian muscles is drawn from [51] and references therein, and we draw heavily on the work of Hutchinson [56,58] for the evolution of pelvic and hind-limb musculature in archosauromorphs. The joint surfaces of the limb elements of *G. prima* are concave and rugose in all specimens, and femoral and humeral articular surfaces are not distinct. This suggests that substantial unpreserved soft tissues were present around limb bone epiphyses, and this hinders our understanding of shoulder and hip joint anatomy [69]. In particular, it means that reconstruction of the angle at which the limbs were held is highly speculative. The femur of *G. prima* superficially resembles that of extant lepidosaurs more than it does of extant archosaurs, in being straight with little torsion of the shaft, and in having a prominent internal trochanter. We interpret this as an indication that the hind limb of *G. prima* was typically less adducted than in archosaurs.

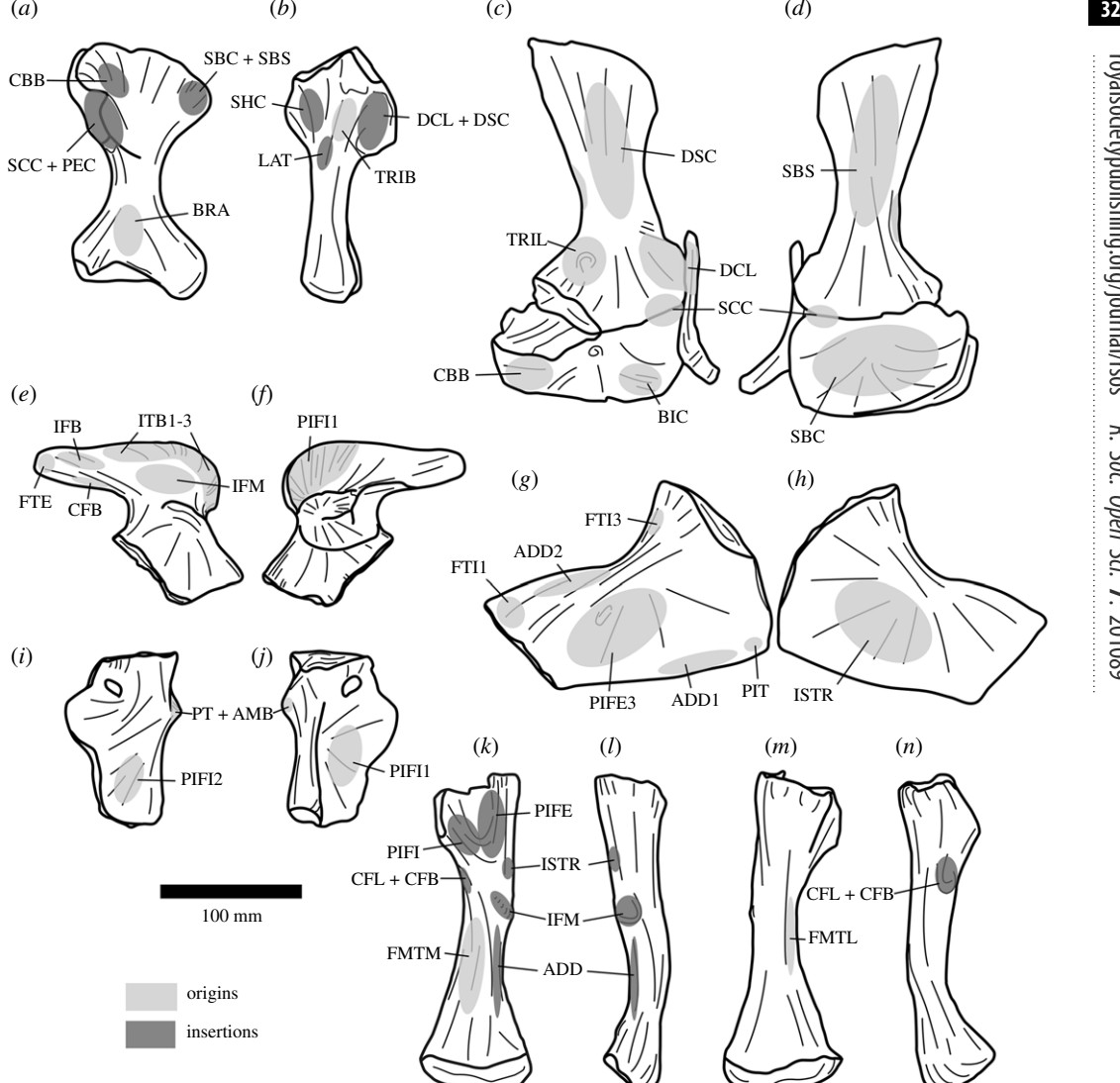

**Figure 26.** Reconstruction of proximal locomotor musculature in *Garjainia*. (*a,b*) Right humerus in (*a*) ventral and (*b*) lateral view. (*c,d*) Right scapulocoracoid in (*c*) lateral and (*d*) medial view. (*e,f*) Right ilium in (*e*) lateral and (*f*) medial view. (*g,h*) Right ischium in (*g*) lateral and (*h*) medial view. (*i,j*) Left pubis in (*i*) anterior and (*j*) posterior view. (*k–n*) Right femur in (*k*) ventral, (*l*) lateral, (*m*) dorsal and (*n*) medial view. All elements are scaled to the size of the large individual (see Association of material, above). **ADD**, adductor; **AMB**, ambiens; **BIC**, biceps; **BRA**, brachialis; **CBB**, coracobrachialis brevis; **CFB**, caudofemoralis brevis; **CFL**, caudofemoralis longus; **DCL**, deltoideus clavicularis; **DSC**, deltoideus scapularis; **FMTL**, femorotibialis lateralis; **FMTM**, femorotibialis medialis; **FTE**, flexor tibialis externus; **FTI**, flexor tibialis internus; **IFB**, iliofibularis; **IFM**, iliofemoralis; **ISTR**, ischiotrochantericus; **ITB**, iliotibialis; **LAT**, latissimus dorsi; **PEC**, pectoralis; **PIFE**, puboischiofemoralis externus; **PIFI**, puboischiofemoralis internus; **PT**, pubotibialis; **SBC**, subcoracoideus; **SBS**, subscapularis; **SCC**, supracoracoideus; **SHC,** scapulohumeralis caudalis; **TRIB**, triceps brevis; **TRIL**, triceps longus. Scale bar equal to 100 mm.

# 4. Discussion

## 4.1. Phylogenetic implications

The revision of the postcranial anatomy of the well-preserved referred specimens of *Garjainia prima* allowed us to recognize and rescore character states in this species that were thought to be restricted to non-erythrosuchid, non-eucrocopodan archosauromorphs. The presence of mammillary processes on neural spines was optimized as a synapomorphy of the clade composed of *Boreopricea funerea* and more crownward archosauromorphs, and their subsequent loss as a synapomorphy of archosauriforms to the exclusion of proterosuchids [34]. The presence of such structures in *G. prima* is

optimized here as an independent acquisition in this species and current evidence indicates that they represent an autapomorphy. The optimization of a bifurcated distal end of the second sacral rib is ambiguous for the common ancestor of Archosauromorpha because of the presence of moderately high homoplasy in its absence/presence among the earliest members of the clade [34]. The presence of a bifurcated sacral rib 2 is unambiguously optimized in the clade composed of Rhynchosauria and more crownward archosauromorphs and its absence is recovered as a synapomorphy of the Erythrosuchidae + Eucrocopoda clade. The presence of a posterolateral process in a bifurcated sacral rib 2 in *G. prima* is optimized as an autapomorphy of the species.

The absence of topological changes between our analysis and those of other recent authors (e.g. [9,11,34]) indicates that the interrelationships of deeply nested erythrosuchids (the clade containing *Garjainia* and *Erythrosuchus*) are relatively stable despite the presence of the very poorly known taxon *Chalishevia cothurnata*. Under the topologically constrained searches, more additional steps are needed to force the position of *G. prima* as the sister taxon to *Erythrosuchus africanus* and *Shansisuchus shansisuchus* than those needed in the data matrix analysed by Ezcurra *et al.* [9]. As a consequence, branch support and topologically constrained searches lead to the interpretation that our results recovered better supported erythrosuchid interrelationships than in analyses of former versions of this dataset [9,11,34,61].

## 4.2. Osteological and myological femoral evolution in archosaurs

There is some debate about whether the internal trochanter of some non-archosaurian archosauriforms is homologous to the fourth trochanter of dinosaurs and extant crocodilians, or whether the fourth trochanter is a novel structure (see discussions in [4,50,58]). The femoral morphology of *Garjainia prima* and other erythrosuchids casts light on this debate (figures 19 and 26; see Femur, above). In dinosaurs and crocodilians, the fourth trochanter is the site of insertion of the m. caudofemoralis longus and brevis, and it projects posteroventrally from the posteromedial surface of the femoral shaft. In extant lepidosaurs, the m. caudofemoralis longus and brevis insert on the distal end of the internal trochanter and distomedial edge of the interotrochanteric fossa (e.g. [70,71]). The lepidosaur configuration is interpreted as the plesiomorphic condition for diapsids, and it is also probably plesiomorphic for Archosauromorpha given the close similarity of the femora of non-archosauriform archosauromorphs and extant lepidosaurs. Thus, during archosaur evolution, m. caudofemoralis insertion migrated distally from a more proximal location on the internal trochanter and intertrochanteric fossa [58]. The muscle scar distal and somewhat medial to the internal trochanter in *G. prima* is interpreted as an insertion site for m. caudofemoralis longus, making it at least partly homologous in terms of muscle attachment to both the internal trochanter of non-archosaurian diapsids and the fourth trochanter of archosaurs. The condition in *G. prima* (and *G. madiba*: [8]) is of interest because the femur resembles that of some non-archosauriform archosauromorphs and lepidosaurs in having a more proximally positioned internal trochanter than in *Erythrosuchus africanus* [4]. Thus, some of the changes in femoral morphology within Erythrosuchidae appear to be convergent with those that occurred also in more crownward archosauriforms.

## 4.3. Body mass in erythrosuchids

Archosauromorphs arose during the middle–late Permian, but were a minor component of terrestrial ecosystems up until the Permo-Triassic mass extinction (PTME) [3]. In the immediate aftermath of the PTME, archosauromorphs increased in abundance and formed a low disparity but globally distributed disaster fauna, which was rapidly replaced in the late Early Triassic (Olenekian) and early Middle Triassic (Anisian) by the diversification of major archosauromorph clades, which included the tanystropheids, rhynchosaurs, allokotosaurs, archosaurs and erythrosuchids [3,49].

A reduction in the body size of organisms, the 'Lilliput effect', is a well-known feature of many mass extinctions [72]. Reduction in size of numerous organisms across the Permo-Triassic boundary has been observed (e.g. [73,74]), including in terrestrial vertebrates [75]. However, some studies have failed to find driven trends in body size evolution across the Permo-Triassic boundary and instead attribute the Lilliput effect after the PTME to changes in primary productivity [74] or the removal of the largest-bodied organisms [75]. Interestingly, archosauriforms appear not to have undergone a size decrease across the Permo-Triassic boundary [76], maintaining similar body sizes from the late Permian into the Early Triassic. Archosauromorph body size then increased around the Olenekian to Anisian transition [77,78] and, unusually, from the Anisian to the Carnian (Middle to early Late Triassic), in which carnivore maximum body size was greater than that of herbivores [77].

The erythrosuchid archosaurs evolved in this post-extinction world. Our mass estimates for individuals of *Garjainia prima* and *Erythrosuchus africanus* are the first to be published for any early archosauromorph. *Garjainia prima*, dated to the late Olenekian, attained a body mass approximately equivalent to a male lion; whereas *Erythrosuchus africanus*, which is known from the slightly younger subzone B of the *Cynognathus* Assemblage Zone (early Anisian; e.g. [79]), attained a much greater body size, with a body mass between 1 and 1.6 tonnes, larger than the largest of today's terrestrial carnivores, and the range matches the estimated body masses of some Jurassic and Cretaceous predatory dinosaurs, such as *Neovenator salerii* (ca 1 tonne), *Megalosaurus bucklandii* (ca 1.4 tonnes) and *Carnotaurus sastrei* and *Majungasaurus crenatissimus* (ca 1.6 tonnes; [80]). Mass estimates are not currently possible for other Middle Triassic erythrosuchids, such as *Chalishevia cothurnata* [11] and *Shansisuchus shansisuchus* [46], but they were also large-bodied (e.g. skull length of 80 cm in *Chalishevia*: [11]). The large body sizes of *Garjainia prima*, *Erythrosuchus africanus* and other Middle Triassic erythrosuchids may have been attained as part of the trend in increasing body size after the PTME, and they radiated to dominate the hypercarnivorous niche of their ecosystems, being the largest predators of the Early and early Middle Triassic.

Data accessibility. All data associated with this work is included in electronic supplementary material.

Authors' contributions. R.J.B. and A.G.S. designed the study. All authors contributed to data collection. S.C.R.M., A.G.S., M.D.E., B.P.H. and R.J.B. analysed the data. All authors contributed to the manuscript.

Competing interests. We declare we have no competing interests.

Funding. Travel to the PIN for R.J.B., S.C.R.M., L.E.M. and E.M.D. was funded by a Royal Society International Exchange grant (no. IEC\R2\170064) to R.J.B. and A.G.S., and an earlier visit for R.J.B., M.D.E. and D.J.G. was funded by the DFG Emmy Noether Programme (BU 2587/3-1 to R.J.B.). A.G.S. was co-funded by awards from the Royal Society International Exchange grant (no. IEC\R2\170064), and the Russian Foundation for Basic Research (RFBR no. 17-54-10013) during the course of this work. A.G.S. and D.I.P. were funded by the Russian Foundation for Basic Research through the research project no. 20-04-00070. A.G.S. was also funded by a subsidy of the Russian Government to support the Program of 'Competitive Growth of Kazan Federal University among World's Leading Academic Centers'. Participation of B.P.H. in this work was funded by the European Union's Horizon 2020 research and innovation programme 2014–2018 under grant agreement no. 677774 (European Research Council [ERC] Starting Grant: TEMPO awarded to Roger Benson [University of Oxford]). T.J.R. was funded by a University of Brighton Science Studentship. M.D.E. is funded by Agencia Nacional de Promoción Científica y Técnica (PICT 2018-01186).

Acknowledgements. D.J.G. thanks Mike Benton (University of Bristol) for supervising his NERC-funded research on erythrosuchids during which he first examined *Garjainia prima*.

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
