## [Reviewer comments · Royal Society Open Science]

Review History

RSOS-201089.R0 (Original submission)

Review form: Reviewer 1 (William Parker)

Is the manuscript scientifically sound in its present form?

Yes

Are the interpretations and conclusions justified by the results?

Yes

Is the language acceptable?

Yes

Do you have any ethical concerns with this paper?

No

Have you any concerns about statistical analyses in this paper?

No

Recommendation?

Accept with minor revision (please list in comments)

Comments to the Author(s)

This is a well written thorough description and discussion of an important set of specimens. The material is absolutely gorgeous. The authors provide numerous comparisons with other taxa and cite the specimen numbers making it clear where the comparisons were made. I only had a few comments/suggestions that are in the attached files (Appendices A & B). One concern is the identification of the processes on the apex of the neural spine as mammillary processes. My understanding is that in mammals these are on the neural arch and transverse processes. How are these homologous? Overall though I find the manuscript to be a solid contribution.

Review form: Reviewer 2 (Jun Liu)**Is the manuscript scientifically sound in its present form?**

Yes

Are the interpretations and conclusions justified by the results?

Yes

Is the language acceptable?

Yes

Do you have any ethical concerns with this paper?

No

Have you any concerns about statistical analyses in this paper?

No

Recommendation?

Accept with minor revision (please list in comments)

Comments to the Author(s)

This work is a description of the postcranial skeleton of *G. prima*. It provides detailed anatomical information and gives a reconstruction on musculature. It is well written and I nearly have nothing to say.

Line 54 add the original publication (Cheng, 1980) for *Fugusuchus*

Line 1261 Fig25 the age of *Shansisuchus* may be limited to Anisian, not Ladinian based on recent zircon dating (Liu et al., 2018)

I suggest the Reconstruction of locomotor musculature could be expanded and give more elaboration.

Review form: Reviewer 3**Is the manuscript scientifically sound in its present form?**

Yes

Are the interpretations and conclusions justified by the results?

Yes

Is the language acceptable?

Yes

Do you have any ethical concerns with this paper?

No

Have you any concerns about statistical analyses in this paper?

No

Recommendation?

Accept as is

Comments to the Author(s)

The main contribution of the paper is the descriptive part which makes the paper worthy of publication.

Recovery after the Permo-Triassic extinction event is an interesting topic. In South Africa *Proterosuchus* is the first archosauromorph to appear and in later zones *Garjainia* and *Erythrosuchus*. In the light of this the Russian archosauromorphs from around the Permo-Triassic boundary is very important to decide whether erythrosuchids are derived proterosuchids or the other way round. This would be a real challenge for phylogenetic analysis in a more inclusive paper.

This paper makes a major contribution in terms of the detailed description of skeletal elements. I view some of the interwoven suggestions such as that *Euparkeria* show miniature erythrosuchid features as speculative.

Anatomical evolution is to be studied in a comprehensive series of related forms. The section about the femoral evolution is commendable but lacks some more taxonomic way points for future studies.

The calculations of body weight are interesting but invoking the lion analogy in stead of a crocodile one for *Garjainia* is open to debate.

Decision letter (RSOS-201089.R0)

Dear Dr Maidment

On behalf of the Editors, we are pleased to inform you that your Manuscript RSOS-201089 "The postcranial skeleton of the erythrosuchid archosauriform *Garjainia prima* from the Early Triassic of European Russia" has been accepted for publication in Royal Society Open Science subject to minor revision in accordance with the referees' reports. Please find the referees' comments along with any feedback from the Editors below my signature.

Please submit your revised manuscript and required files (see below) no later than 7 days from today's (ie 13-Oct-2020) date. Note: the ScholarOne system will 'lock' if submission of the revision is attempted 7 or more days after the deadline. If you do not think you will be able to meet this deadline please contact the editorial office immediately.

on behalf of Kevin Padian (Subject Editor)
openscience@royalsociety.org

Reviewer comments to Author:
Reviewer: 1

Comments to the Author(s)

This is a well written thorough description and discussion of an important set of specimens. The material is absolutely gorgeous. The authors provide numerous comparisons with other taxa and cite the specimen numbers making it clear where the comparisons were made. I only had a few comments/suggestions that are in the attached file. One concern is the identification of the processes on the apex of the neural spine as mammillary processes. My understanding is that in mammals these are on the neural arch and transverse processes. How are these homologous? Overall though I find the manuscript to be a solid contribution.

Reviewer: 2

Comments to the Author(s)

This work is a description of the postcranial skeleton of *G. prima*. It provides detailed anatomical information and gives a reconstruction on musculature. It is well written and I nearly have nothing to say.

Line 54 add the original publication (Cheng, 1980) for *Fugusuchus*

Line 1261 Fig25 the age of *Shansisuchus* may be limited to Anisian, no Ladinian based on recent zircon dating (Liu et al., 2018)

I suggest the Reconstruction of locomotor musculature could be expanded and give more elaboration.

Reviewer: 3

Comments to the Author(s)

The main contribution of the paper is the descriptive part which makes the paper worthy of publication.

Recovery after the Permo-Triassic extinction event is an interesting topic. In South Africa *Proterosuchus* is the first archosauromorph to appear and in later zones *Garjainia* and

Erythrosuchus. In the light of this the Russian archosauromorphs from around the Permo-Triassic boundary is very important to decide whether erythrosuchids are derived proterosuchids or the other way round. This would be a real challenge for phylogenetic analysis in a more inclusive paper.

This paper makes a major contribution in terms of the detailed description of skeletal elements. I view some of the interwoven suggestions such as that *Euparkeria* show miniature erythrosuchid features as speculative.

Anatomical evolution is to be studied in a comprehensive series of related forms. The section about the femoral evolution is commendable but lacks some more taxonomic way points for future studies.

The calculations of body weight are interesting but invoking the lion analogy in stead of a crocodile one for *Garjainia* is open to debate.

===PREPARING YOUR MANUSCRIPT===

===PREPARING YOUR REVISION IN SCHOLARONE===

Author's Response to Decision Letter for (RSOS-201089.R0)

See Appendix C.

Decision letter (RSOS-201089.R1)

Dear Dr Maidment,

It is a pleasure to accept your manuscript entitled "The postcranial skeleton of the erythrosuchid archosauriform *Garjainia prima* from the Early Triassic of European Russia" in its current form for publication in Royal Society Open Science.

on behalf of Kevin Padian (Subject Editor)
openscience@royalsociety.org

Appendix A

The postcranial skeleton of the erythrosuchid archosauriform *Garjainia prima* from the Early Triassic of European Russia

Online Supplementary Material

Susannah C. R. Maidment^{1,2*}, Andrey G. Sennikov^{3,4}, Martín D. Ezcurra^{2,5}, Emma M. Dunne², David J. Gower¹, Brandon Hedrick⁶, Luke E. Meade², Thomas J. Raven^{1,7}, Dmitriy I. Paschchenko³, Richard J. Butler²

1. The Natural History Museum, Cromwell Road, London SW7 5BD, United Kingdom
2. School of Geography, Earth and Environmental Sciences, University of Birmingham, Edgbaston, Birmingham, B15 2TT, United Kingdom
3. Borissiak Paleontological Institute RAS, Profsoyuznaya Street 123, Moscow 117647, Russia
4. Institute of Geology and Petroleum Technologies, Kazan Federal University, Kremlyovskaya Street 4, Kazan 420008, Russia
5. Sección Paleontología de Vertebrados, CONICET—Museo Argentino de Ciencias Naturales “Bernardino Rivadavia”, Ángel Gallardo 470 (C1405DJR), Buenos Aires, Argentina
6. Department of Cell Biology and Anatomy, School of Medicine, Louisiana State University Health Sciences Center, New Orleans, LA 70112, USA.
7. School of Environment and Technology, University of Brighton, Lewes Road, Brighton BN2 4GJ, United Kingdom

Axial skeleton

Figure S1. Articulating series of presacral vertebrae, PIN 951/64, in left lateral and dorsal views. Scale bar equal to 50 mm.

The following is a list of referred (non-type) vertebral material of *Garjainia prima* preserved in addition to the articulated series of vertebrae catalogued as PIN 951/64.

PIN 951/65, no sub-number: dorsal centrum, ventral margins reconstructed in plaster, missing neural arch and spine.

PIN 951/65, no sub-number: anterior caudal vertebra with damaged anterior centrum face, caudal ribs broken, missing most of neural arch and neural spine.

PIN 951/65, no sub-number: vertebra from cervico-dorsal transition, c. presacral 8, with three articular facets.

PIN 951/65, no sub-number: vertebra from cervico-dorsal transition with three articular facets, missing much of neural arch, spine, distal part of right transverse process.

PIN 951/65-17: vertebra from mid-dorsal region (c. presacral 16–17), missing neural spine.

PIN 951/65, no sub-number: vertebra from mid-dorsal region (c. presacral 16–17), missing most of neural spine.

PIN 951/65, no sub-number: posterior dorsal vertebra lacking left transverse process.

PIN 951/65, no sub-number: posterior dorsal vertebra lacking most of neural spine and right transverse process.

PIN 951/65, no sub-number: anterior caudal vertebra lacking left caudal rib and distal end of right caudal rib.

PIN 951/65, no sub-number: vertebra from cervico-dorsal transition with three articular facets, almost complete.

PIN 951/65, no sub-number: posterior dorsal vertebra lacking most of neural spine and transverse processes.

PIN 951/65-10: large mid-dorsal vertebra, missing right transverse process.

PIN 951/65-15: posterior dorsal vertebra, missing transverse processes and most of neural spine.

PIN 951/65-24: anterior caudal vertebra, missing caudal ribs and neural spine.

PIN 951/65-12: large mid-dorsal vertebra, almost complete.

PIN 951/65-18: large mid-dorsal vertebra, mostly complete, missing right transverse process.

PIN unnumbered: anterior caudal vertebra missing neural spine, most of neural arch except right prezygapophysis, and caudal ribs.

PIN 951/37: sacral vertebra 1, largely complete but missing left sacral rib.

PIN 951/65-4: anterior cervical vertebra, almost complete.

PIN 951/65-30: caudal vertebra, missing prezygapophyses and caudal ribs.

PIN 951/65-6: anterior cervical vertebra, complete, with cervical ribs glued into articulation.

PIN 951/65-8: anterior to mid-dorsal vertebra, lacking neural spine and right transverse process.

PIN 951/65-9: vertebra from cervico-dorsal transition with three articular facets, almost complete.

PIN 951/65-27: anterior caudal vertebra, lacking neural spine, most of neural arch, caudal ribs.

PIN 952/65-13: anterior dorsal vertebra with three rib articulation facets, probably equivalent to presacral 12. Largely complete.

PIN 952/65-14: mid-dorsal vertebra. Largely complete.

PIN 952/65-21: mid-dorsal vertebra. Largely complete.

PIN 952/65-16: mid-dorsal vertebra, missing neural spine and most of transverse processes.

PIN 951/65-5: anterior cervical vertebra, complete.

PIN 952/65-22: mid-dorsal vertebra. Largely complete.

PIN 952/65-19: posterior dorsal vertebra. Largely complete.

PIN 952/65-20: posterior dorsal vertebra. Largely complete.

Various isolated fragments of neural spines are present.

Figure S2. PIN 951/37, articulating sacral vertebrae 1 and 2 and ilia. **A**, dorsal view; **B**, anterior view. **bif**, bifurcation of distal end of sacral rib 2. Scale bar equal to 50 mm.

The following table lists measurements for the articulated vertebral series PIN 951/64, described in the manuscript.

Vertebra number	Anteroposterior length of the centrum	Dorsoventral height of the anterior face of the centrum	Transverse width of the anterior face of the centrum	Dorsoventral height of the posterior face of the centrum	Transverse width of the posterior face of the centrum	Total height of the vertebra from base of centrum to tip of neural spine	Height of neural spine from the top of the postzygophyses to tip of neural spine
2	36	32	24	24	19	106	65
3	38	32	25	34	23	115	50
4	36	33	24	34	26	118	53
5	38	29	23	38	24	121	53
6	38	36	27	38	24	126	55
7	36	40	24	39	23	131	54
8	32	39	26	40	27	132	55
9	33	33	40	35*	26	132	58
10	36	37	24	39	23	133	59
11	38	37	25	34*	20*	133	63
12	36	34	23	35	21	130	60
13	38	32	23	32	19	126	61
14	37	34	19	31	21	129	62
15	37	32	23	NA	NA	125	60
16	41	NA	NA	27	21	125	62
17	41	28	17	27	20	125	63
18	42	27	20	27	20	124	59
19	38	30	19	27	19	123	61
20	37	27	19	30	21	122	61
21	39	30	15	29	19	122	63

22	32	32*	23*	32*	27	119	66
23	30	31	27	30	29	118	58
24	29	31	28	34*	27	118	59
25	28	31	30	32	30	113	60
36	33	44	25	30	NA	NA
34	23	32	22*	29*	NA	NA
24	29	28	NA	NA	NA	NA
31	29*	20*	26	21	97	47
25	28	26	26*	21*	NA	NA
27	26	16*	26	18*	NA	NA

*Slightly incomplete or damaged

Appendicular skeleton

The following lists all referred (non-type) appendicular material of *Garjainia prima* in the PIN collections and provides measurements of this material.

Coracoids

Two complete scapulocoracoids are similar in size and preservation and likely belong to the same individual. The right is **PIN 951/65-33**, and the left is **951/65-34**. **PIN 951/65-34** also has stickers with two other numbers: the coracoid bears the number **PIN 951/12** and the scapula bears the number **PIN 951/11**. The clavicle is also preserved fused to the dorsal margin of the coracoid and scapula in the right element but not on the left, and this appears to be the only scapulocoracoid in the collection that preserves the clavicle as well. **PIN 951/3** is given to two scapulocoracoids: one is a large left proximal scapula and is associated with (glued to) a partial coracoid, with the coracoid missing its ventral margin, and one is a smaller left coracoid unfused to a scapula. **PIN 951/2** is given to two coracoids that are unfused to scapulae: one is a large right element; the other is a smaller right element which is similar in size and preservational appearance to the smaller element catalogued as **PIN 951/3** and may be from the same individual. A fragment of another very small coracoid is present and unnumbered.

Specimen number	Anteroposterior length (mm)	Dorsoventral height (mm)
951/2 large	139.7	102.9
951/2 small	116.8	78.7
951/3 scap+corac	135.1	Incomplete
951/3 small	Incomplete	80.8
951/65-33	128.7	78.0
951/65-34	Incomplete	79.9

Scapulae

There are seven complete and four partial scapulae in the collection. **PIN 951/3** is a proximal scapula with articulating coracoid. **PIN 951/65-33** and **951/65-34** are a probable pair of articulating scapulocoracoids. The remaining scapulae are not articulated to coracoids (although some may be associated) and comprise: **PIN 951/4**, a large left element, a medium left element, and a small left element, and **PIN 951/7**, comprising a large right element, which based on size and morphology may be from the same individual as the large left element of **PIN 951/4**, and a smaller right element, which based on size, preservation and morphology is probably from the same individual as the medium-sized element from **PIN 951/4**. Based on size alone, the small left and right coracoids catalogued under **PIN 951/2** and **PIN 951/3** may belong to these scapulae. Based on size and preservation, the large scapula of **PIN 951/7** and the large coracoid of **951/2** are likely from the same individual. **PIN 951/70** is a distal end of a scapula from the same locality but collected at a later date (1974) and possibly from a different stratigraphic horizon so that might not be from the same bone bed. **PIN 951/56** is a partial distal end of a scapula of a small specimen. **PIN 951/41** is the glenoid region of a small scapula. Based on these assumptions, seven individuals are represented by scapulocoracoid material in the collection.

Specimen number	Dorsoventral height (mm)	Dorsal end width (mm)	Ventral end width (mm)	Transverse thickness of glenoid (mm)
PIN 951/4 small	160.0	Incomplete	79.0	21.4
PIN 951/4 medium	188.0	Incomplete	97.9	41.3
PIN 951/4 large	225.0	Incomplete	116.7	62.6
PIN 951/7 large	228.0	137.0	111.9	51.7
PIN 951/7 small	190.0	Incomplete	94.5	30.0
PIN 951/65-33	208	Incomplete	103.1	47.7
PIN 951/65-34	207	97.2	109.4	45.8

Humeri

There are six partial or complete humeri in the collections. These are: **PIN 951/36**, the proximal end of a very small right humerus; **PIN 951/62**, a small right humerus; **PIN 951/65-37** which also bears the number **PIN 951/36** and **PIN 951/?56** (the number is unclear); a small left humerus, which based on size may belong to the same individual as **951/65-37** and is numbered as **PIN 951/65-40**. This also bears the numbers **PIN 951/?36** (the number is unclear) and a sticker with the number **PIN 951/13**. A large right humerus also has the number **PIN 951/36**, and a large left humerus with the numbers **PIN 951/31** and **PIN 951/36**.

Specimen number	Humerus length (mm)	Maximum width, proximal end (mm)	Maximum width, distal end (mm)	Minimum shaft circumference (mm)
PIN 951/62	153.1	85.7	86.9	83
PIN 951/65-40	177.0	107.4	89.4	84
PIN 951/65-37	165.0	93.0	88.8	83
PIN 951/36	Incomplete	106.8	Incomplete	94
PIN 951/31	193.0	123	98.7	107

Ulnae

Two right ulnae of slightly different lengths are present. **PIN 951/15** is the larger element and is better preserved; **PIN 951/42** is slightly smaller, and is glued to a radius (**PIN 951/16**) from an old mount.

Specimen number	Length (mm)	Maximum proximal width (mm)	Maximum distal width (mm)
PIN 951/15	154.1	53.1	36.7
PIN 951/42	141.6	53.2	28.5

Radii

Two right radii are present. **PIN 951/16** is a larger specimen and is glued to an ulna for an old mount. It was mounted upside down. The other radius is small and unnumbered.

Specimen number	Length (mm)	Proximal width (mm)	Distal width (mm)
PIN 951/16	136.6	48.6	32.9
Unnumbered	112.2	40.1	28.8

Ilia

In the collection there are seven ilia: **PIN 951/49** and **PIN 951/48** a pair of ilia from a single individual; **PIN 951/8**, a pair of ilia from a single individual; a second pair of ilia from a single individual with the left also bearing the number **PIN 951/24**; and a single ilium missing its postacetabular process.

Specimen number	Preacetab-postacetab process length (mm)	Postacetabular length (mm)	Maximum height (mm)
PIN 951/49 left	Incomplete	76	Incomplete
PIN 951/48 right	138	80	107.6
PIN 951/8 left mounted (951/24)	156	83	125
PIN 951/8 right mounted	158.7	85	122
PIN 951/8 right	134	69.6	122.5
PIN 951/8 left	138.6	68.3	118.6
PIN 951/8 incomplete	Incomplete	Incomplete	115

Pubes

Six pubes and two additional iliac peduncles are present. **PIN 951/1** and an unnumbered fragment are the two isolated iliac peduncles. **PIN 951/5** are a pair of pubes which were part of the mount and another left pubis; **PIN 951/25** is a left pubis; **PIN 951/53** is a partial right pubis that may be the same individual as **PIN 951/52**; **PIN 951/52** is a partial left pubis.

Specimen number	Maximum length (mm)	Maximum width (mm)
PIN 951/5 right (mounted)	127.4	65.4
PIN 951/5 left (mounted)	134	75.5
PIN 951/5 left	123.6	Incomplete
PIN 951/25	119.1	Incomplete
PIN 951/53	113.7	Incomplete
PIN 951/52	109.5	Incomplete

Ischia

Six ischia, corresponding to three pairs, are present. Two ischia are glued together for the mount. The left is labelled as **PIN 951/1** and **951/65-43**, the right is labelled as **PIN 951/1** and **951/26**. These are the largest. A smaller pair are also labelled **PIN 951/1**. The final pair are **PIN 951/51** (right) and **PIN 951/50** (left). These articulate with ilia and pubis with sequential numbers and are clearly from a single individual.

Specimen number	Length (mm)
PIN 951/51	150.6
PIN 951/50	Incomplete
PIN 951/1 (mounted) left	185
PIN 951/1 (mounted) right	190
PIN 951/1 left	157.7
PIN 951/1 right	160.4

Figure S3. Articulating pelvis, PIN 951/8, 951/51 and 951/25. A, left lateral and B, left medial views. Scale bar equal to 50 mm.

Femora

There are two femora and an additional distal end in the collection. A large right femur **PIN 951/27**, a small left femur **PIN 951/61**, and the distal end **PIN 951/6**.

Specimen number	Length (mm)	Prox width (mm)	Dist width (mm)	Minimum circumference (mm)
PIN 951/27	243	71	87	106
PIN 951/61	211	57	Incomplete	84

Tibiae

There are two tibiae in the collections. Identification is difficult but **PIN 951/28** is a right tibia and **PIN 951/43** is a smaller left element with a fibula glued to it. The fibula is glued in the wrong place.

Specimen number	Length (mm)	Proximal width (mm)	Distal width (mm)
PIN 951/28	217	74	48
PIN 951/43	200	65	45

Fibulae

There are two fibulae in the collections. One is glued to a tibia in the wrong orientation, is a right element and has the number **PIN 951/29**. The other is larger, right element, and is **PIN 915/30**.

Specimen number	Length (mm)	Proximal width (mm)	Distal width (mm)
PIN 951/30	225	35	39
PIN 951/29	197	32	Incomplete

Metatarsals

Specimen number	Length (mm)	Proximal width (mm)	Minimum transverse width of shaft (mm)	Distal width (mm)
PIN 951/86	74	31	11	23
PIN 951/84	68	28	10	21
PIN 951/87	70	27	22	23
PIN 951/85	59	29	18	18
PIN 951/88	51	22	10	18
PIN 951/31	49	31	11	19
PIN 951/33	40	31	12	20
PIN 951/32	41	28	15	18
PIN 951/34	35	26	14	16

Phalanges

Specimen number	Length (mm)	Proximal width (mm)	Minimum transverse width of shaft (mm)	Distal width (mm)
PIN 951/35	29	19	10	17
PIN 951/36	22	16	9	13
PIN 951/37	20	15	9	13
PIN 951/89	-	21	-	-
PIN 951/90	30	23	15	17
PIN 951/91	29	20	11	15
PIN 951/92	30	19	10	15
PIN 951/93	27	21	14	17
PIN 951/94	24	19	10	16
PIN 951/95	24	18	10	16
PIN 951/96	30	16	9	13
PIN 951/97	22	17	9	12
PIN 951/98	19	15	9	15
PIN 951/99	17	16	10	14
PIN 951/101	16	13	8	11
PIN 951/102	18	13	9	11
PIN 951/103	15	13	9	11
PIN 951/104	23	8	8	10
PIN 951/105	14	9	5	5

Ungual phalanges

Specimen number	Length (mm)
PIN 951/21	32
PIN 951/86	18
PIN 951/29	25
PIN 951/78	22

Appendix B**ROYAL SOCIETY
OPEN SCIENCE****The postcranial skeleton of the erythrosuchid
archosauriform *Garjainia prima* from the Early Triassic of
European Russia**

Journal:	Royal Society Open Science
Manuscript ID	RSOS-201089
Article Type:	Research
Date Submitted by the Author:	17-Jun-2020
Complete List of Authors:	Maidment, Susannah; Natural History Museum, Department of Earth Sciences; University of Birmingham, School of Geography, Earth and Environmental Sciences Sennikov, Andrey G.; Borissiak Paleontological Institute of the Russian Academy of Sciences; Kazan Federal University, Institute of Geology and Petroleum Technologies Ezcurra, Martín; CONICET, Sección Paleontología de Vertebrados, Museo Argentino de Ciencias Naturales Dunne, Emma; University of Birmingham, School of Geography, Earth and Environmental Sciences Gower, David; Natural History Museum, Department of Life Sciences Hedrick, Brandon; Louisiana State University Health Sciences Center, Department of Cell Biology and Anatomy Meade, Luke E.; University of Birmingham, School of Geography, Earth and Environmental Sciences Raven, Thomas; Natural History Museum, Department of Earth Sciences; University of Brighton, School of Environment and Technology Paschchenko, Dmitriy I.; Borissiak Paleontological Institute of the Russian Academy of Sciences Butler, Richard; University of Birmingham, School of Geography, Earth and Environmental Sciences
Subject:	palaeontology < BIOLOGY, Palaeontology < EARTH SCIENCES
Keywords:	Erythrosuchidae, Permo-Triassic mass extinction, osteology, Vjushkovia triplicostata
Subject Category:	Organismal and Evolutionary Biology

**Author-supplied statements**

Relevant information will appear here if provided.

***Ethics***

*Does your article include research that required ethical approval or permits?:*

This article does not present research with ethical considerations

*Statement (if applicable):*

CUST_IF_YES_ETHICS :No data available.

***Data***

*It is a condition of publication that data, code and materials supporting your paper are made publicly*
*available. Does your paper present new data?:*

Yes

*Statement (if applicable):*

All data associated with this work is included in Electronic Supplementary Material.

***Conflict of interest***

I/We declare we have no competing interests

*Statement (if applicable):*

CUST_STATE_CONFLICT :No data available.

***Authors' contributions***

This paper has multiple authors and our individual contributions were as below

*Statement (if applicable):*

RJB and AGS designed the study. All authors contributed to data collection. SCRM, AGS, MDE, BPH,
and RJB analysed the data. All authors contributed to the manuscript.

**The postcranial skeleton of the erythrosuchid archosauriform *Garjainia prima***
**from the Early Triassic of European Russia**

Susannah C. R. Maidment^{1,2*}, Andrey G. Sennikov^{3,4}, Martín D. Ezcurra^{2,5}, Emma M. Dunne²,
David J. Gower¹, Brandon P. Hedrick⁶, Luke E. Meade², Thomas J. Raven^{1,7}, Dmitriy I.
Paschchenko³, Richard J. Butler²

- 1. The Natural History Museum, Cromwell Road, London SW7 5BD, United Kingdom
2. School of Geography, Earth and Environmental Sciences, University of Birmingham, Edgbaston,
Birmingham, B15 2TT, United Kingdom
3. Borissiak Paleontological Institute RAS, Profsoyuznaya Street 123, Moscow 117647, Russia
4. Institute of Geology and Petroleum Technologies, Kazan Federal University, Kremlyovskaya
Street 4, Kazan 420008, Russia
5. Sección Paleontología de Vertebrados, CONICET—Museo Argentino de Ciencias Naturales
“Bernardino Rivadavia”, Ángel Gallardo 470 (C1405DJR), Buenos Aires, Argentina
6. Department of Cell Biology and Anatomy, School of Medicine, Louisiana State University Health
Sciences Center, New Orleans, LA 70112, USA.
7. School of Environment and Technology, University of Brighton, Lewes Road, Brighton BN2 4GJ,
United Kingdom

[*susannah.maidment@nhm.ac.uk](mailto:susannah.maidment@nhm.ac.uk)

**Abstract**

Erythrosuchidae were large-bodied, quadrupedal, predatory archosauriforms that
dominated the hypercarnivorous niche in the aftermath of the Permo-Triassic mass
extinction. *Garjainia*, one of the oldest members of the clade, is known from the late
Olenekian of European Russia. The holotype of *Garjainia prima* comprises a well-preserved
skull, but highly incomplete postcranium. Recent taxonomic reappraisal demonstrates that
material from a bone bed found close to the type locality, previously referred to as
'*Vjushkovia triplicostata*', is referable to *G. prima*. At least seven individuals comprising
cranial remains and virtually the entire postcranium are represented, and we describe this
material in detail for the first time. An updated phylogenetic analysis confirms previous
results that a monophyletic *Garjainia* is the sister taxon to a clade containing *Erythrosuchus*,
*Shansisuchus* and *Chalishevia*. Muscle scars on many limb elements are clear, allowing
reconstruction of the proximal locomotor musculature. We calculate the body mass of *G.*
*prima* to have been 147–248 kg, similar to that of an adult male lion. Large body size in
erythrosuchids may have been attained as part of a trend of increasing body size after the
Permo-Triassic mass extinction, and allowed erythrosuchids to become the dominant
carnivores of the Early and Middle Triassic.

**Keywords:** Erythrosuchidae; Permo-Triassic mass extinction; osteology; *Vjushkovia*
*triplicostata*

**Introduction**

Erythrosuchidae is a clade of archosauriform archosauromorph reptiles that radiated to occupy the
apex terrestrial predator niche in the aftermath of the devastating Permo-Triassic mass extinction
(PTME; Raup 1979; Erwin 1994; Ezcurra & Butler 2018). Large-bodied (2–5 m in length) and
quadrupedal, erythrosuchids possessed large skulls relative to their body size (Gower 2003; Ezcurra
et al. 2013; Butler et al. 2019a). A large cranium appears to have evolved in concert with other
carnivorous adaptations (e.g., dorsoventrally tall and subrectangular cranial profile), and is perhaps
associated with the clade evolving hypercarnivory (Butler et al. 2019a). Erythrosuchidae was one of
the earliest archosauromorph groups to diversify after the PTME, and comprises seven to nine valid
species known from the latest Early Triassic to Middle Triassic of South Africa, China, and Russia
(Parrish 1992; Gower 2003; Ezcurra et al. 2013; Gower et al. 2014; Ezcurra et al. 2019a; Butler et al.
2019a, b, c), with undescribed material also reported from India (Bandyopadhyay 1988). The earliest
members of the group are *Garjainia prima* and *G. madiba*, from the late Olenekian of Russia and
South Africa, respectively, and *Fugusuchus hejapanensis* from the late Olenekian or early Anisian of
China (Ochev 1958; Gower & Sennikov 2000; Gower et al. 2014; Ezcurra et al. 2019a; Butler et al.
2019a).

The holotype of *Garjainia prima* is a well-preserved specimen comprising a skull and an incomplete
postcranium from Kzyl-Sai II, near Orenburg, Russia, and has received detailed study (Ochev 1958;
Gower & Sennikov 2000; Ezcurra et al. 2019a). A second erythrosuchid, '*Vjushkovia triplicostata*',
from Rassypnaya, also close to Orenburg, and from a similar stratigraphic horizon to the holotype of
*G. prima*, is represented by the cranial and postcranial remains of at least seven individuals (Huene
1960; Tatarinov 1961; Ochev 1975, 1981; Sennikov 1995, 2008; Clark et al. 1993; Gower 1996;
Gower & Sennikov 1996a, b; Butler et al. 2019a). Butler et al. (2019a) recently formally synonymised
*Garjainia prima* and '*Vjushkovia triplicostata*', referring all material previously described as *V.*
*triplicostata* to *G. prima* (see also Gower & Sennikov 2000; Ezcurra et al. 2019a). As a result, *G. prima*
is now one of the most completely known archosauromorphs from the Early Triassic. Butler et al.

(2019a) described the skull material referred to *G. prima*, but the postcranial material of the
hypodigm of '*V. triplicostata*' has previously not been comprehensively or thoroughly described and,
with exception of the tarsus, has been figured only as line drawings (Huene 1960; Gower 1996).
Here, we describe and figure in detail the extensive and well-preserved postcranial material of '*V.*
*triplicostata*' for the first time, and use it to reassess phylogenetic affinities, make the first
assessments of erythrosuchid body mass, and to reconstruct locomotor musculature.

**Systematic Palaeontology**

Diapsida Osborn, 1903 *sensu* Laurin (1991)

Sauria Gauthier, 1984 *sensu* Gauthier, Kluge & Rowe (1988)

Archosauromorpha Huene, 1946 *sensu* Dilkes (1998)

Archosauriformes Gauthier, Kluge & Rowe (1988)

Erythrosuchidae Watson, 1917 *sensu* Ezcurra et al. (2010)

*Garjainia* Ochev, 1958

***Garjainia prima* Ochev, 1958**

**Holotype:** PIN 2394/5, a partial skeleton of a single individual (Ochev 1958; Ezcurra et al. 2019a).

**Referred specimens:** Almost the entire skeleton, with the exception of some caudal vertebrae and
elements from the carpus and manus, from a bone bed of a single locality. The quarry was
approximately 100 m x 50 m in area. The material is representative of at least seven different
individuals of different sizes (Huene 1960; Gower & Sennikov 2000; Ezcurra et al. 2019a; see Online

Supplementary Material for full list of postcranial material; see Butler et al. 2019a for details of
cranial material).

[revised manuscript text omitted]

individual because it appears to be the most complete, but there is no way to be certain unless
further details of the original excavation emerge. Herein we make the assumption that this large
postcrania belongs to a single individual based on proportional similarity with *Erythrosuchus*
*africanus*, and base our body mass estimates on these elements (see below).
(4) A pair of fused scapulocoracoids, one with an associated clavicle (PIN 951/65-33 and -34) are
from the same individual as the large right humerus (PIN 951/36) based on size and
preservational appearance.
(5) A pair of medium-sized scapulae (PIN 951/4 and PIN 951/7) with articulating but unfused
coracoids (PIN 951/2 and PIN 951/3) are probably from the same individual as a pair of humeri
(PIN 951/65-37 and PIN 951/65-40) based on size and preservation. Based on size alone, these
may be from the same individual as the previously mounted ulna and radius (PIN 951/42 and PIN
951/16).
(6) A very small partial coracoid (unnumbered), scapula (PIN 951/4) and humerus (PIN 951/36) are
from the same individual based on size and preservation.
(7) A pair of ilia (PIN 951/8) articulate well with a left pubis (PIN 951/25) and a pair of ischia (PIN
951/1) and are clearly associated.
(8) Articulating pelvic elements from both sides numbered sequentially from PIN 951/48 – 53 are
clearly from the same individual and, based on preservational appearance, the smaller femur
(PIN 951/61) might also belong to this individual.

**Phylogenetic analysis**

*Methods*—We checked and updated the character scorings of *Garjainia prima* following the new
information provided by Butler et al. (2019a) and this paper, as well as incorporated revised scorings

for *Chalishevia cothurnata* and *Shansisuchus shansisuchus* following Butler et al. (2019c), in the most
extensive phylogenetic dataset currently available for Permian and Triassic archosauromorphs
(Ezcurra 2016, as modified by subsequent authors). We worked on the latest version of this data
matrix, which was recently published by Scheyer et al. (2020) to test the phylogenetic position of
*Colobops noviportensis*. In particular, the scorings of the lower jaw and postcranium of *Chalishevia*
*cothurnata* were changed to missing data because the elements belonging to these regions have
been recently reinterpreted as belonging to an indeterminate erythrosuchid rather than to the
hypodigm of the species (Butler et al. 2019c). We modified character 393 and changed some
scorings accordingly (Supplementary Information). We deactivated 35 terminals before the tree
searches, which were included by Ezcurra & Butler (2018) only for the purpose of morphological
disparity analyses, and also removed character 119 (following Ezcurra et al. [2017] and Butler et al.
[2019b]), resulting in a data set of 121 active terminals and 710 characters. The matrix is available in
Online Supplementary Material.

[revised manuscript text omitted]

(ca. 1.4 tonnes), and *Carnoaurus sastrei* and *Majungasaurus crenatissimus* (ca. 1.6 tonnes; Benson
et al. 2014). Mass estimates are not currently possible for other Middle Triassic erythrosuchids, such
as *Chalishevia cothurnata* (Butler et al. 2019c) and *Shansisuchus shansisuchus* (Wang et al. 2013),
but they were also large-bodied (e.g. skull length of 80 cm in *Chalishevia*: Butler et al. 2019c). The
large body sizes of *Garjainia prima*, *Erythrosuchus africanus*, and other Middle Triassic
erythrosuchids may have been attained as part of the trend in increasing body size after the PTME,
and they radiated to dominate the hypercarnivorous niche of their ecosystems, being the largest
predators of the Early and early Middle Triassic.

**Acknowledgements**

Travel to the PIN for RJB, SCR, LEM, and EMD was funded by a Royal Society International
Exchange Grant (IEC\R2\170064) to RJB and AGS, and an earlier visit for RJB, MDE and DJG was

funded by the DFG Emmy Noether Programme (BU 2587/3-1 to RJB). AGS was co-funded by awards
from the Royal Society International Exchange Grant (IEC\R2\170064), and the Russian Foundation
for Basic Research (RFBR No. 17-54-10013) during the course of this work. AGS and DIP were funded
by the Russian Foundation for Basic Research through the research project No. 20-04-00070. AGS
was also funded by a subsidy of the Russian Government to support the Program of “Competitive
Growth of Kazan Federal University among World’s Leading Academic Centers”. Participation of BPH
in this work was funded by the European Union’s Horizon 2020 research and innovation program
2014–2018 under grant agreement 677774 (European Research Council [ERC] Starting Grant: TEMPO
awarded to Roger Benson [University of Oxford]). TJR was funded by a University of Brighton Science
Studentship. MDE is funded by Agencia Nacional de Promoción Científica y Técnica (PICT 2018-
01186). DJG thanks Mike Benton (University of Bristol) for supervising his NERC-funded research on
erythrosuchids during which he first examined *Garjainia prima*.

**References**

1. Raup DM. 1974 Size of the Permo-Triassic bottleneck and its evolutionary implications. *Science*
**206**, 217–218.
2. Erwin DH. 1994 The Permo-Triassic extinction. *Nature* **367**, 231–236.
3. Ezcurra MD, Butler RJ. 2018 The rise of the ruling reptiles and ecosystem recovery from the
Permian-Triassic mass extinction. *Proc. Roy. Soc. Lond. B.* **285**, 20180361. (doi:
10.1098/rspb.2018.0361).
4. Gower DJ. 2003 Osteology of the early archosaurian reptile *Erythrosuchus africanus* Broom. *Ann.*
*South Afr. Mus.* **110**, 1–84.
5. Ezcurra MD, Butler RJ, Gower DJ. 2013 ‘Proterosuchia’: the origin and early history of
Archosauriformes. In Nesbitt SJ, Desojo JB, Irmis RB (eds). *Anatomy, phylogeny and*
*palaeobiology of early archosaurs and their kin*. Geological Society Special Publication **379**,
9–33

6. Butler RJ, Sennikov AG, Dunne EM, Ezcurra MD, Hedrick BP, Maidment SCR, Meade LE, Raven TJ,
Gower DJ. 2019a Cranial anatomy and taxonomy of the erythrosuchid archosauriform
‘*Vjushkovia triplicostata*’ Huene, 1960, from the Early Triassic of European Russia. *Roy. Soc.*
*Open Sci.* **6**, 191289. (doi: 10.1098/rsos.191289).
7. Parrish JM. 1992 Phylogeny of the Erythrosuchidae (Reptilia: Archosauriformes). *J. Vertebr.*
*Paleontol.* **12**, 93–110.
8. Gower DJ, Hancox PJ, Botha-Brink J, Sennikov AG, Butler RJ. 2014 A new species of *Garjainia*
Ochev, 1958 (Diapsida: Archosauriformes: Erythrosuchidae) from the Early Triassic of South
Africa. *PLoS One* **9**, e111154. (doi: 10.1371/journal.pone.0111154).
9. Ezcurra MD, Gower DJ, Sennikov AG, Butler RJ. 2019a The osteology of the holotype of the early
erythrosuchid *Garjainia prima* (Diapsida: Archosauromorpha) from the upper Lower Triassic
of European Russia. *Zoo. J. Linn. Soc.* **185**, 717–783.
10. Butler RJ, Ezcurra MD, Liu J, Sookias RB, Sullivan C. 2019b. The anatomy and phylogenetic
position of the erythrosuchid archosauriform *Guchengosuchus shiguaiensis* from the earliest
Middle Triassic of China. *PeerJ* **7**, e6435. (doi: 10.7717/peerj.6435).
11. Butler RJ, Sennikov AG, Ezcurra MD, Gower DJ. 2019c The last erythrosuchid? A revision of
*Chalishevia cothurnata* Ochev, 1980 from the late Middle Triassic of European Russia. *Acta*
*Palaeontol. Pol.* **64**. (doi: 10.4202/app.00648.2019).
12. Bandyopadhyay S. 1988 Vertebrate fossils from the Pranhita-Goavari Valley of India with special
reference of the Yerrapalli Formation. *Modern Geology* **13**, 107–117.
13. Ochev VG. 1958. New data concerning the Pseudosuchia of the USSR. *Dokl. Akad. Nauk.* **123**,
749–751.
14. Gower DJ, Sennikov AG. 2000 Early archosaurs from Russia. In Benton MJ, Kurochkin EN, Shishkin
MA, Unwin DM (eds). *The age of dinosaurs in Russia and Mongolia*. Pp. 140–159.
15. Huene F. 1960 Ein grosser Pseudosuchier aus der Orenburger Trias. *Palaeontogr. Abt. A* **114**,
105–111.

16. Tatarinov LP. 1961 Pseudosuchians of the USSR. *Paleontol. J.* **1961**, 117–132. [In Russian].
17. Ochev VG. 1975 The palate in the Proterosuchia. *Paleontol. J.* **1975**, 98–105.
18. Ochev VG. 1981 On *Erythrosuchus (Garjainia) primus* Ochev. *Voprosy Geologii Yuzhnogo Urala I*
*Povolzh'ya* **22**, 3–22. [In Russian].
19. Sennikov AG. 1995 Early thecodonts of Eastern Europe. *Trudy Paleontologicheskogo Instituta*
*RAN* **263**, 1–141. [In Russian].
20. Sennikov AG. 2008 Subclass Archosauromorpha. In *Fossil Vertebrates from Russia and Adjacent*
*Countries: Fossil Reptiles and Birds. Part 1.* (eds Ivakhnenko MF, Kurochkin EN) Moscow:
GEOS: 266–318. [In Russian].
21. Clark JM, Welman J, Gauthier JA, Parrish JM. 1993 The laterosphenoid bone of early
archosauriforms. *J. Vertebr. Paleontol.* **13**, 48–57. (doi:10.1080/02724634.1993.10011487)
22. Gower DJ. 1996 The tarsus of erythrosuchid archosaurs, and implications for early diapsid
phylogeny. *Zoo. J. Linn. Soc.* **116**, 347–375.
23. Gower DJ, Sennikov AG. 1996 Morphology and phylogenetic informativeness of early archosaur
braincases. *Palaeontology* **39**, 883–906.
24. Gower DJ, Sennikov AG. 1996 Endocranial casts of early archosaurian reptiles. *Paläontol. Zeit.* **70**,
579–589.
25. Osborn HF. 1903 The reptilian subclasses Diapsida and Synapsida and the early history of the
Diaptosauria. *Mem. Am. Mus. Nat. Hist.* **1**, 449–519.
26. Laurin M. 1991 The osteology of a Lower Permian eosuchian from Texas and a review of diapsid
phylogeny. *Zoo. J. Linn. Soc.* **101**, 59–95.
27. Gauthier JA. 1984 A cladistic analysis of the higher categories of Diapsida. Unpublished PhD
thesis, University of California.
28. Gauthier JA, Kluge AG, Rowe T. 1988 Amniote phylogeny and the importance of fossils. *Cladistics*
**4**, 105–209.

29. Huene F. 1946 Die grossen Stämme der Tetrapoden in den geologischen Zeiten. *Biologisches*
*Zentralblatt* **65**, 268–275.
30. Dilkes DW. 1998 The Early Triassic rhynchosaur *Mesosuchus browni* and the interrelationships
of basal archosauromorph reptiles. *Phil. Trans. Roy. Soc. Lond. B* **353**, 501–541.
31. Watson DMS. 1917 A sketch classification of the Pre-Jurassic tetrapod vertebrates. *Proc. Zoo.*
*Soc. Lond.* **1917**, 167–186.
32. Ezcurra MD, Lecuona A. Martinelli A. 2010 A new basal archosauriform diapsid from the lower
Triassic of Argentina. *J. Vertebr. Paleontol.* **30**, 1433–1450.
33. Nesbitt SJ, Flynn JJ, Prichard AC, Parrish JM, Ranivoharimanana L, Wyss AR. 2015 Postcranial
anatomy and relationships of *Azendohsaurus madagaskarensis*. *Bull. Am. Mus. Nat. Hist.*
**398**, 1–126.
34. Ewer RF. 1965 The anatomy of the thecodont reptile *Euparkeria capensis* Broom. *Phil. Trans. Roy.*
*Soc. Lond. B* **751**, 379–435.
35. Young C-C. 1964 The pseudosuchians in China. *Pal. Sin. New Series, C* **151**, 1–205.
36. Ezcurra MD. 2016. The phylogenetic relationships of basal archosauromorphs, with an emphasis
on the systematics of preterosuchian archosauriforms. *PeerJ* **4**, e1778. (doi:
10.7717/peerj.1778).
38. Ezcurra MD, Scheyer T, Butler RJ. 2014 The origin and early evolution of Sauria: reassessing the
Permian saurian fossil record and the timing of the crocodile-lizard divergence. *PLOS ONE* **9**,
e89165.
39. Sengupta S, Ezcurra MD, Bandyopadhyay S. 2017 A new horned and long-necked herbivorous
stem-archosaur from the Middle Triassic of India. *Sci. Rep.* **7**, 8366.
40. Nesbitt SJ, Butler RJ, Ezcurra MD, Barrett PM, Stocker MR, Angielczyk KD, Smith RMH, Sidor CA,
Niezwiedzki G, Sennikov AG, Charig AJ. 2017. The earliest bird-line archosaurs and the
assembly of the dinosaur body plan. *Nature* **544**, 484–487.

41. Gower DJ, Schoch R. 2009 Postcranial anatomy of the rauisuchian archosaur *Batrachotomus*
*kupferzellensis*. *J. Vert. Paleontol.* **17**, 60–73.
42. Gower DJ. 2001 Possible postcranial pneumaticity in the last common ancestor of birds and
crocodylians: evidence from *Erythrosuchus* and other Mesozoic archosaurs.
*Naturwissenschaften* **88**, 119–122.
43. Butler RJ, Barrett PM, Gower DJ. 2012 Reassessment of the evidence for postcranial skeletal
pneumaticity in Triassic archosaurs, and the early evolution of the avian respiratory system.
*PLoS ONE* **7**, e34094. Doi: 10.1371/journal.pone.0034094.
44. Wang R, Xu S, Wu X, Li C, Wang S. 2013 A new specimen of *Shansisuchus shansisuchus* Young,
1964 (Diapsida: Archosauriformes) from the Triassic of Shanxi, China. *Acta Geol. Sin.* **87**,
1185–1197.
45. Gow CE. 1975 The morphology and relationships of *Youngina capensis* Broom and *Prolacerta*
*broomi* Parrington. *Pal. Afr.* **18**, 89–131.
46. Nesbitt, S. J., R. J. Butler, M. D. Ezcurra, A. J. Charig, and P. M. Barrett. 2018. The anatomy of
*Teleocrater rhadinus*, an early avemetatarsalian from the lower portion of the Lifua Member
of the Manda Beds (Middle Triassic); pp. 142–177 in C. A. Sidor and S. J. Nesbitt (eds.),
Vertebrate and Climatic Evolution in the Triassic Rift Basins of Tanzania and Zambia. Society
of Vertebrate Paleontology Memoir 17.
47. Ezcurra MD, Jones AS, Gentil AR, Butler RJ. 2020. Early archosauromorphs: the crocodile and
dinosaur precursors. Encyclopedia of Geology, 2nd edition. Reference Module in Earth
Systems and Environmental Sciences. <https://doi.org/10.1016/B978-0-12-409548-9.12439-X>.
48. Nesbitt SJ. 2011 The early evolution of archosaurs: relationships and the origin of major clades.
*Bull. Am. Mus. Nat. Hist.* **352**, 1–292.
49. Maidment SCR, Barrett PM. 2011 The locomotor musculature of basal ornithischian dinosaurs. *J.*
*Vertebr. Paleontol.* **31**, 1265–1291.

50. Peng J-H. 1991 A new genus of Proterosuchia from the Lower Triassic of Shaanxi, China. *Vert.*
*PalAsiatica* **29**, 95–107.
51. Romer AS. 1956 *Osteology of the Reptiles*. Chicago, University of Chicago Press.
52. Romer AS. 1972 The Chañares (Argentina) Triassic reptile fauna. XVI. Thecodont classification.
*Breviora* **395**, 1–24.
53. Rusconi C. 1951 Rastros de patas de reptiles Permicos de Mendoza. *Rev. Soc. Hist. Geog. Cuyo* **3**,
1–14.
54. Hutchinson JR. 2001a The evolution of pelvic osteology and soft tissues on the line to extant
birds (Neornithes). *Zoo. J. Linn. Soc.* **131**, 123–168.
55. Irmis, R. B., Nesbitt, S. J., Padian, K., Smith, N. D., Turner, A. H., Woody, D., & Downs, A. (2007). A
Late Triassic dinosauro-morph assemblage from New Mexico and the rise of dinosaurs.
*Science*, 317(5836), 358–361.
56. Hutchinson JR. 2001b The evolution of femoral osteology and soft tissues on the line to extant
birds (Neornithes). *Zoo. J. Linn. Soc.* **131**, 169–197.
57. Rieppel O. 1989. The hind limb of *Macrocnemus bassanii* (Nopcsa) (Reptilia, Diapsida):
development and functional anatomy. *J. Vert. Paleontol.* **9**, 373–387.
58. Pritchard AC, Sues HD. 2019 Postcranial remains of *Teraterpeton hrynewichorum* (Reptilia:
Archosauromorpha) and the mosaic evolution of the saurian postcranial skeleton. *J. Syst.*
*Palaeont.* **17**, 1745–1765.
59. Scheyer TM, Spiekman SN, Sues H-D, Ezcurra MD, Butler RJ, Jones ME. 2020. *Colobops*: a juvenile
rhynchocephalian reptile (Lepidosauromorpha), not a diminutive archosauromorph with an
unusually strong bite. *Roy. Soc. Open Sci.* **7**, 192179.
60. Ezcurra MD, Fiorelli LE, Martinelli AG, Rocher S, von Baczko MB, Ezpeleta M, *et al.* 2017 Deep
faunistic turnovers preceded the rise of dinosaurs in southwestern Pangaea. *Nat. Ecol. Evol.*
**1**, 1477–1483. (doi: 1410.1038/s41559-41017-40305-41555)

[revised manuscript text omitted]

Appendix C

Responses to reviewer comments

Reviewer: 1

Comments to the Author(s)

This is a well written thorough description and discussion of an important set of specimens. The material is absolutely gorgeous. The authors provide numerous comparisons with other taxa and cite the specimen numbers making it clear where the comparisons were made. I only had a few comments/suggestions that are in the attached file. One concern is the identification of the processes on the apex of the neural spine as mammillary processes. My understanding is that in mammals these are on the neural arch and transverse processes. How are these homologous? Overall though I find the manuscript to be a solid contribution.

Page 8: any prep notes or comments? We don't have very much information about prep of the specimens unfortunately. However, we have previously provided a more detailed account of the discovery of the specimens in Butler et al. 2019a, which we reference here.

Page 11: typo corrected

Page 20 and above: Mammillary processes? The use of the term "mammillary process" for a lateral expansion adjacent to the dorsal margin of the neural spine is not new for the diapsid literature, dating back more than 50 years (e.g. description of the Permian diapsid *Araeoscelis*; Vaughn, 1955: Bulletin of the Museum of Comparative Zoology). Herein, the term is used in a purely descriptive sense, rather than to infer homology with any feature in mammals. We have added "(sensu Ezcurra 2016)" after the first time we use this terminology to make the sense in which we are using this term clearer.

Page 25 suggested word change to 'adhered' corrected

Page 26 are sutures still visible? Information added

Page 42: It's amazing how much this looks like a tibia. You almost need to differentiate the two. The ulna has a clear although small olecranon process, that we describe. The tibia is also more greatly expanded proximally and more anteroposteriorly compressed, as well as having a clear pit for the *m. puboischiotibialis*. We compared these elements with other archosauromorphs and they are quite similar in morphology.

Page 57: but you don't figure DT4 below. Corrected.

Page 76: Caps? Corrected

Supplementary data: what are A–C? Distinguish vertebrae. Labelled A–C in the figure caption and numbered vertebrae.

Reviewer: 2

Comments to the Author(s)

This work is a description of the postcranial skeleton of *G. prima*. It provides detailed anatomical information and gives a reconstruction on musculature. It is well written and I nearly have nothing to say.

Line 54 add the original publication (Cheng, 1980) for *Fugusuchus*

Added

Line 1261 Fig25 the age of *Shansisuchus* may be limited to Anisian, no Ladinian based on recent zircon dating (Liu et al., 2018)

The *Shansisuchus*-bearing levels are constrained between ca. 243.53 Ma and ca. 241.48 Ma based on radioisotopic dates (which ranges from the Anisian to the Ladinian). As far as we are aware, it cannot be determined if the Anisian-Ladinian boundary occurs within the Ermaying Formation or Tongchuan Formation. The former unit has yielded *Shansisuchus shansisuchus* and the latter *Shansisuchus* sp. The thick black lines of the time calibrated cladogram represent chronostratigraphic uncertainties, not actual ranges, so on the figure the range of *Shansisuchus shansisuchus* is extended into the Ladinian to reflect this uncertainty. We have added the following sentence in the figure caption of figure 25: "The thick black lines represent chronostratigraphic uncertainty in the age of the species rather than actual temporal ranges."

I suggest the Reconstruction of locomotor musculature could be expanded and give more elaboration.

Without further guidance about what aspect of the reconstruction can be expanded, it is difficult to do this. We feel that we have gone as far with this as we can based on the available data. Because the ends of the limbs were not ossified, it is not possible to infer how the limbs were articulated, and therefore muscle lines of action and moment arms would be highly speculative. We do not wish to contribute unfounded speculation to the literature.

Reviewer: 3

Comments to the Author(s)

The main contribution of the paper is the descriptive part which makes the paper worthy of publication.

Recovery after the Permo-Triassic extinction event is an interesting topic. In South Africa *Proterosuchus* is the first archosauromorph to appear and in later zones *Garjainia* and *Erythrosuchus*. In the light of this the Russian archosauromorphs from around the Permo-Triassic boundary is very important to decide whether erythrosuchids are derived proterosuchids or the other way round. This would be a real challenge for phylogenetic analysis in a more inclusive paper.

One of us (MDE) has published an in-depth phylogenetic analysis of the 'proterosuchian' archosauriforms (Ezcurra, MD 2016. They phylogenetic relationships of basal archosauromorphs, with an emphasis on the systematics of proterosuchian archosauriforms. *PeerJ* 4:e1778). The analysis we present here has its foundations in that work. We agree that new discoveries and re-analysis of previously-known specimens (as we have done here) will help to shed light on the early evolution of archosauromorphs.

This paper makes a major contribution in terms of the detailed description of skeletal elements. I view some of the interwoven suggestions such as that *Euparkeria* show miniature erythrosuchid features as speculative.

Throughout the paper we compare the osteology of *Garjainia* with a number of other archosauromorphs, of which *Euparkeria* is one. We have not suggested that *Euparkeria* shows miniature erythrosuchid characters; we have simply noted where features are shared and when the character states differ.

Anatomical evolution is to be studied in a comprehensive series of related forms. The section about the femoral evolution is commendable but lacks some more taxonomic way points for future studies.

Without more specific guidance, it is difficult to know what the reviewer would like us to add here. As stated in the responses to Reviewer 2, we are disinclined to speculate.

The calculations of body weight are interesting but invoking the lion analogy instead of a crocodile one for *Garjainia* is open to debate.

Body masses of extinct animals are often quoted in the literature but what this means relative to living animals is difficult to visualize. We used a lion because it is an extant apex predator that will be familiar to the majority of people who read this paper. It was simply an aid to help the reader understand the size of the animal in the context of something living. We don't feel a crocodile is a better analogy, particularly. From a mode-of-life point of view, crocs are aquatic, while *Garjainia* was terrestrial. The largest alligators reach ~170 kg in size, which is towards the lower end of the mass estimate for *Garjainia*, so potentially is a bit small. *Garjainia* was more closely related to crocodiles than to lions, but we don't think this is particularly relevant.